# Control of working memory by phase–amplitude coupling of human hippocampal neurons

Jonathan Daume[1,2,3✉], Jan Kamiński[1,4], Andrea G. P. Schjetnan[5], Yousef Salimpour[6], Umais Khan[1], Michael Kyzar[1], Chrystal M. Reed[2], William S. Anderson[6], Taufik A. Valiante[5], Adam N. Mamelak[1] & Ueli Rutishauser[1,2,3,7✉]

Retaining information in working memory is a demanding process that relies on cognitive control to protect memoranda-specific persistent activity from interference[1,2]. However, how cognitive control regulates working memory storage is unclear. Here we show that interactions of frontal control and hippocampal persistent activity are coordinated by theta–gamma phase–amplitude coupling (TG-PAC). We recorded single neurons in the human medial temporal and frontal lobe while patients maintained multiple items in their working memory. In the hippocampus, TG-PAC was indicative of working memory load and quality. We identified cells that selectively spiked during nonlinear interactions of theta phase and gamma amplitude. The spike timing of these PAC neurons was coordinated with frontal theta activity when cognitive control demand was high. By introducing noise correlations with persistently active neurons in the hippocampus, PAC neurons shaped the geometry of the population code. This led to higher-fidelity representations of working memory content that were associated with improved behaviour. Our results support a multicomponent architecture of working memory[1,2], with frontal control managing maintenance of working memory content in storage-related areas[3–5]. Within this framework, hippocampal TG-PAC integrates cognitive control and working memory storage across brain areas, thereby suggesting a potential mechanism for top-down control over sensory-driven processes.

Working memory (WM), the ability to maintain and manipulate a limited amount of information in mind for a brief period of time[6], is a crucial component of cognition that is often compromised in disease. WM maintenance is an active process that retains information that is no longer available in the external world. A mechanism that is thought to support WM is persistent neural activity[7–10]. In humans, memoranda-specific persistent activity has been observed in the human medial temporal lobe (MTL)[11,12], an area of the brain that becomes essential for WM when distractors are present or memory load is high[13]. It is thought that cognitive control is required to support the maintenance of WM content under these circumstances[1,2]. Models of WM assign the role of control to the frontal lobes[3,4,14], but little is known about how storage and control mechanisms interact.

A ubiquitous macroscopic electrophysiological phenomenon is TG-PAC[15–21]. Although its functional role remains poorly understood, a major hypothesis is that PAC enables the integration of local sensory information processing with brain-wide cognitive control[22,23]. Within this framework, local increases in power in the gamma-frequency range (30–140 Hz)[24–27] reflect local processing, whereas long-range interareal interactions in the theta range (3–7 Hz) mediate cognitive control[28–30]. TG-PAC could therefore serve as a tool to integrate these two processes in local circuitries[16,31,32]. However, to date, it remains unclear how these theories translate to single-neuron activity and how PAC exerts control over WM maintenance processes. Here we test the hypothesis that neurons whose activity is modulated by both theta phase and gamma amplitude are engaged in interareal interactions between the frontal and temporal lobes, thereby exerting PAC-mediated cognitive control over WM storage. We examined whether top-down control directly modulated the cells that carry information about the memoranda currently held in WM or whether, alternatively, control is exerted indirectly through a different group of cells.

We recorded single-cell and local field potential (LFP) activity from the medial frontal cortex and MTL while patients who had undergone neurosurgery performed a WM task (36 patients, 44 sessions; Supplementary Table 5) with pictures as stimuli. All of the pictures belonged to one of five visual categories. In each trial, the patients maintained either

[1]Department of Neurosurgery, Cedars-Sinai Medical Center, Los Angeles, CA, USA. [2]Department of Neurology, Cedars-Sinai Medical Center, Los Angeles, CA, USA. [3]Center for Neural Science and Medicine, Department of Biomedical Sciences, Cedars-Sinai Medical Center, Los Angeles, CA, USA. [4]Center of Excellence for Neural Plasticity and Brain Disorders: BRAINCITY, Nencki Institute of Experimental Biology, Polish Academy of Sciences, Warsaw, Poland. [5]Krembil Research Institute and Division of Neurosurgery, University Health Network (UHN), University of Toronto, Toronto, Ontario, Canada. [6]Department of Neurosurgery, Johns Hopkins School of Medicine, Baltimore, MD, USA. [7]Division of Biology and Biological Engineering, California Institute of Technology, Pasadena, CA, USA. ✉e-mail: jonathan.daume@cshs.org; ueli.rutishauser@cshs.org

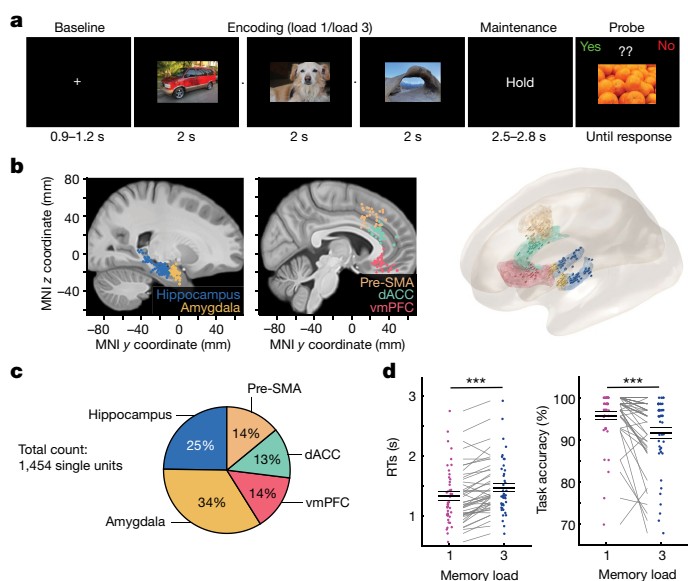

**Fig. 1 | Task, recording sites and behaviour. a**, An example trial. Each trial consisted of either one (load 1) and three (load 3) consecutively presented pictures, each presented for 2 s (separated by a variable blank screen of up to 200 ms as indicated by a small dot). After a variable maintenance period with an average duration of 2.7 s, a probe picture was presented. The task was to decide whether the probe picture has been part of the pictures shown during encoding in this trial (the correct answer was 'No' in the example shown). For copyright reasons, the pictures shown are similar but not identical to those used in the study. **b**, The recording locations. Each coloured dot represents the location of a microwire bundle across all 44 sessions shown on a standardized MNI152 brain template (left) and in a 3D model using the Brainnetome Atlas (right). The slices (https://osf.io/r2hvk/) were obtained under a Creative Commons licence CC BY 4.0. **c**, The proportions of neurons recorded in each brain area. **d**, The behaviour of the participants. Patients made fewer errors ($P = 0.0001$) and responded faster ($P = 0.0001$) in load 1 compared with load 3 trials. Statistical analysis was performed using two-sided permutation-based $t$-tests with 10,000 permutations. Each line connects the two dots belonging to the same session. $n = 44$ sessions. The RT was measured relative to the probe stimulus onset. Data are mean ± s.e.m. ***$P < 0.001$.

one (load 1) or three (load 3) consecutively presented pictures in their WM for 2.5–2.8 s (Fig. 1a). The patients were then asked whether a probe stimulus shown was identical to one of the item(s) they were holding in WM. Mean accuracy was 93.66 ± 7.04% (mean ± s.d.; 78.34 ± 20.09% of all errors were false negatives) and the participants responded slower (1.46 s versus 1.33 s; $t_{43} = 6.42$, $P < 0.001$) and less accurately (91.60% versus 95.71%; $t_{43} = -4.45$, $P < 0.001$) in load 3 trials compared with load 1 trials (Fig. 1d).

## Hippocampal PAC is modulated by WM load

We recorded from 1,454 single neurons (Fig. 1b,c) and from 1,922 microwire channels with LFP (Extended Data Fig. 1f,g) across the hippocampus, amygdala, pre-supplementary motor area (pre-SMA), dorsal anterior cingulate cortex (dACC) and ventromedial prefrontal cortex (vmPFC). To determine whether PAC was present in the LFP during the WM maintenance period, we estimated PAC as a function of low-frequency (2–14 Hz) phase and high-frequency (30–150 Hz) power. Across all of the recorded channels (Fig. 2a; average plots per brain area are shown in Extended Data Fig. 2a), PAC was strongest between the phase in the theta range (3–7 Hz) and the amplitude in two different gamma-frequency bands—a lower (30–55 Hz) and a higher gamma range (70–140 Hz). For each channel, we then separately averaged the normalized modulation indices in the theta to

low-gamma and the theta to high-gamma combinations and assessed which of the channels exhibited significant PAC across both load conditions in the given frequency combination (averaged $z$ score > 1.64; $P < 0.05$; right sided). For theta to high-gamma PAC, 137 out of 586 hippocampal channels showed significant PAC across all correct trials from both load conditions (Fig. 2b). For these 137 channels, we then compared the PAC estimates between the two load conditions. This comparison revealed significantly weaker theta–high-gamma PAC in load 3 compared with load 1 trials ($t_{136} = -4.26$, $P < 0.001$; Fig. 2c (left); false-discovery rate (FDR) corrected for the five brain regions of interest[33]; Fig. 2d shows PAC for a single example channel in the hippocampus; Fig. 2e shows the average PAC across all significant PAC channels in the hippocampus).

Two other brain areas also exhibited substantial proportions of channels with significant PAC: the amygdala and the vmPFC (Fig. 2b). However, in contrast to the hippocampus, theta–high-gamma PAC did not differ significantly between the two load conditions in neither of these two areas (Fig. 2c; $t_{129} = 1.43$, $P = 0.38$, Bayes factor (BF)$_{01} = 1.40$; and $t_{39} = 0.16$, $P = 0.87$, BF$_{01} = 5.84$, respectively). In the other two frontal areas that we examined (pre-SMA and dACC), only a small proportion of channels had significant theta–high-gamma PAC (Fig. 2b) and PAC in these channels did not differ significantly between the two loads (pre-SMA: $t_4 = -0.16$, $P = 0.87$; dACC: $t_{13} = -0.82$, $P = 0.73$). These observations were qualitatively comparable when averaging channels within each patient (Extended Data Fig. 2b). For PAC involving the low gamma band (30–55 Hz), there were no significant differences between the load conditions in any of the regions (Extended Data Fig. 2c and Supplementary Table 1). Next, we asked whether PAC is associated with reaction times (RTs), with the idea that faster RTs indicate success of control. In the hippocampus, faster RTs were associated with stronger single-trial estimates of TG-PAC (Methods and Supplementary Table 2; Fig. 2f shows univariate correlation coefficients for illustration; all statistics and conclusions are based on the generalized linear model (GLM) results). By contrast, there were no significant correlations between PAC and RT in the amygdala and vmPFC (Fig. 2f and Supplementary Table 2).

Lastly, we tested whether the differences in hippocampal PAC between loads could be explained by changes in power, theta waveform shape, cross-frequency phase–phase coupling, preferred phase or PAC peak frequency but did not find evidence for any of those factors (Extended Data Fig. 3). These findings suggest that PAC is related to ongoing WM processes during the maintenance period in the hippocampus, but not in the amygdala or frontal lobe.

## Category cells support WM maintenance

We next investigated whether cells remained persistently active during WM maintenance and, if so, whether their activity was related to PAC. We first selected for category neurons, which are cells of which the firing rate (FR) was related to the visual category of the stimuli shown during encoding (Methods). We identified such cells at numbers higher than expected by chance within the hippocampus (89 neurons (24.72%)), amygdala (181 neurons (36.49%)) and vmPFC (37 neurons (17.96%)) but not within the pre-SMA and dACC (Extended Data Fig. 4a; an example hippocampal neuron is shown in Fig. 3a). During the maintenance period, the FRs of the identified category neurons remained elevated during the maintenance period compared with the baseline across all of the correct trials in the MTL, but not in the vmPFC (Fig. 3b shows the hippocampus and amygdala combined for simplicity; statistics per area are shown in Extended Data Fig. 4b). Furthermore, the FRs of category cells were significantly higher in trials in which a stimulus of the preferred category of a cell was held in WM relative to when patients held stimuli in mind that belonged to the non-preferred categories of a cell in the MTL, but not the vmPFC ($t_{269} = 2.93$, $P = 0.001$; Fig. 3b; see Extended Data Fig. 4b for vmPFC). Category neurons in the MTL but not

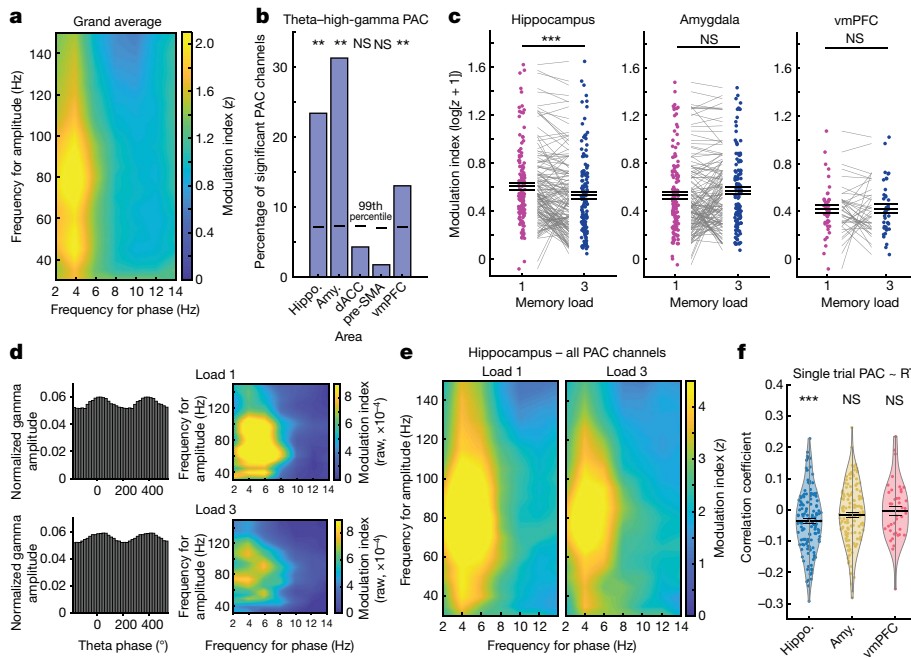

**Fig. 2 | TG-PAC. a**, Average normalized modulation indices for all phase–amplitude pairs. *n* = 1,917 channels. **b**, The proportion of channels with significant theta–high-gamma PAC in each area, determined by comparisons to trial-shuffled surrogates (both loads). The horizontal lines indicate the 99th percentile of the surrogate null distribution per area (*P* = 0.005 for hippocampus (hippo.), amygdala (amy.) and vmPFC; right-sided permutation test, no adjustment for multiple comparisons). **c**, log-normalized modulation indices were averaged within the theta–high-gamma band in each load and compared between the loads in each significant PAC channel in each region. Only in the hippocampus, theta–high-gamma PAC differed as a function of load, with PAC higher in load 1 versus load 3 trials (left: *n* = 137 channels, *P* = 0.0005; amygdala (middle): *n* = 130, *P* = 0.38; vmPFC (right): *n* = 40, *P* = 0.87; two-sided permutation-based *t*-tests; FDR corrected for all five brain areas). *z*-scored values were shifted into a positive range by an offset of 1 and log-transformed for illustrative

purposes only. All statistics are based on non-transformed *z* values. **d**, Example gamma amplitude distribution over theta phase as well as comodulograms with raw modulation indices in each load for a representative hippocampal channel. Note the wider distribution of gamma amplitude over theta phase in load 3 trials, which leads to lower levels of PAC (further analysis is provided in Extended Data Fig. 3a). Normalized MI values were as follows for the two examples shown: load 1, *z* = 16.52; load 3, *z* = 7.92. **e**, Average normalized modulation indices for significant hippocampal PAC channels. *n* = 137. **f**, TG-PAC was significantly negatively correlated with RTs in the hippocampus (*n* = 137, *P* = 6 × 10⁻⁵, mixed-effects GLM), but not in the amygdala (*n* = 130, *P* = 0.48) or the vmPFC (*n* = 40, *P* = 0.24). GLM results are shown in Supplementary Table 2. Each dot represents a significant PAC channel. For **c**,**f**, data are mean ± s.e.m. \**P* < 0.05; NS, not significant.

the frontal lobe therefore exhibited stimulus-specific persistent activity (statistics per area are shown in Extended Data Fig. 4b). For this reason, we focus on MTL category neurons for the remainder of the paper.

The activity of category neurons in the MTL was modulated by load, with FRs higher in load 1 than in load 3 in preferred trials ($t_{269}$ = 2.65, *P* = 0.004; Fig. 3c), but not in non-preferred trials ($t_{269}$ = −1.46, *P* = 0.14; Extended Data Fig. 5a) during the maintenance period. Moreover, FRs were higher in correct as compared to incorrect trials across both load conditions ($t_{245}$ = 2.43, *P* = 0.02; Fig. 3d; 24 neurons were excluded from this comparison due to insufficient data in the incorrect condition; patient-level statistics are shown in Extended Data Fig. 5b). There was no significant difference in the FRs between fast and slow RT trials (median split; computed per load condition and then averaged across loads for the preferred category and correct trials only; hippocampus, $t_{88}$ = 0.73, *P* = 0.47; amygdala, $t_{180}$ = 1.38, *P* = 0.18). Together, these data demonstrate the relevance of category cells to WM maintenance.

We next examined how spike timing of category neurons in the MTL relates to the phase of LFPs by examining their spike-field coherence (SFC; Methods) during the maintenance period in channels with significant PAC (Fig. 3e). In the hippocampus, high-gamma-band SFC was significantly stronger in preferred trials than in non-preferred trials for neuron-to-channel combinations that involved significant PAC channels (cluster *P* = 0.004; Fig. 3f). This difference in gamma-band SFC was present in both load 1 and load 3 (Fig. 3g; preferred versus non-preferred trials in load 1 ($t_{150}$ = 3.14, *P* = 0.003) and load 3 ($t_{150}$ = 2.88,

*P* = 0.004)). Computing the same statistic for theta-band SFC did not reveal any significant effects (Extended Data Fig. 5c), confirming specificity to gamma. Notably, this gamma cluster spanned approximately the same frequencies at which theta–high-gamma PAC was present (Extended Data Fig. 5f,g shows comparisons including all channels and patient-level results). We did not find similar effects for the theta band (Fig. 3f), non-PAC channels (Extended Data Fig. 5e) or the amygdala (Fig. 3f and Extended Data Fig. 5e,f). Finally, SFC was different between preferred and non-preferred trials only for spikes that occurred during high ($t_{150}$ = 3.06, *P* = 0.002) but not low high-gamma band power in the hippocampus ($t_{150}$ = 0.26, *P* = 0.85; Extended Data Fig. 5d). Thus, specifically in periods in which gamma amplitude was high, spikes of category neurons were more strongly synchronized to the phase of gamma-band LFP.

We next determined whether spiking activity of category cells during the WM maintenance period correlated with PAC trial-by-trial (Methods). In the hippocampus, PAC was weakly but significantly positively correlated with the FR of category neurons (Supplementary Table 3; Fig. 3h shows correlation coefficients for illustration only; all conclusions are based on the GLM results). In the amygdala, there were no significant correlations between single-trial TG-PAC and FR of category neurons (Fig. 3h and Supplementary Table 3). Together, these results show that persistently active hippocampal category-selective neurons were more synchronized with gamma LFPs when their preferred category was held in WM. This effect was specific to channels that

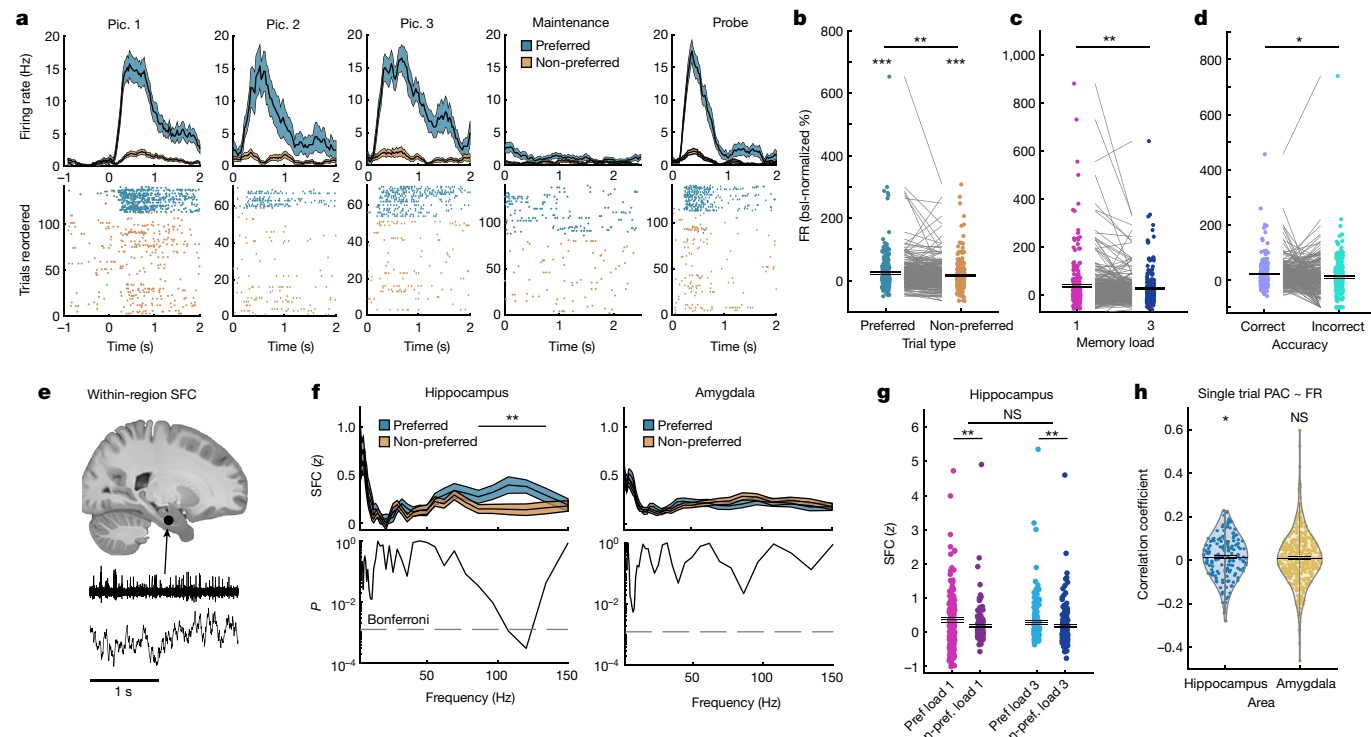

**Fig. 3 | FRs and SFC of category neurons in MTL. a**, Example hippocampal category neuron. The preferred category of this neuron was 'animals'. Pic., picture. **b**, Category neurons ($n = 270$) remained active (preferred (pref.) versus baseline (bsl), $P = 0.0001$; non-preferred (non-pref.), $P = 0.0001$) and retained their selectivity during the maintenance period, with FRs higher for preferred compared with non-preferred categories ($P = 0.001$). FRs are shown as the percentage change compared with the baseline ($-0.9$ to $-0.3$ s before picture 1 onset). **c**, FRs of category neurons were higher in load 1 compared with load 3 trials with their preferred category held in WM ($n = 270$, $P = 0.004$) during the maintenance period. **d**, Category neurons fired more in correct compared with incorrect trials ($n = 246$, $P = 0.02$). **e**, SFC between spikes and LFPs from the same area. The slice (https://osf.io/r2hvk/) was obtained under a Creative Commons licence CC BY 4.0. **f**, Paired with PAC channels, hippocampal category neurons were more strongly phase-locked to local gamma LFPs with the preferred category held in WM ($n = 151$ combinations; $P = 0.004$). Differences were not significant in the amygdala ($n = 423$) or non-PAC channels (Extended Data Fig. 5e). Statistical analysis was performed using two-sided cluster-based permutation $t$-tests. **g**, Gamma (70–140 Hz) SFC for hippocampal category neurons was stronger for preferred versus non-preferred trials in both load conditions (main effect preference, $F_{1,150} = 16.23$, $P = 0.0001$; load 1, $P = 0.003$; load 3, $P = 0.004$). No main effect of load ($P = 0.25$) or interaction ($P = 0.33$) was found. Each dot is a neuron–LFP channel pair ($n = 151$). **h**, TG-PAC was positively correlated with the FR of category neurons in the hippocampus ($n = 151$, $P = 0.017$, mixed-effects GLM), but not in the amygdala ($n = 423$, $P = 0.45$). The GLM results are shown in Supplementary Table 3. For **b**–**d**,**g**, statistical analysis was performed using two-sided permutation-based $t$-tests (**b**–**d**, lower brackets in **g**) and $F$-tests (top bracket in **g**). For **a**–**d**,**f**–**h**, data are mean ± s.e.m. (coloured areas in **a**,**f**); **$P < 0.01$.

showed significant TG-PAC during the maintenance period. Also, FRs of hippocampal category neurons were correlated with single-trial estimates of PAC.

## Category neurons are not PAC neurons

Although the above results indicate a relationship between category neurons and PAC within the hippocampus, they alone do not definitively demonstrate that the spiking activity of category neurons was sensitive to the nonlinear interaction between theta-phase and gamma amplitude as would be expected from PAC. Thus, we next selected for MTL neurons whose activity was modulated by PAC and determined whether they significantly overlapped with the persistently active category neurons. We defined PAC neurons as neurons whose FR was a function of the interaction between theta-phase and gamma amplitude (Methods; an example is shown in Fig. 4). In the hippocampus, 79 (37.29%) out of 212 available neurons ($P < 0.005$; 200 permutations (Methods); pre-processing steps removed broadband LFPs for some of the neurons and those were therefore not part of this analysis) qualified as PAC neurons. In the amygdala, 163 (45.53%) out of 358 neurons ($P < 0.005$) qualified as PAC neurons.

We next examined whether the selected PAC neurons were also category neurons. In the hippocampus, 28 (35.44%) out of the 79 PAC

neurons were both PAC neurons as well as category neurons. In the amygdala, this was the case for 68 (41.72%) out of the 163 PAC neurons (Fig. 4b). The proportion of category neurons among PAC neurons was not significantly higher than expected by independent subpopulations in any of the two regions (both $P > 0.05$; Methods). To further corroborate this finding, we trained a linear decoder to differentiate between the five different picture categories based on the FRs extracted during picture presentation (encoding). As expected, the decoder was able to differentiate between the picture categories when it was trained on FRs from the category neurons (hippocampus, 72.77%, $P = 0.001$; amygdala, 88.71%, $P = 0.001$; Extended Data Fig. 7a). However, the decoder could not differentiate between the categories when it was trained on FRs from PAC neurons that were not also category neurons in both MTL areas (hippocampus, 26.06%, $P = 0.16$; amygdala, 25.86%, $P = 0.15$; chance level = 20%). In summary, the probabilities of a neuron being a PAC or a category neuron were independent and the activity of PAC neurons did not differ between the category of the presented stimuli.

## Properties of PAC neurons

We next investigated whether PAC neuron activity was related to WM maintenance in ways other than persistent activity. We identified three such relationships for PAC neurons in the hippocampus.

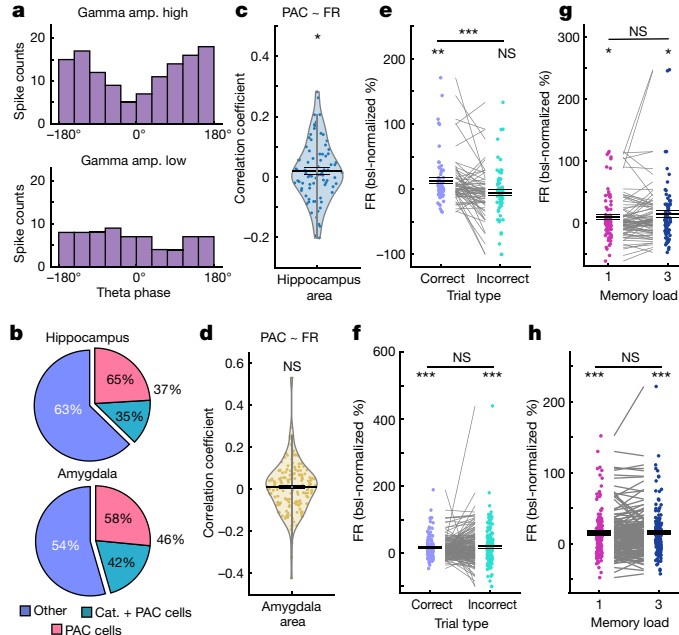

**Fig. 4 | PAC neuron selection and local activity. a**, The binning used for PAC neuron selection for an example hippocampal neuron. Theta phase, binned into ten groups, and gamma amplitude (amp.), median split into low and high, were used to predict the spike counts of each neuron from the MTL during the delay. The spike count was higher during high-gamma amplitudes (gamma main effect) and differed in their theta phase distribution between high and low gamma amplitudes (interaction effect), resulting in selecting this neuron as a PAC neuron. **b**, The proportions of neurons qualifying as PAC neurons. Cat., category. **c,d**, FRs of PAC neurons were positively correlated with single-trial estimates of TG-PAC in the hippocampus (**c**; $n = 79$, $P = 0.028$, mixed-effects GLM), but not in the amygdala (**d**; $n = 163$, $P = 0.98$; GLM results are shown in Supplementary Table 4). **e,f**, FRs of PAC neurons during the maintenance period differed between correct and incorrect trials in the hippocampus (**e**; $n = 63$; correct versus baseline, $P = 0.01$; incorrect, $P = 0.33$; correct − incorrect, $P = 0.0001$; 16 neurons were excluded due to insufficient data in the incorrect condition), but not in the amygdala (**f**; $n = 156$; correct, $P = 0.0001$; incorrect, $P = 0.0001$; correct − incorrect, $P = 0.45$; 7 were neurons excluded due to insufficient data in the incorrect condition). FRs are shown as the percentage change compared with the baseline (−0.9 to −0.3 s). **g,h**, FRs did not differ between loads (hippocampus (**g**): $n = 79$; load 1, $P = 0.03$; load 3, $P = 0.01$; load 3 − load 1, $P = 0.20$; amygdala (**h**): $n = 163$; load 1, $P = 0.0001$; load 3, $P = 0.0001$; load 3 − load 1, $P = 0.80$). For **g**–**j**, statistical analysis was performed using two-sided permutation-based $t$-tests. For **e**–**j**, data are mean ± s.e.m.

First, FRs of PAC neurons were positively correlated with estimates of single-trial PAC (Methods and Supplementary Table 4; Fig. 4c,d shows univariate correlation coefficients for illustration). Second, their FR was elevated throughout the maintenance period compared with the baseline ($t_{78} = 2.43$, $P = 0.01$). Third, FRs varied as a function of accuracy (correct versus incorrect trials ($t_{62} = 3.82$, $P < 0.001$; Fig. 4e; 16 neurons were excluded from this comparison due to insufficient data in the incorrect condition), and were elevated as compared to the baseline in correct trials ($t_{62} = 2.67$, $P = 0.01$) but not in incorrect trials ($t_{62} = −0.98$, $P = 0.33$). FRs were not significantly different between the two load conditions (load 3 − load 1, $t_{78} = 1.38$, $P = 0.20$), but FRs were elevated compared with the baseline in each condition considered separately (load 1, $t_{78} = 2.14$, $P = 0.03$; load 3, $t_{78} = 2.45$, $P = 0.01$; Fig. 4g; similar results concerning theta and gamma SFC are shown in Extended Data Fig. 7b). PAC neurons did not show a load-dependent shift in preferred theta phase (Extended Data Fig. 7d), or significant FR differences between fast and slow RT trials ($t_{78} = −0.07$, $P = 0.94$). By contrast, in the amygdala, FRs of PAC neurons were not significantly

correlated with single-trial estimates of TG-PAC (Fig. 4d and Supplementary Table 4). PAC neurons in the amygdala also showed higher FRs during the maintenance period compared with the baseline ($t_{162} = 6.40$, $P < 0.001$), but there were no significant differences between correct and incorrect WM trials ($t_{156} = −0.77$, $P = 0.45$; Fig. 4f), loads (load 3 − load 1, $t_{162} = 0.26$, $P = 0.80$; Fig. 4h; SFC and phase shift results in the amygdala are shown in Extended Data Fig. 7c,d) or fast and slow RT trials ($t_{162} = 1.87$, $P = 0.07$). Thus, hippocampal PAC neurons, despite not being tuned to WM content, were engaged in WM maintenance because their FRs were elevated during WM maintenance and differed as a function of behavioural accuracy.

## PAC neurons phase lock to the frontal cortex

Given the properties of PAC neurons shown above, we hypothesized that PAC neurons might be involved in cognitive control of WM. We therefore next examined whether the activity of PAC neurons is related to frontal activity[34]. To do so, we computed cross-regional SFC between spiking activity of PAC neurons in the MTL and the LFPs recorded in the pre-SMA, dACC and vmPFC (Fig. 5a). If PAC neuron activity is related to frontal cognitive control, we expected cross-regional SFC in the theta range to be stronger in load 3 than in load 1 trials. The data support this hypothesis—SFC was significantly stronger in load 3 compared with load 1 between spiking activity of PAC neurons in the hippocampus and theta-band LFPs recorded in the vmPFC (Fig. 5b and Methods; cluster $P < 0.001$). We did not observe significant differences for other frequency bands, nor for the other two frontal brain areas (see Extended Data Fig. 8a–h for narrow versus broad-spiking neurons, patient-level statistics, load comparisons for within-vmPFC and cross-regional SFC between vmPFC LFPs and hippocampal spiking, as well as comparisons of non-specific global changes). Repeating the same analysis for category neurons did not reveal significant differences in SFC strength (Fig. 5c and Extended Data Fig. 8c). Furthermore, the difference in SFC strength between the two load conditions for random subsets of hippocampal neurons (the same number of connections as for PAC neurons) was smaller for all tested 10,000 combinations compared with that for hippocampal PAC neurons ($P = 0.0001$; Fig. 5e). Lastly, PAC neurons from the amygdala did not show significant cross-regional SFC differences in any of the tested frequencies (Fig. 5d) or regions. We further hypothesized that, if theta-band cross-regional SFC indeed reflects levels of cognitive control, it should be stronger for faster RTs. This was the case—theta SFC was stronger for fast compared with slow RTs for PAC neurons from the hippocampus ($t_{166} = 2.10$, $P = 0.03$; Fig. 5f), but not from the amygdala ($t_{705} = 1.40$, $P = 0.16$). Thus, we conclude that the theta-band phase locking of PAC neurons in the hippocampus to the vmPFC is related to cognitive control.

## Information-enhancing noise correlations

Cells that by themselves carry no information in their FR can influence the representation of a variable at the population level if their activity is correlated with other cells[35,36]. We hypothesized that PAC neurons might have this role during WM maintenance. In both the hippocampus (162 pairs; $t_{161} = 5.26$; $P < 0.001$) and the amygdala (892 pairs; $t_{891} = 15.51$; $P < 0.001$), pairs of category neurons and PAC neurons had on average positive co-fluctuations of spike counts (Fig. 6a (left); see Extended Data Fig. 9a for amygdala). As a control, we shuffled trials within conditions to remove noise correlations while leaving all other properties of the signal intact, including common category-related input, interspike-interval distributions, FR distributions and temporal relations to task events (Methods). The correlation coefficients across all PAC and category neuron pairs were significantly greater than the same correlations computed after shuffling trials this way (see Fig. 6a (right) for hippocampal pairs; noise correlations computed across trials are shown in Extended Data Fig. 9b–d).

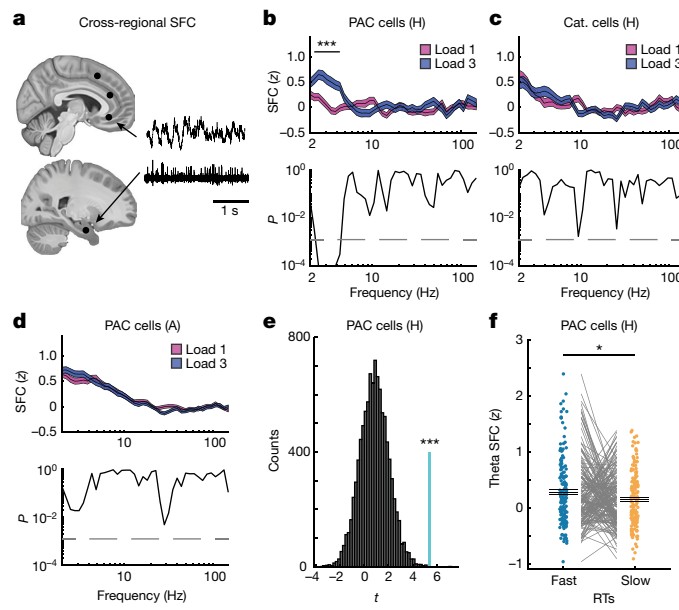

**Fig. 5 | Remote connectivity of PAC neurons in the MTL to frontal theta LFPs. a**, Long-range SFC between MTL spiking activity and LFPs recorded from all three frontal regions. The slices (https://osf.io/r2hvk/) were obtained under a Creative Commons licence CC BY 4.0. **b**, Spikes of hippocampal PAC neurons were more strongly synchronized with theta-band LFPs recorded in the vmPFC during the maintenance period during load 3 compared with load 1 trials (*n* = 175 connections; cluster *P* = 0.0001). This was not the case for the pre-SMA and dACC (Extended Data Fig. 8). **c,d**, Category neurons from the hippocampus (**c**; *n* = 215) or PAC neurons from the amygdala (**d**; *n* = 767) did not show significant SFC differences between loads relative to the vmPFC LFP. For **b**–**d**, statistical analysis was performed using two-sided cluster-based permutation *t*-tests with a Bonferroni-corrected alpha-level for two MTL areas, three frontal areas and two cell populations. **e**, Hippocampal PAC cells (*n* = 175, cyan line) yielded the strongest long-range theta SFC difference between load 3 and load 1 trials among 10,000 random selections of hippocampus–vmPFC connections (*P* = 0.0001, right-sided permutation test). *t* values correspond to comparisons between load 3 and load 1 trials for an average of SFC values in the significant theta range (2.5–4.3 Hz). For **b**–**d**, data are mean (centre line) ± s.e.m. (coloured areas). **f**, The remote theta-band SFC between spiking activity of PAC neurons and LFPs recorded in the vmPFC for fast compared with slow RT trials (*P* = 0.03, two-sided permutation-based *t*-test). Each dot is a neuron-channel connection (*n* = 167; 8 connections were excluded due to inefficient spike count in at least one of the conditions). Data are mean ± s.e.m. A, amygdala; H, hippocampus.

This was also true for correlations among pairs of category neurons and PAC neurons that were not also category neurons (Extended Data Fig. 9e,f).

We next examined whether PAC neurons contributed to the decodability of image category during WM maintenance at the population level. We iteratively added neurons to the population through greedy selection of the neuron that adds most decodability above and beyond that provided by the already included neurons (Methods; see Fig. 6b for an example from a single session for neurons up to peak decodability). In the hippocampus, adding single PAC neurons to the optimized decoding ensemble significantly enhanced category decoding when noise correlations were intact ($t_{20}$ = 3.16, *P* < 0.001) but not when they were removed ($t_{20}$ = −0.12, *P* = 0.91, $BF_{01}$ = 4.37; intact versus removed, $t_{20}$ = 3.33, *P* = 0.003; Fig. 6c). This suggests that specifically the noise correlations of PAC neurons enhanced the decodability of category information, not their category-related FRs per se.

We next compared the maximal decoding performance for intact and removed noise correlations before and after all PAC neurons were removed from the ensembles. PAC neurons enhanced the decodability

of category, but only when noise correlations were intact ($t_{21}$ = 2.94; *P* = 0.004; FDR corrected for all comparisons; Fig. 6d). Removing PAC neurons from the ensemble after noise correlations were removed did not significantly change the decoding performance ($t_{21}$ = 1.36, *P* = 0.20; Fig. 6d (red)). The decoding performance dropped to a similar extent when only removing PAC neurons that were not also category neurons (Extended Data Fig. 9g). Together, this pattern of results suggests that the decrease in decoding performance was not caused by removing PAC neurons that also carried category information. Rather, dropping neurons that were modulated by PAC from the ensemble led to a reduction in decoding accuracy. In accordance with these results, the maximal differences between intact and removed noise correlations were larger when PAC neurons were part of the ensembles as compared to being removed ($t_{21}$ = 3.59, *P* < 0.001; Fig. 6e).

To assess the specificity of those results, we repeated the above analysis after removing randomly chosen neurons that were not PAC neurons from the ensembles (Fig. 6d (without non-PAC); averaged over 500 random selections; the same number as for PAC neurons in each session). This revealed that removing non-PAC neurons in the ensembles without noise correlations led to further decreases of decoding performance ($t_{21}$ = 3.89; *P* < 0.001), thereby indicating that non-PAC neurons contributed information to the population that did not depend on noise correlations. Accordingly, we did not find a significant difference when comparing the maximal decoding difference between intact and removed noise correlations for ensembles for which we removed randomly selected non-PAC neurons ($t_{21}$ = −1.77, *P* = 0.10; Fig. 6e). In contrast to in the hippocampus, in the amygdala, PAC neurons contributed to category decodability not only when noise correlations were intact but also when noise correlations were removed; Extended Data Fig. 9j,k).

We quantified geometric features of the data manifold to determine why noise correlations enhanced decodability of WM content. We quantified the angle between the two vectors that describe the signal and the noise axis in the *n*-dimensional space formed by the *n* simultaneously recorded neurons in a given session. Whether noise correlations are information limiting or enhancing depends on this angle[37]. As noise correlations are information enhancing in our case, we hypothesized that the angle between the signal and the noise axes should (1) be relatively large to begin with; and (2) increase when noise correlations are present compared with when they are absent (Fig. 6f). This was the case—when noise correlations were intact, the signal–noise axis angle was around 69° (out of maximally 90°). After removing noise correlations, this angle became significantly smaller, as hypothesized ($t_{31}$ = 2.77, *P* = 0.009; Fig. 6g).

We examined the variance of the signal projected onto the signal axis[38] in populations with and without PAC neurons present to examine whether noise correlations are related to PAC. Removing PAC neurons from the ensembles significantly increased the s.d. of the projection values (main effect for ensemble, $F_{1,18}$ = 12.55, *P* = 0.0013; Fig. 6h). Moreover, the s.d. of the projected values was larger when noise correlations were removed (main effect for noise correlations, $F_{1,18}$ = 7.24, *P* = 0.014). This was only the case if PAC neurons were part of the ensembles ($t_{18}$ = −2.66, *P* = 0.014), and there was no significant difference between intact and removed noise correlations when PAC neurons were removed ($t_{18}$ = −1.69, *P* = 0.11). These findings suggest that specifically the noise correlations introduced by PAC neurons affected the geometry of the population code.

If noise correlations are beneficial to WM, they should be stronger in correct fast RT trials as compared to correct slow RT trials, specifically when category neurons' preferred categories were maintained in WM. In the hippocampus, noise correlations were significantly stronger for fast compared with slow RT trials ($t_{161}$ = 2.15, *P* = 0.028; Fig. 6i (left); patient-level results are shown in Extended Data Fig. 9h) for pairs of PAC–category cells in trials in which the preferred stimulus of the category cell in the pair was maintained in WM (Methods).

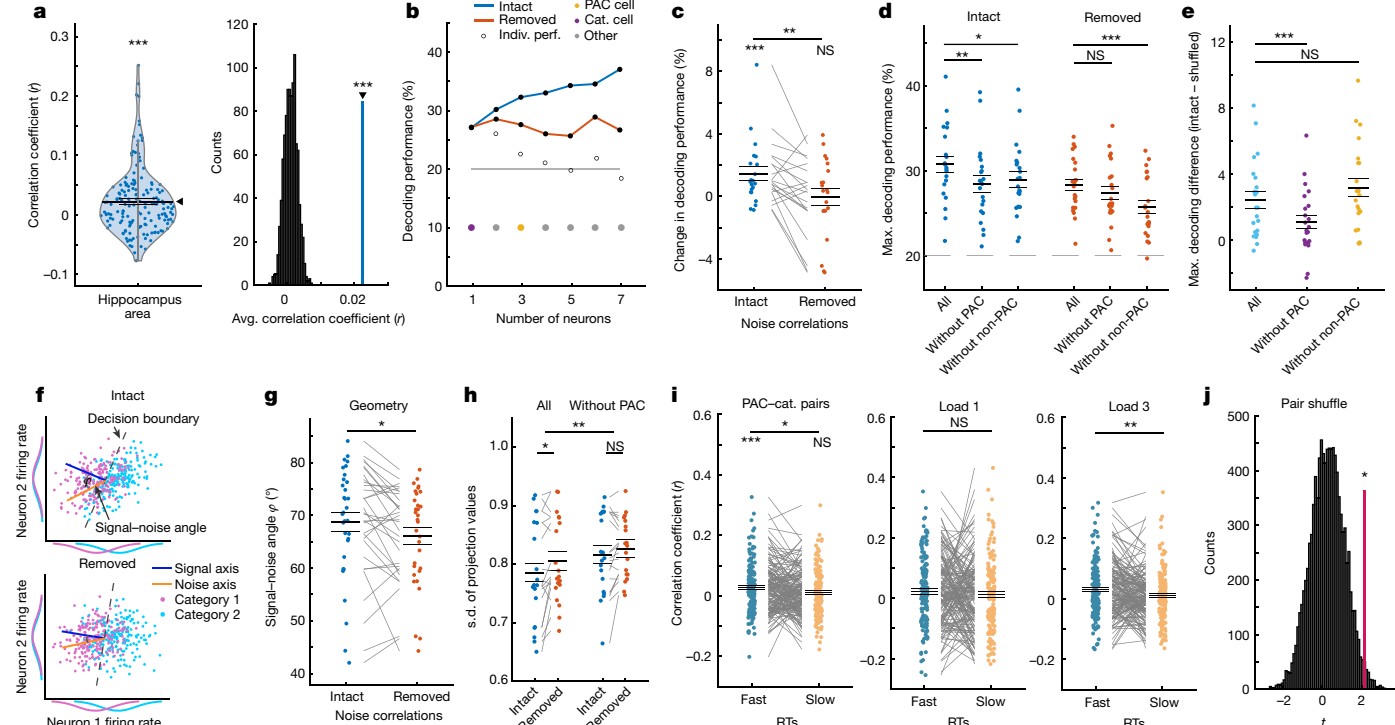

**Fig. 6 | Noise correlations of PAC neurons in the hippocampus. a**, The mean correlation for all category–PAC neuron pairs was positive ($n = 162$, $P = 0.0001$) (left). Right, the mean correlation was larger than the null distribution of mean correlations without noise correlations ($P = 0.001$, right-sided permutation test). **b**, The decoding accuracy for category from FR in the maintenance period with intact or removed noise correlations for an example session. The white dots signify accuracy, and the coloured dots indicate the identity for each neuron. Indiv., individual. **c**, Adding single PAC neurons to the ensemble with intact noise correlations increased the decoding accuracy ($n = 21$; intact, $P = 0.0007$; removed, $P = 0.91$; intact − removed, $P = 0.003$). **d**, Removing PAC neurons reduced decodability with intact noise correlations only ($n = 22$ sessions; intact, $P = 0.004$; removed, $P = 0.20$). Removing non-PAC neurons decreased decoding for intact ($P = 0.010$) and removed ($P = 0.0001$) correlations. **e**, The decoding difference with and without noise correlations was lower only when PAC neurons were removed ($n = 22$ sessions; PAC, $P = 0.0007$; non-PAC, $P = 0.10$). **f**, The

signal–noise axis for simulated tuned (neuron 1) and untuned (neuron 2) neurons. The signal–noise axis angle is reduced without noise correlations. **g**, The signal–noise angle in the data was reduced after removing noise correlations ($n = 32$ sessions, $P = 0.009$). **h**, The variance along the signal axis was reduced when noise correlations were intact only when PAC neurons were present ($n = 19$ sessions, $P = 0.014$; main effect ensemble: $P = 0.0013$; permutation-based $F$-test). **i**, Correlations of PAC–category pairs ($n = 162$) were stronger in fast ($P = 0.0001$) compared with slow ($P = 0.06$; median split; fast − slow, $P = 0.028$; preferred category trials only) RT trials. This effect was significant in load 3 ($P = 0.009$) but not in load 1 ($P = 0.34$) trials. **j**, The PAC–category pair correlation difference between fast and slow RT trials was larger than for random non-PAC–category pairs ($P = 0.016$, right-sided permutation test). For **a**,**c**–**e**,**g**–**i**, statistical analysis was performed using two-sided permutation-based $t$-tests. Data are mean ± s.e.m.

Noise correlations were on average significantly positive only in fast trials ($t_{161} = 4.10$, $P < 0.001$) but not in slow trials ($t_{161} = 1.94$, $P = 0.06$). For non-preferred trials, we did not observe a significant difference between fast and slow RT trials (Extended Data Fig. 9h). Separating trials into the two load conditions, we observed a significant difference only between fast and slow trials in load 3 ($t_{161} = 2.60$, $P = 0.009$; Fig. 6i (right)), but not in load 1 ($t_{161} = 0.96$, $P = 0.34$; Fig. 6i (middle)). In the amygdala, comparing fast to slow RT trials in preferred trials did not reveal a significant difference (Extended Data Fig. 9l). Lastly, we examined whether the effect of noise correlations on RTs in the hippocampus was specific to PAC-to-category neuron pairs or a common feature across the entire population of simultaneously recorded neurons. PAC-to-category neuron pairs showed a significantly stronger effect compared with most randomly selected cell pairs ($P = 0.016$; Fig. 6j), showing that noise correlations between category and PAC neurons within the hippocampus contributed to enhanced WM fidelity.

## Discussion

Although TG-PAC is ubiquitous at the electroencephalogram and LFP level, it has remained unclear how PAC is reflected at the single-neuron

level. We find that TG-PAC has a direct relationship to the spiking activity of individual neurons. However, while by definition PAC neurons were related to local gamma activity, they were not directly involved in the processing of the memoranda held in WM per se. Rather, their activity was coordinated with frontal theta, with stronger phase locking for higher WM load and faster RTs. This finding indicates that PAC neurons have a role in cognitive control.

Long-range theta phase locking between frontal and temporal/occipital areas has been suggested to reflect frontal cognitive control exerted over task-relevant brain processes[29,30,39]. In WM maintenance, frontotemporal interactions are crucial, especially when involving the hippocampus[31,32,40–43]. Theta-based prefrontal coordination of posterior WM content-specific processes could ensure efficient information processing at phases that are optimal for network-wide communication within the memoranda-processing population of neurons[30,44]. According to this model, higher levels of cognitive control are indicated by stronger phase locking between regions to facilitate faster and more efficient readout of WM content. Our results support this model and provide evidence for a specific mechanism to implement it. SFC between hippocampal PAC neurons and vmPFC theta was stronger in periods in which more cognitive control was required. Cross-regional hippocampus–vmPFC SFC was enhanced for fast compared with slow

RT trials, suggesting that a more efficient interaction between distant areas might lead to a behaviourally beneficial read-out of WM content with stronger levels of control.

The goal of successful cognitive control is to increase the fidelity of the retained memories. Here we show that one way PAC neurons contributed to achieving this goal is by introducing noise correlations that enhance information content at the population level. Noise correlations are a population phenomenon and are therefore not expected to occur only between specific subpopulations of neurons. However, our results show that the noise correlations of PAC neurons with other neurons predicted behaviour better than pairs not involving PAC neurons. Although the numerical strength of the pairwise noise correlations between single pairs of neurons was low (as expected[45]), noise correlations of this magnitude can have large effects at the population level[46,47]. Noise correlations are typically thought to be information limiting[48,49], but theoretical work shows that this is not always the case[35,50–52]. Rather, in our case, noise correlations of PAC neurons shaped the geometry of stimulus category information such that decodability of WM content improved. That is, noise correlations were information enhancing. These decodability enhancements were abolished when noise correlations among neurons were removed. By contrast, although non-PAC neurons also improved decoding accuracy, these improvements were not due to noise correlations because decoding improvement did not change when noise correlations were removed. Geometrically, the effect of PAC neurons was to increase the angle between the signal and the noise axis, which improved the read-out of encoded WM content and enhanced the fidelity of WM memoranda representations.

Taken together, our results are in agreement with a multicomponent view of WM[1,2], whereby frontal control processes regulate and manage maintenance of WM content in storage-related areas such as the hippocampus[3–5,14]. This interplay between the control and processing of WM content was revealed by jointly analysing the activity of both PAC and category neurons. PAC neurons are a single-cell correlate of the widespread macro-scale phenomena of TG-PAC. They mediate interareal interactions that have a role in cognitive control and shape WM fidelity through noise correlations with information-carrying, persistently active category neurons, with stronger interactions in trials with successful control. PAC-mediated interareal interactions might serve as a general mechanism for top-down control to influence bottom-up processes, a hypothesis that we confirm here for WM, but that remains to be tested for other high-level cognitive functions involving top-down control from frontal regions such as attention[53], decision making[54,55], speech comprehension[56] and long-term memory retrieval[34].

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

## Methods

### Patients

A total of 36 patients (44 sessions; 21 female individuals; 15 male individuals; age: 40.47 ± 13.76 years; Supplementary Table 5; no statistical methods were used to predetermine sample sizes) participated in the study. All of the patients had Behnke–Fried hybrid electrodes (AdTech) implanted for intracranial seizure monitoring and evaluation for surgical treatment of drug-resistant epilepsy. Their participation was voluntary, and all of the patients gave their informed consent. This study was part of an NIH Brain consortium between three institutions (Cedars-Sinai Medical Center, Toronto Western Hospital and Johns Hopkins Hospital) and was approved by the Institutional Review Board of the institution at which the patient was enrolled. A pre-operative magnetic resonance imaging (MRI) image together with MRI or computed tomography post-operative images were used to localize the electrodes using Freesurfer as previously described[34]. Electrode positions are plotted on the CITI168 Atlas Brain[57] in MNI152 coordinates for the sole purpose of visualization (Fig. 1b). The 3D plot was generated using FieldTrip (v.20200409) and the Brainnetome Atlas[58]. Coordinates appearing in white matter or outside of the target area is due to usage of a template brain. Electrodes that were localized outside of the target area in native space were excluded from analysis (8 out of a total of 265 recording sites). Data used in this study are available in the DANDI archive[59].

### Task

The task consists of 140 trials and 280 novel pictures. Each trial started with a fixation cross presented for 0.9 to 1.2 s (Fig. 1a). Depending on the load condition, the fixation cross was followed by either one (load 1; 70 trials) or three (load 3; 70 trials) consecutively presented pictures, each remaining on the screen for 2 s. In load 3 trials, pictures were separated by a blank screen randomly shown for 17 to 200 ms. Picture presentation was followed by a 2.55 to 2.85 s long maintenance period in which only the word "HOLD" was presented on the screen. The maintenance period was terminated by the presentation of a probe picture, which was either one of the pictures shown earlier in the trial (match) or a picture already presented in one of the previous trials (non-match; see below). The task was to indicate whether the probe picture matched on of the pictures shown earlier in the same trial or not. The probe picture was shown until the patients provided their response through a button press. The response mapping switched after half the trials, which was communicated to patients during a short half-time break. Responses were provided using a Cedrus response pad (RB-844; Cedrus). All of the pictures were novel (that is, the patient had never seen this particular image) and were drawn from five different visual categories: faces, animals, cars (or tools depending on the version), fruits and landscapes. Images (width × height: 10.5 × 7 visual degrees) were presented in the centre of the screen and never more than twice (that is, when serving as the probe picture). Pictures were only repeated when presented as the probe stimulus. To make sure that also the non-match probe pictures were never completely new to patients (as were the matching probe pictures), which could have been used as a strategy to solve the task without using WM, we always used a picture that the patients had seen already in one of the earlier trials, randomly drawn from one of the categories not used during encoding. If a patient participated in more than one session, we used a completely new set of pictures in each session to ensure that all pictures were novel in all of the sessions. The overall longer duration of load 3 as compared to load 1 trials ensured increased cognitive control demands in trials with higher load. The maintenance period was the same length regardless of load, and all analysis of neural activity during the maintenance period was performed within this time window (0–2.5 s after the maintenance period onset). Note that, in load 3 trials, the three encoded items were from three different categories, assuring that the participants always had to maintain pictures from three different categories. Thus, when comparing trials between load 1 and 3 for preferred trials, each load condition always contained exactly one item from the preferred category.

### Spike sorting

Each hybrid depth electrode contained eight microwires from which we recorded the broadband LFP signal between 0.1 and 8,000 Hz at a sampling rate of 32 kHz (ATLAS system, Neuralynx; Cedars-Sinai Medical Center and Toronto Western Hospital) or 30 kHz (Blackrock Neurotech; Johns Hopkins Hospital) depending on the institution. All recordings were locally referenced within each recording site by using either one of the eight available micro channels or a dedicated reference channel with lower impedance provided in the bundle, especially when all channels contained recordings of neuronal spiking. To detect and sort spikes from putative single neurons in each wire, we used the semi-automated template-matching algorithm OSort (v.4.1)[60]. Spikes were detected after band-pass filtering the raw signal in the 300–3,000 Hz band (single-cell quality metrics are shown in Extended Data Fig. 1). All analysis in this paper (including the LFP) is based on signals recorded from micro wires. We isolated 360 neurons in the hippocampus, 496 in the amygdala, 204 in the pre-SMA, 188 in the dACC and 206 in the vmPFC. Of the LFP channels, 586 channels were in the hippocampus, 421 in the amygdala, 283 in the pre-SMA, 325 in the dACC and 307 in the vmPFC.

### LFP preprocessing

Before analysing the LFPs, we removed spike waveforms (action potentials) and excluded trials with interictal discharges and high-amplitude noise. First, to avoid leakage of spiking activity into lower frequency ranges[61,62], we removed the waveforms of detected spikes from the raw signal by linear interpolating the raw data from −1 to 2 ms around each spike onset in the raw recording before downsampling. As the same spike can, in rare instances, be recorded on more than one wire, we not only interpolated the data for the wire on which the neuron was detected but also for all other wires in the same wire bundle. We then low-pass filtered the raw signal using a zero phase-lag filter at 175 Hz and downsampled to 400 Hz. Line noise was then removed between 59.5 and 60.5 Hz as well as between 119.5 and 120.5 Hz using zero phase-lag band-stop filters. Extended Data Fig. 1f,g shows the raw LFP as well as the log–log power spectrum for an example channel from the hippocampus. The slope of log–log power spectra did not differ significantly between load 1 and load 3 trials in hippocampal channels ($n$ = 586 channels; mean slope −1.7526 ± 0.3902 versus −1.7517 ± 0.3928, $t_{585}$ = −0.86, $P$ = 0.39).

Artefact and inter-ictal discharge detection was performed on a per trial and wire basis using a semiautomated algorithm together with subsequent visual inspection of the data. To detect high-amplitude noise events as well as inter-ictal discharges, we $z$-scored the amplitude in each channel across all trials. To avoid artefactual amplitude biasing, we first capped the data at 6 s.d. from the mean and re-performed the $z$-scoring on the capped data[63,64]. If a single time sample in each trial and wire exceeded a threshold of 4 s.d., the trial was removed from the analysis for that wire. Jumps in the signal were detected by $z$-scoring the difference between every fourth sample of the capped signal. Trials in which any jump exceeded a $z$-score of 10 s.d. were removed. The result of this cleaning process was visually inspected in every recording and any remaining artefactual trials were removed manually. If a wire or brain region showed excessive noise or epileptic activity, it was entirely removed from the analysis. On average, 20.4 ± 13.9 trials (14.6% of the data) were removed per wire.

### PAC

We measured the strength of PAC for a wide set of frequency combinations in all of the recorded micro channels (except those excluded, see above) using the modulation index (MI) as introduced previously[65].

As the cleaning process described above produced a different set of available trials for each channel, we first randomly subsampled from all correct trials in each channel such that the number of trials were the same for both load 1 and load 3. We then extracted the LFP starting at −500 until 3,000 ms following the maintenance period onset in each selected trial. We then filtered (using pop_eegfiltnew.m from EEGLAB, v.2019.1)[66] each trial separately within the respective frequency bands of interest (see below for more details). We then extracted the instantaneous phase from the lower-frequency signal and the amplitude from the higher-frequency signal using the Hilbert transform. Lastly, we cut each trial to the final time window of interest of 0–2,500 ms relative to maintenance period onset. This last step ensures that filter artefacts that arise at the edges of the signal are removed. All analysis of neural activity during the maintenance period was performed in this 2.5-s-long time window that started at the onset of the maintenance period. The length of the analysis window was the same in both load conditions. Next, we concatenated the phase and the amplitude signal across trials and computed the MI as described previously[65] (18 phase bins). We computed MIs separately for load 1 and load 3 trials. All subsampled trials from both load conditions were used to select for significant PAC channels in an unbiased fashion (see below). Example code to reproduce parts of the results in this study in published at Zenodo[67].

To standardize the MI in each channel, frequency and condition, we computed 200 surrogate MIs by randomly combining the phase and amplitude signals from different trials (trial-shuffling), again separately for load 1, load 3 and for all trials. We fit a normal distribution to these surrogate data (normfit.m) to obtain the mean and s.d. of each distribution. These values were then used to z-transform the raw MI values. Standardizing MI values eliminates potential systematic differences that might arise due to load-related power or phase differences, which could drive observed differences in PAC. Moreover, low frequencies are more vulnerable to non-specific correlations to high-frequency power due to non-stationarities in the LFP signal caused by factors such phase resets. Comparing raw modulation indices to trial-shuffled surrogates within the same condition will reduce PAC that is caused by such non-specific interactions (discussed in detail previously[68]). In addition to providing a measure of significance, normalizing the MI values therefore allows for comparisons across conditions, frequencies and channels[69]. A channel was indicated as having significant PAC present if the normalized MI computed across all subsampled trials (both loads) exceeded a z-score of 1.64 ($P < 0.05$, right-sided).

We repeated the above procedure for all frequency combinations. The phase signals were extracted for centre frequencies between 2 and 14 Hz in steps of 2 Hz (2 Hz fixed bandwidth). The amplitude signals were extracted for frequencies between 30 and 150 Hz in steps of 5 Hz. The bandwidth of the amplitude signals was variable and depended on the centre frequency of the low-frequency signal. It was chosen such that it constituted twice the centre frequency of the phase signal (for example, if combined with an 8 Hz centre frequency for the phase signal, the bandwidth of the amplitude signal was chosen to be 16 Hz). This procedure ensures that the side peaks that arise if the amplitude signal is modulated by a lower-frequency phase signal are included[68].

To determine the influence of theta waveform shape on PAC, we tested for differences in theta waveform peak-to-trough as well as rise-to-decay asymmetries between the two load conditions, which could potentially cause differences in TG-PAC[70,71]. To extract and characterize each theta cycle during the delay period in all significant hippocampal PAC channels, we used the bycycle toolbox[72] in Python. We averaged estimates for peak-to-trough as well as rise-to-decay asymmetries across cycles during the maintenance period from the same trials used for our PAC analysis within each load and tested the estimates between the conditions. Results of this analysis are presented in Extended Data Fig. 3c.

Moreover, we determined the number of significant PAC channels that showed theta–high-gamma nesting using the method described previously[69]. To do so, in each PAC channel we determined the theta phase bin for which gamma amplitude was maximal, that is, the preferred theta phase of gamma amplitude. In the band-pass-filtered and phase-binned theta (3–7 Hz) signal, we then determined all instances during the delay periods of all of the correct trials in which this phase bin occurred, and extracted the precise timepoint at which the concurrent instantaneous gamma amplitude (70–140 Hz) was maximal within each bin. To obtain the average waveform, we selected a window of 500 ms centred on each timepoint in the raw (unfiltered) LFP recording and averaged the signals across all windows in each channel. Example average waveforms from two channels are shown in Extended Data Fig. 3e. In accordance with ref. 69, we characterized a waveform as being nested if at least three local maxima fell within a window of 45 ms (that is, 3 cycles at 70 Hz) around the preferred phase. Results are presented in Extended Data Fig. 3e.

### Relationship between single-trial PAC and FR or RT
We calculated single-trial estimates of TG-PAC for all significant PAC channels of both MTL regions and the vmPFC. We used mixed-effect GLMs to assess whether RT is related to PAC in a trial-by-trial manner (using only correct trials). We included load as a confounder and modelled random intercepts for each significant PAC channel nested into patientID. To examine whether there was a correlation between FR of category neurons (see below) and single-trial estimates of PAC, we used a mixed-effects GLM with load as a confounder and modelled random intercepts for each neuron to significant PAC channel combination. Only correct trials were used.

### Category cell selection
We selected for neurons of which the response after stimulus onset during encoding differed systematically between the picture categories of the stimuli shown. To do so, for each trial, we counted the number of spikes a neuron fired in a window between 200 to 1,000 ms after stimulus onset (all encoding periods and the probe period). We then grouped spike counts based on the category of the picture shown in that trial. For each neuron, we computed a 1 × 5 permutation-based analysis of variance (ANOVA) with visual category as the grouping variable, followed by a post hoc one-sided permutation-based t-test between the category with maximum spike count and all other categories. We classified a given neuron as a category neuron if both tests were significant ($P < 0.05$, 2,000 permutations (see below)). We refer to the category with the maximum FR as the preferred category of a cell. To test whether the observed number of category cells was significantly larger than that expected by chance in each area, we repeated the above selection for 500 times after shuffling the category labels for each stimulus across all picture presentations. If the observed number of category cells in the unshuffled data was higher than the 99th percentile ($P < 0.01$) of the resulting shuffled distribution (which, across all five brain areas, corresponds to a Bonferroni-corrected alpha level of 0.05), we considered the number of category cells observed in a given area as significant. Note that category cells are selected only using spiking activity from the encoding period, leaving the FRs during the maintenance period independent for later analyses.

### SFC
To measure how strongly the spiking activity of a neuron followed the phase of an LFP in a certain frequency, we computed the SFC. We measured SFC as the mean vector length (MVL) across spike-phases for all neuron-to-channel combinations available within a bundle or across regions (within the same hemisphere) in correct trials[73]. To estimate the instantaneous phase from LFPs in different frequency ranges in each trial, we applied continuous wavelet transforms using 40 complex Morlet wavelets[74] spanning from 2 to 150 Hz in logarithmic steps. The number of cycles for each wavelet changed as a function of frequency from 3 to 10 cycles, also in 40 logarithmic steps[75]. This ensured a higher

temporal precision for longer wavelets at low frequencies and higher frequency precision for faster wavelets at high frequencies as compared to using a constant number of cycles across all frequencies[76]. Extended Data Figure 1 shows the temporal and spectral characteristics across all wavelets used in this analysis. To assess the quality of our wavelet transform, we tested how well we were able to reconstruct the original signal after applying the wavelets to our data. To reconstruct the signal, we extracted the real-valued (bandpass-filtered) signal after applying each wavelet to the data and then summed up these signals across all frequencies. This resulted in a signal that closely followed the original recording in each trial (an example trial is shown in Extended Data Fig. 1j). We assessed how well the reconstructed signal predicted the original signal by computing $R^2$ values extracted from linear models using the reconstructed signal as predictors and the original signal as response variables in each trial and channel. As the quality of the reconstruction could change as a function of frequency or time, we performed this analysis for several time and frequency bins. First, we band-pass filtered both signals within the spectral bandwidth of each wavelet and then applied the linear model in sliding windows of 500 ms with a step size of 25 ms. The results of this analysis are presented in Extended Data Fig. 1.

For our SFC analysis, we first extracted data between −500 and 3,000 ms around the maintenance period onset from all clean trials in each channel and then computed a complex Morlet wavelet convolution to extract the instantaneous phase of the LFP as described above. The trials were then cut to the final time window of interest of 0 to 2,500 ms after the maintenance period onset to remove filter artefacts at the edges of each trial. To further avoid a bias of the MVL based on differences in spike count, we subsampled spikes such that an equal number of spikes was available in each condition. We included neurons that had at least 50 spikes available in each condition (we used a minimum of ten spikes for the preferred versus non-preferred analysis in category neurons due to a potentially low spike count in the non-preferred condition[77]). Next, we extracted the phase in the LFP closest to the timestamp of each spike, averaged across all spike-phases in polar space, and computed the MVL for each of the 40 frequencies. We repeated this subsampling 500 times and averaged the resulting MVLs across all repetitions within conditions. To avoid potential bias of load within the preferred versus non-preferred (category neurons; Fig. 3) or fast versus slow RT SFC comparison (cross-regional analysis; Fig. 5), we computed the SFC estimates within each load condition and then averaged across the loads.

The resulting MVL in each neuron-to-channel combination was further normalized using a surrogate distribution, which was computed after adding random noise to the timestamps of all spikes within a condition 500 times. Potential biases of the MVL based on systematic differences between the conditions (such as power differences between conditions within a given frequency band) were thereby reduced. Like for the measure of PAC (see above), we fit a normal distribution to the surrogate data and used the mean and the s.d. of that distribution to z-score the raw MVL within each condition.

To compare SFC between preferred and non-preferred trials, we computed SFC for all category neuron-to-channel combinations within the same region in frequencies between 2 and 150 Hz during the maintenance period and compared trials in which preferred or non-preferred stimuli were correctly maintained. We used cluster-based permutation statistics to identify ranges of frequencies with significant differences (Fig. 3f). To determine whether the observed gamma SFC difference between preferred and non-preferred trials was dependent on gamma amplitude, we tested whether gamma SFC (averaged across 70–140 Hz) for category neurons in the hippocampus differed between preferred and non-preferred trials for high and low gamma amplitudes separately (median split).

Whether theta or gamma-band SFC was related to the preference of the cell and/or load was tested by averaging SFC within the theta (3–7 Hz)

or gamma band (70–140 Hz) and computing a 2 × 2 permutation-based ANOVA with the factors load and preference for all category neuron to PAC channel combinations in the hippocampus.

To examine whether cross-regional SFC differed between the two load conditions, we computed SFC for all neuron-to-channel combinations between the respective areas in each load condition. We then used cluster-based permutation statistics to identify ranges of frequencies with significant differences (Fig. 5b; alpha level Bonferroni-corrected for all tests across two MTL areas, three frontal areas and two cell populations). To further determine a relationship to RT (Fig. 5f), we performed a median split of RTs for all correct trials within each load condition and compared cross-regional SFC between hippocampal PAC neurons and the vmPFC, averaged within and the significant theta range, between fast and slow RTs (averaged across both load conditions).

## Selection of PAC neurons

We selected for neurons whose FR was correlated with both theta phase and gamma amplitude during the maintenance period of the task. For all neuron-to-channel combinations within a bundle of microwires, we extracted the data from correct trials between −500 and 3,000 ms relative to the maintenance period onset and estimated the phase of theta signals by filtering between 3 and 7 Hz and computing a Hilbert transform in each trial. Gamma amplitude was determined by computing wavelet convolutions for frequencies between 70 and 140 Hz in frequency steps of 5 Hz (each wavelet using 7 cycles). Trials were cut to 0 to 2,500 ms after maintenance period onset to remove edge artefacts, and were then concatenated. The extracted amplitudes in each gamma frequency were z-scored across all trials and averaged across all frequencies. Computing wavelet convolutions in 5 Hz steps and z-scoring the data before averaging avoided biasing power estimates to lower frequencies due to the power law. Next, for each neuron–channel pair, we performed a median split of gamma amplitudes and binned all amplitudes into low and high gamma, respectively. In each of the two gamma groups, we further binned the corresponding theta phases into 10 groups (36° bins), resulting in a total of 20 bins (Fig. 4a). In each of those bins, we then counted the number of spikes that occurred in each theta–gamma bin.

We fit three Poisson GLMs for each neuron-to-channel combination. In model 1, spike count (SC) was a function of theta phase (10 levels), gamma amplitude (2 levels), and the interaction between theta phase and gamma amplitude. We included theta separately as cosine and sine due to the circularity of phase values[78], which enabled us to treat theta phase as a linear variable. Model 2 included the theta phase and gamma amplitude as main effects but not the interaction term. Model 3 included a main effect for theta phase and an interaction term but no main effect for gamma amplitude:

$$\text{Model 1: SC} \sim 1 + \text{Theta}_{cos} + \text{Theta}_{sin} + \text{Gamma} + (\text{Theta}_{cos} + \text{Theta}_{sin}) \times \text{Gamma}$$

$$\text{Model 2: SC} \sim 1 + \text{Theta}_{cos} + \text{Theta}_{sin} + \text{Gamma}$$

$$\text{Model 3: SC} \sim 1 + \text{Theta}_{cos} + \text{Theta}_{sin} + (\text{Theta}_{cos} + \text{Theta}_{sin}) \times \text{Gamma}$$

We next compared pairs of models using a likelihood-ratio test between model 1 and the two other models (using compare.m). A neuron qualified as a PAC neuron if model 1 explained variance in spike counts significantly better than both of the other two models ($P < 0.01$, FDR corrected for all possible channel combinations). The rationale behind each model comparison was as follows. First, we were specifically interested in neurons that followed the interaction of theta phase and gamma amplitude, that is, PAC, and not just theta phase or gamma amplitude alone. Selecting neurons for which model 1,

including the interaction term, explained spike count variance of a given neuron significantly better than model 2, lacking the interaction term, ensured extracting those neurons. Second, we also compared model 1 to model 3, lacking the gamma term, for the following reason. Assume that a given neuron–channel combination has an LFP with strong PAC at the field potential level, that is, strong interactions between theta phase and gamma amplitude, and a neuron of which the FR is not related to neither theta phase nor gamma amplitude. Nevertheless, this situation would result in a significant interaction term in model 1 because the spikes that fall into the low and high gamma amplitude groups will have different theta phases (due to PAC). This is only the case if the underlying PAC in the LFP is very strong (an illustration and further discussion is provided in Extended Data Fig. 6). However, in this scenario, the gamma amplitude term (or the theta phase term) would not be significant. Comparing model 1 to model 2 and model 3 therefore ensures that cells were selected only at PAC neurons in which the interaction term explained variance above and beyond the main effects and interactions alone.

As we did not observe strong PAC nor persistently active category neurons in frontal regions, we restricted this analysis to channels from the MTL regions and performed it separately in each load condition. If spike count variance was significantly better explained by model 1 than the two other models in either of the load conditions for at least one neuron-to-channel combination, we included this neuron as a PAC neuron. If a neuron was selected in more than one neuron-to-channel combination, we selected the combination with the highest $R^2$ in the full model (model 1). This combined channel was later used for within-region SFC as well as FR correlation analyses. Lastly, to determine whether the number of selected PAC neurons per area was significantly higher than chance, we repeated the entire selection process 200 times after pairing spikes and LFPs from different, randomly selected trials, therefore destroying their relationship with theta phase and gamma amplitude. The $P$ values indicate the proportion of repetitions that resulted in a higher number of selected neurons using the shuffled data than the original number of PAC neurons determined using the unshuffled data.

## Properties of PAC neurons

We used mixed-effects GLMs with load as a confounder and modelling a random intercept for each PAC neuron-to-channel combination nested into patientID (using only correct trials and the LFP channel selected for each neuron; see above) to examine the relationship between FR of PAC neurons and single-trial estimates of PAC. Note that trial-by-trial correlations are independent from the selection procedure as PAC neurons were selected on the basis of trial-averaged theta–gamma interactions, irrespective of their trial-by-trial variance.

## Noise correlations and population category decoding

We investigated the effect of noise correlations among groups of simultaneously recorded neurons on population decoding accuracies for the image category currently held in mind and on WM behaviour during the maintenance period. To estimate noise correlations among pairs of category and PAC neurons, for each neuron, we counted spikes in bins of 200 ms that slid across the maintenance period (0–2.5 s after the last picture offset) in steps of 25 ms. We then computed the correlation coefficient across all 101 time bins in each single trial for each pair of neurons and averaged across all considered trials within each condition. We used only correct trials for this analysis, and paired only neurons that were recorded in the same session and within the same brain region. Pairs of neurons recorded on the same channel were not considered as a precaution against spurious correlations caused by spike sorting inaccuracies. To assess the significance of noise correlations among pairs of neurons, we shuffled trial labels within conditions, that is, within the preferred and non-preferred category as

well as within each load, 1,000 times in each pair and recomputed the average correlation coefficient across all pairs. The original average correlation coefficient was then compared against the distribution of all average correlation coefficients obtained from the 1,000 trial shuffles (Fig. 6a (right)).

To investigate the contribution of PAC neurons to the population category decoding accuracy when noise correlations among neurons were intact or removed, we used an approach introduced previously[36] (Fig. 6b). To measure how much a single neuron affects the decoding accuracy of an ensemble of neurons, this approach finds optimized neuron ensembles that have maximal decoding accuracy by adding each single neuron to the ensemble in a stepwise manner. Each neuron's contribution to the ensemble can thereby be determined. In more detail, using a linear decoder, first the decoding performance for each single neuron in each region is determined from all correct trials. The neuron with the best decoding performance is then paired with each remaining neuron to determine which pair yields the best decoding accuracy. This most informative pair of neurons is then again combined with each remaining neuron to determine the most informative triplet of neurons, and so on. These steps were repeated until all neurons were part of the decoding ensemble.

As we were most interested in decoding picture category from FRs in the maintenance period, we used trials from load 1 only. This is because the maintenance period in load 3 trials contains intermixed information about the three different categories maintained in WM. We trained a linear support vector machine (SVM) decoder (fitcecoc.m; one-versus-one) on 80% of trials and tested it on the remaining 20% using z-scored FRs. To ensure an equal amount of data for all five categories, we subsampled trials to match the lowest number of trials available in each stimulus category. Noise correlations among neurons were left intact by using the same trials for each neuron or removed by shuffling trials per neuron within each category. Shuffling trials within each category ensured that the original category label remained correct but correlations among neurons were removed. Any decoding benefit that is purely based on category-selective firing activity is therefore not affected (Fig. 6b (red)). Note that, if PAC neurons enhanced the decodability by 'residual coding' of category information, they should have done so also when noise correlations were removed through shuffling. We repeated each decoding analysis 500 times and averaged the results to generalize across trial selections.

To test the influence of PAC neurons as well as their noise correlations on decoding performances, we first tested contributions between intact and removed noise correlations for PAC neurons that were added to the ensemble before maximal decoding performance was reached in each session and area[36]. This approach therefore tested the effect that single PAC neurons had on information encoding within a neural population (Fig. 6c). To determine a functional specificity of PAC neurons as a group, we further compared the maximal decoding performance before and after all PAC neurons were removed from the neuronal ensemble in each session. We did this for all sessions that had at least one PAC neuron, and at least two neurons left after removing all PAC neurons. We then compared this effect with removing the same number of non-PAC neurons from the ensembles (averaged across 500 iterations of random selections).

We quantified the effect of noise correlations on the geometry of the population response. The effect that noise correlations have on the encoded information in a population of neurons depends on the angle between the signal and the noise axis[37]. To illustrate how the angle between the noise and the signal axes changes with noise correlations, we first simulated neural responses for a population of neurons that were partially tuned to two different categories (see Fig. 6f for a simulation of two neurons for which one was tuned and the other was untuned to category). Firing rates for each neuron were drawn from a normal distribution with variable variance. We simulated 200 trials for each category. For tuned neurons, a variable offset was added to the mean

of one of the categories. To add noise correlations to the population of neurons, in each trial, we added a random number drawn from a normal distribution to the FRs of all neurons. To compare our simulation to a condition with removed noise correlations, we shuffled trials within each category to destroy noise correlations within conditions but leave signal correlations among neurons intact. We then determined the signal axis by training a linear SVM classifier on the FRs from all neurons and extracting the hyperplane (decision boundary) obtained from the model. The signal axis is defined as a vector orthogonal to that plane. The noise axis was determined by extracting the first principal component of the data across both categories.

We then quantified and compare the angle between the signal and the noise axis in the recorded data (Fig. 6g). For each recording session, we extracted the signal and the noise axis for the neuronal ensemble at which the difference in category decoding was maximal between removed and intact noise correlations. For this analysis, we included all sessions that had at least two hippocampal neurons available. To obtain the direction of the signal axis, we extracted the hyperplane from each of the ten trained binary SVM classifiers (trained on 80% of the data; one-versus-one decoding, see above) and derived a vector orthogonal to that plane using a QR decomposition. The noise axis was determined by extracting the first principal component of the data across categories. The resulting angle between the two vectors was determined and then averaged across all 10 binary learners and all 500 decoding repetitions, resulting in one angle per session separately for intact and removed noise correlations.

To further determine the functional specificity of PAC neurons, we projected the population responses onto the signal axes and determined the variance of the projection values before and after PAC neurons were removed from the ensembles at which the difference between intact and removed noise correlations was maximal (Fig. 6h). This analysis was performed for all sessions that had at least one hippocampal PAC neuron, and at least two neurons left after removing all PAC neurons from those ensembles. The rationale of the analysis was based on the idea that the variance of the projected values should be small when the angle between the noise and signal axis is large and vice versa[38]. For each binary classifier, we projected the population responses for each category onto the signal axis and determined their s.d. We then averaged the obtained variances across both categories, all 10 binary classifiers and all 500 iterations, and tested the variances between intact and removed noise correlations before and after all PAC neurons were removed from the ensembles.

To compare noise correlations between fast and slow RT trials, we examined all possible PAC–category cell pairs in a given session (Fig. 6i). We analysed the trials in which the preferred or non-preferred categories of the category cell were held in WM separately. Fast and slow RT trials were defined by median split, computed separately in each load condition, and then averaged to avoid a bias of load in RTs. To assess the specificity of the fast versus slow RT trial difference to PAC neurons, we randomly paired category neurons with any other non-PAC neuron and compared noise correlations between fast and slow RT trials (for $n = 162$ randomly selected pairs; same $n$ as for PAC-to-category neuron pairs).

The significance of population decoding (Extended Data Fig. 7a) was assessed by comparing the original decoding accuracy to a distribution of 1,000 decoding accuracies after randomly shuffling category labels.

## Statistics

Throughout this Article, we use (cluster-based) nonparametric permutation tests (statcond.m as implemented in EEGLab, using option 'perm', or ft_freqstatistics.m in FieldTrip), that is, tests that do not make assumptions about the underlying distributions, or mixed-effects GLMs (fitglme.m in MATLAB) to assess statistical differences between conditions. In these tests, random permutations of condition labels were performed to estimate an underlying null distribution, which was then used to assess the statistical significance of the effect. The paired permutation $t$-tests that we performed are equivalent to computing pair-wise condition differences and testing the differences against zero. All permutations statistics used 10,000 permutations, and $t$-tests were tested two-sided unless stated otherwise. The corresponding $t$ and $F$ estimates, which are computed based on a normal distribution, are provided as a reference only. Bayes factors were computed using the BayesFactor package[79]. $BF_{01}$ indicates the evidence of H0 (null hypothesis; no evidence between conditions) over H1. A value of 1 indicates equal evidence for H0 and H1, and values larger than 1 indicate more evidence for H0 over H1 and vice versa. SFC estimates tested across several frequencies were corrected for multiple comparisons using cluster-based permutation statistics as implemented in FieldTrip[80] with 10,000 permutations and an alpha level of 0.025 for each one-sided cluster, which was also Bonferroni corrected for the number of tests involved. Depending on whether we used $z$-scored FRs or spike counts, we used mixed-effects GLMs based on a normal or Poisson distribution, respectively. Finally, error bars shown in figures show the s.e.m. unless otherwise stated.

## Reporting summary

Further information on research design is available in the Nature Portfolio Reporting Summary linked to this article.

## Data availability

All data used in this study are publicly available in the DANDI Archive[59] (https://dandiarchive.org/dandiset/000673). The published dataset contains the timestamps and waveforms of the sorted neurons, LFPs, electrode coordinates, behavioural data, as well as the stimuli, triggers, experimental parameters, anonymized patient metadata of each session.

## Code availability

Example code to reproduce the results is published at GitHub[81] (https://github.com/rutishauserlab/SBCAT-release-NWB) and Zenodo[67] (https://doi.org/10.5281/zenodo.10494533).

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

**Acknowledgements** We thank the patients who volunteered to participate in this study; the members of the clinical teams at Cedars-Sinai Medical Center, J. Chung, L. Bateman, and the staff at Toronto Western Hospital and John's Hopkins School of Medicine for patient management and care and support of data acquisition; T. Rusch, J. Minxha and S. Fusi for discussion; N. Chandravadia for data processing; and I. Reucroft for data acquisition. This study was supported by a German National Academy of Sciences Leopoldina Postdoctoral fellowship (to J.D.), a Center for Neural Science and Medicine at Cedars-Sinai Postdoctoral fellowship (to J.D.), the BRAIN initiative through the National Institute of Neurological Disorders and Stroke (U01NS103792 and U01NS117839 to U.R.) and the National Science Foundation (BCS-2219800 to U.R.).

**Author contributions** J.D., J.K. and U.R. conceived the project. J.D., J.K., A.G.P.S. and Y.S. performed experiments. J.D., J.K. and U.K. performed data analyses. C.M.R. provided patient care and supported data acquisition. M.K. managed the data release and demo code. T.A.V., W.S.A. and A.N.M. managed patients and performed surgeries. J.D. and U.R. wrote the manuscript with input from all of the authors. J.D. and U.R. acquired funding.

**Competing interests** The authors declare no competing interests.

**Additional information**
**Correspondence and requests for materials** should be addressed to Jonathan Daume or Ueli Rutishauser.

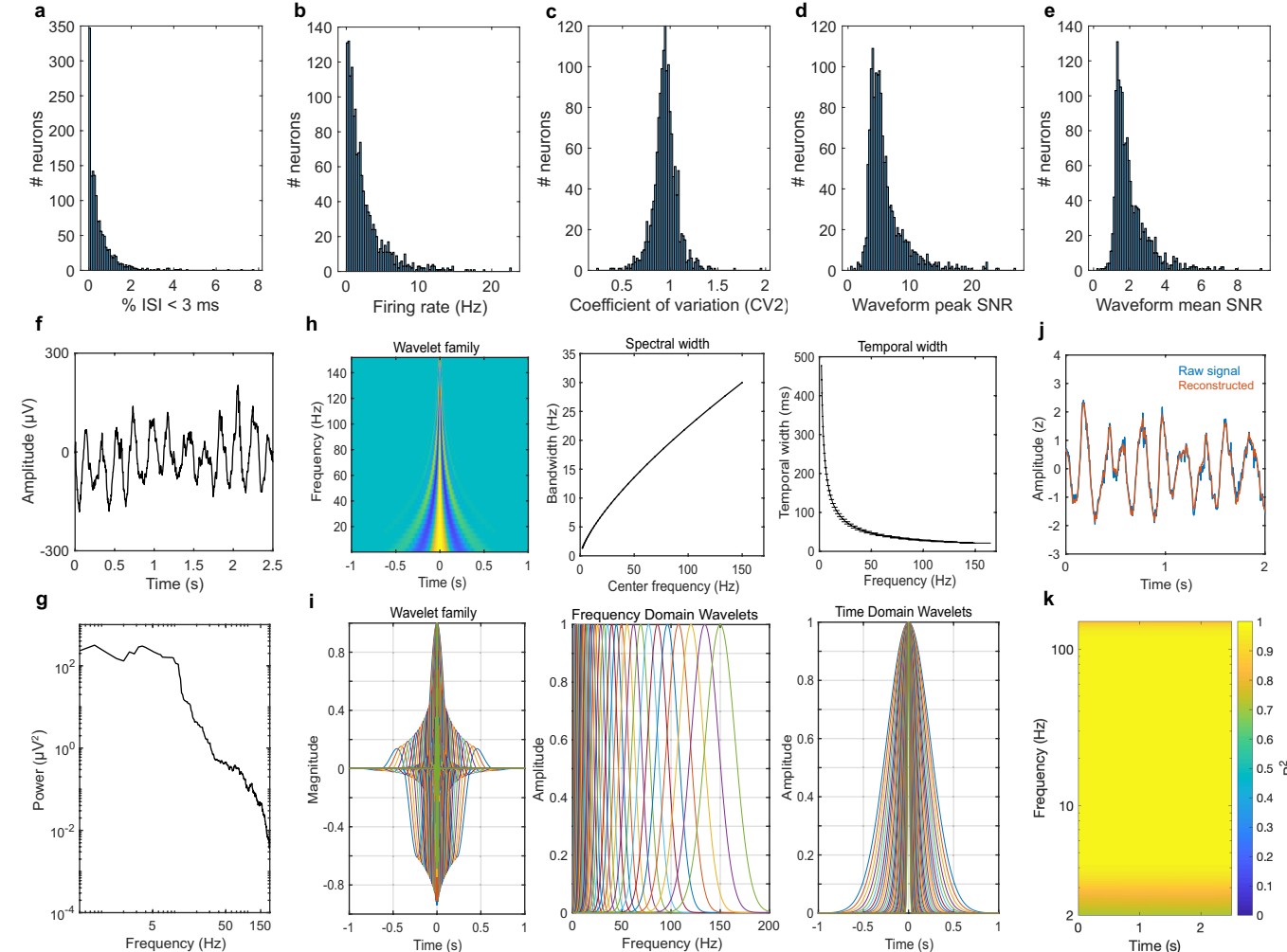

**Extended Data Fig. 1 | Spike-sorting quality metrics for all identified putative single units and wavelet characteristics.** (a-e) Spike-sorting quality metrics. (**a**) Proportion of inter-spike intervals (ISI) below 3 ms. (**b**) Average firing rate. (**c**) Coefficient-of-variation. (**d**) Signal-to-noise ratio (SNR) for the peak of the mean waveform across all spikes as compared to the standard deviation of the background noise. (**e**) Mean SNR of the waveform. (**f**) Example raw LFP recorded in a hippocampal channel during the delay period of a single trial (time 0 denotes onset of the delay period). (**g**) Power-spectrum of LFP data shown in (f). (**h**,**i**) Wavelet characteristics for all 40 wavelets used. Left: Wavelet family. The upper panel shows the temporal outline and the magnitude of the real part for all wavelets smoothed across all frequencies. The maximal magnitude of each wavelet is scaled to 1. Warm colours denote positive, cold colours negative magnitude. The lower panel shows the real part of all wavelets plotted on top of each other. Centre: The upper panel shows the spectral bandwidth of each wavelet as a function of centre frequency. The lower panel plots the FFT-spectrum for each wavelet. Right: The upper panel shows the temporal width of all wavelets as a function of centre frequency. The horizontal lines indicate the spectral bandwidth for each wavelet. The lower panel contains the amplitude envelope for each wavelet as a function of time. (**j**) Example original and reconstructed signal after applying the continuous wavelet transform (see Methods). Small deviations from the original signal are due to the fact that signals at frequencies lower or higher than the edge frequencies of 2 and 150 Hz, respectively, were not represented by the wavelet transform but present in the original signal. (**k**) Assessment of the wavelet-based signal reconstruction. We computed linear models using the reconstructed signal as predictor for the original signal and extracted R-squared values as a function of time and frequency in each trial and channel. Values were averaged across all trials and all hippocampal channels. An R-squared values of close to 1 indicates almost perfect reconstruction of the original signal. As stated above, the slight drop in reconstruction quality at extreme frequencies is explained by the fact that signals at frequencies lower or higher than the edge frequencies, respectively, were not represented by the wavelet transform but present in the original signal.

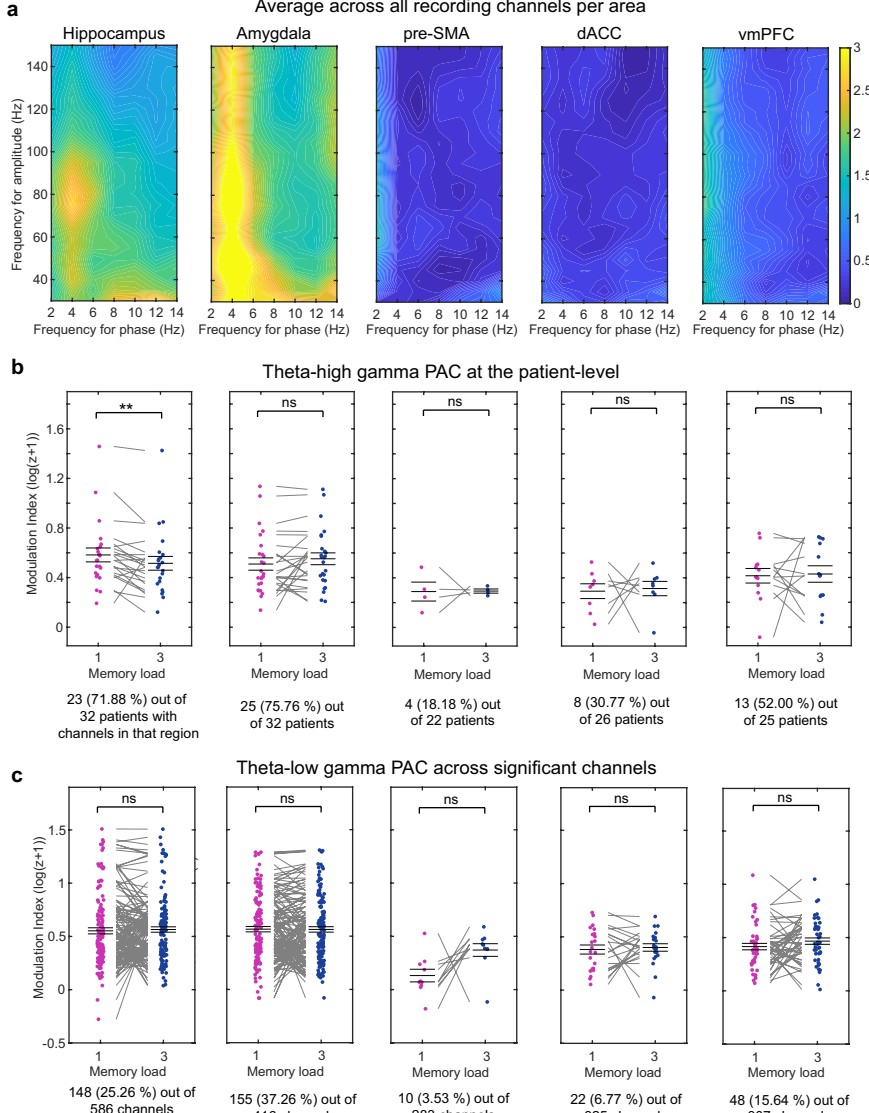

**Extended Data Fig. 2 | Additional PAC analyses. (a)** PAC comodulogram averaged across all channels separately for each area. Strongest PAC was observed between theta and gamma in both areas of the MTL. Frontal areas did not show strong PAC, with weak PAC at <2 Hz in pre-SMA and vmPFC. We focused our analysis on frequencies above 2 Hz (1–3 Hz bandpass) to ensure that at least 2 full cycles fit within our analysis window of 2.5 s (length of maintenance period). **(b)** In addition to testing theta-gamma PAC across significant channels (see main text), we tested PAC between the load conditions across patients after averaging all significant PAC channels within each patient. The results were similar to the analysis across channels, with strongest PAC in MTL areas and only weak PAC in frontal channels (see percent of patients with significant PAC channels below each figure), suggesting that the results were not driven by channels from a single patient. Only in the hippocampus again, PAC was stronger for load 1 as compared to load 3 (n = 23 patients, p = 0.0049), observable in

almost each single patient. No significant differences were found between the load conditions in other regions (amygdala: n = 25; pre-SMA: n = 4; dACC: n = 8; vmPFC: n = 13). z-scored PAC values were shifted into a positive range by an offset of 1 and log-transformed for illustrative purposes only. All statistics are based on non-transformed z-values. **(c)** Theta to low gamma (30–55 Hz) PAC analyses. We also found strongest theta-low gamma PAC in MTL regions as opposed to frontal regions (see percentages below each figure). But for the low gamma band, we did not observe significant differences between the load conditions in any of the regions (hippocampus: n = 148 channels; amygdala: n = 155; pre-SMA: n = 10; dACC: n = 22; vmPFC: n = 48). See Supplementary Table 2 for additional PAC analysis separated into slow and fast theta. In (b,c), we performed two-sided permutation-based t-test and centre values denote mean ± s.e.m.; ** p < 0.01; ns = not significant.

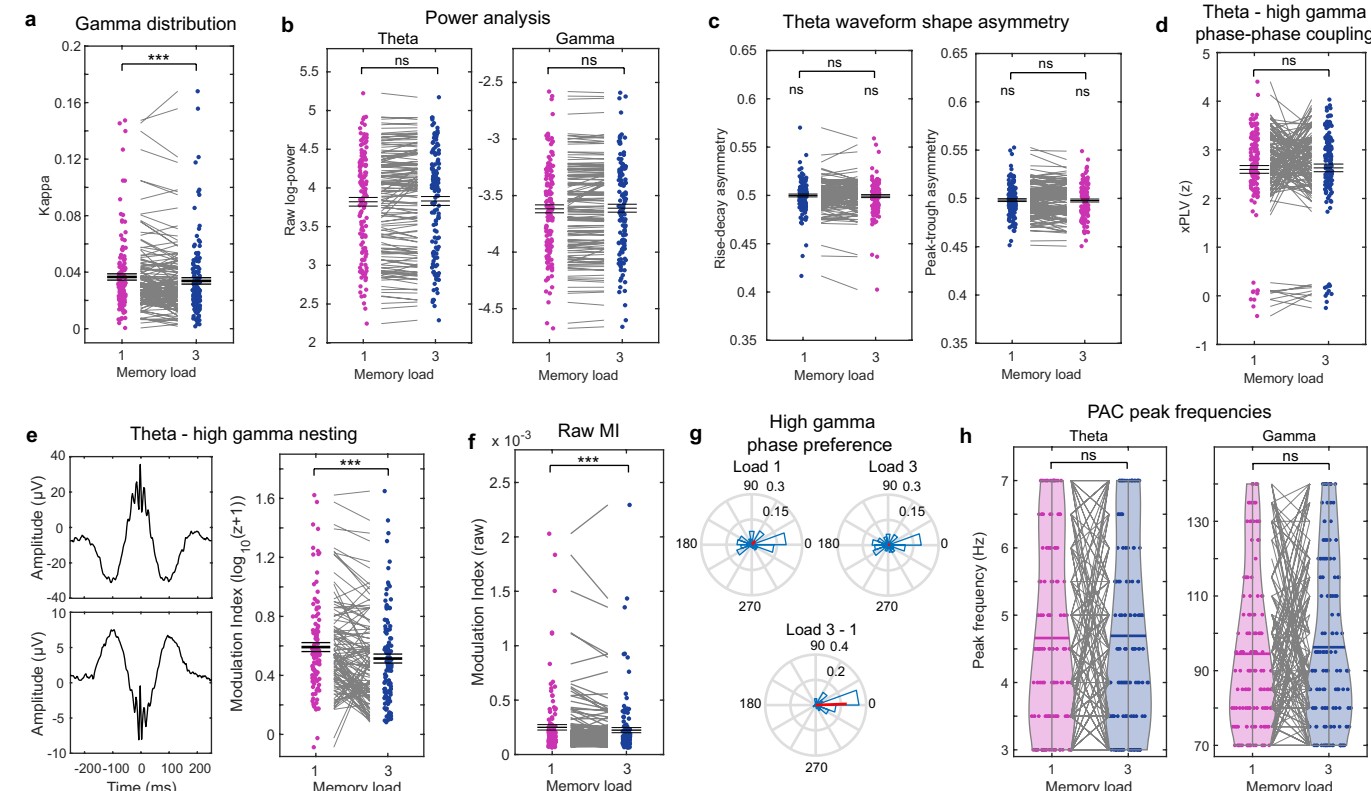

**Extended Data Fig. 3 | Theta-high gamma PAC control analyses in the hippocampus.** (**a**) Higher memory loads are thought to be accompanied by a wider distribution of gamma amplitudes across theta phases, thereby leading to lower PAC values[82]. To quantify the width of the distribution of gamma amplitude as a function of theta phase in load 3 than load 1, we estimated kappa. Kappa is a measure that describes the concentration (inverse of variance) of a circular variable around the mean direction. Across all PAC channels, kappa was significantly lower in load 3 compared to load 1 (n = 137 channels, t(136) = −3.7453, p = 0.0001) trials. This shows that gamma amplitudes are high for a wider range of theta phases for higher memory loads, thereby explaining why PAC decreases for higher memory loads. (**b**) Comparison of theta and gamma power. The significant hippocampal PAC channels showed no significant differences in theta or gamma power between the two load conditions (n = 137). (**c**) To determine the influence of theta waveform shape on PAC, we tested for differences in theta waveform peak-to-trough as well as rise-to-decay asymmetries between the two load conditions (see Methods). We did not find systematic differences between the conditions for both measures (n = 137). Moreover, average theta waveforms were overall symmetric as both measures were not significantly different from .5 in any of the conditions. (**d**) Moreover, if the differences between the load conditions observed for PAC channels in the hippocampus were explained by waveform shape differences/theta harmonics, we should also observe an effect for cross-frequency phase-phase coupling between the same frequency bands. We tested for that in all significant hippocampal PAC channels and did not observe a significant difference (n = 137). Theta-high gamma phase-phase coupling was computed as described in[73,83]. (**e**) We determined the number of significant PAC channels that showed theta-high gamma nesting as described by Vaz et al.[69] The left upper and lower panels

show two examples of significant PAC channels from the hippocampus that were determined to have nesting by the Vaz et al. method (at least three local maxima within a window of 45 ms around the preferred phase (see Methods)). 110 of the 137 significant PAC channels (80.29%) in the hippocampus showed nesting between high gamma and theta. When testing PAC between the load conditions after removing channels that did not show significant nesting, PAC was still significantly lower in load 3 than 1 (n = 110; t(109) = −4.10; p = 0.0001). (**f**) Comparison of theta-gamma PAC strength in the hippocampus assessed using the raw modulation index rather than the of z-transformed MI. Raw theta-gamma PAC was significantly larger in load 1 compared to load 3 (n = 137; t(136) = −4.0264, p = 0.0001). (**g**) Distribution of theta phases at which gamma amplitude was maximal across all significant PAC channels in the hippocampus in load 1 and 3 (upper part). In most channels, gamma amplitude was maximal at the peak or the trough of theta. Note that the local referencing scheme in our data does not allow do make statements about the polarity of theta. Red bars indicate the mean vector length across all phases. The difference in theta phase at which gamma amplitude was maximal between the two load conditions was not significantly different from zero (bottom part). (**h**) We further assessed whether PAC peak frequencies differed between the load conditions either within the theta or the gamma band. To do so, we recomputed PAC using a finer resolution for phase frequencies (i.e., a step size of 0.5 instead of 2 Hz) and determined the frequency bin for which PAC was maximal for the theta and the gamma band separately for all channels and both loads. We did not find significant systematic shifts in PAC peak frequencies between the load conditions in theta or gamma frequencies (n = 137). In (a-f,h), we performed two-sided permutation based-t-tests and centre values denote mean ± s.e.m. *** p < 0.001, ns = not significant.

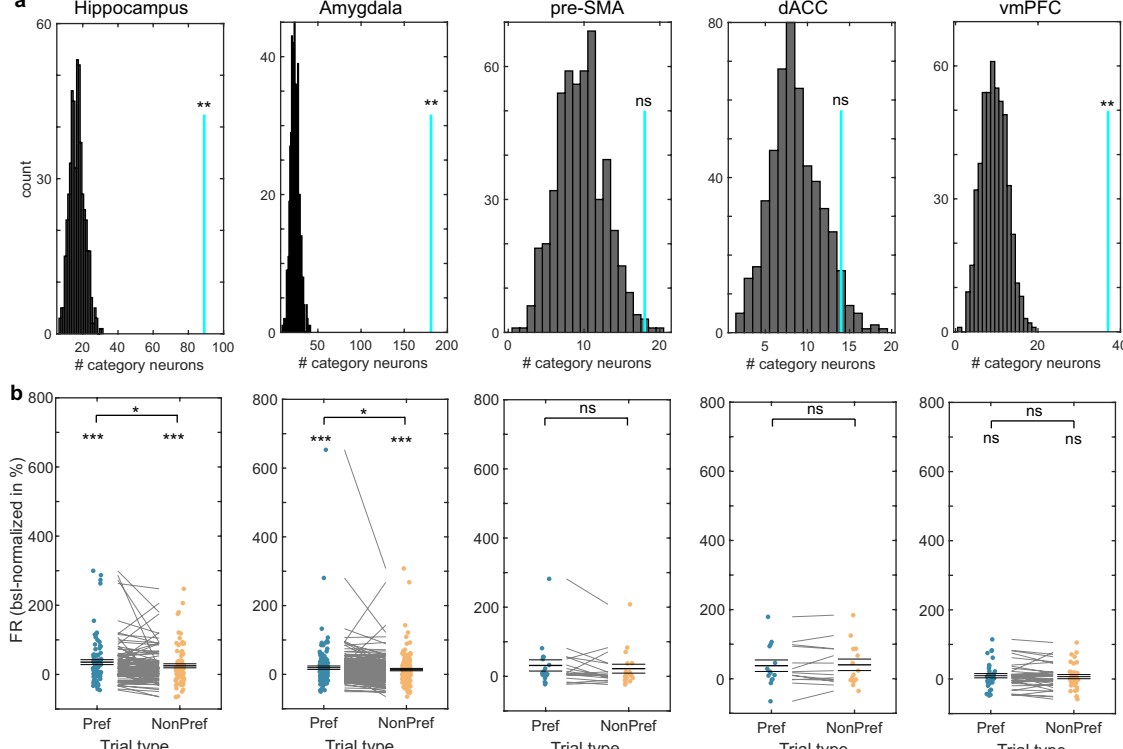

**Extended Data Fig. 4 | Category neuron selection and persistent activity.**
(**a**) In hippocampus, amygdala, and vmPFC (all p = 0.002, right-sided permutation test) the selected number of category neurons was larger than expected by chance (p < 0.01; Bonferroni corrected; see Methods). The null distribution (grey) was estimated by repeating the selection procedure after shuffling the category labels for 500 times. Numbers of selected neurons in dACC and pre-SMA were not significantly different from those expected by chance. (**b**). Category neurons in both areas of the MTL, hippocampus (n = 89; pref. vs. baseline: p = 0.0001; non-pref.: p = 0.0001) and amygdala (n = 181; pref.: p = 0.0001; non-pref.: p = 0.0001), showed persistent activity during the delay period of the task, during which no picture was presented on the screen. Note that category neurons were selected during the encoding period only, making the delay period independent from the selection criteria. FR remained

significantly higher when their preferred as compared to non-preferred categories were maintained in memory (hippocampus: p = 0.025; amygdala: p = 0.037). The activity of category neurons in the vmPFC (right) during the delay period was not significantly larger than that during baseline (n = 37; pref.: p = 0.12; non-pref.: p = 0.26). Also, their FRs did not differ significantly between when the preferred and the non-preferred category of a cell was maintained in WM (t(36) = 1.03, p = 0.32, $BF_{01}$ = 3.47). The FR of vmPFC neurons thus went back to baseline levels when no stimulus was presented on the screen. FR for selected neurons in the pre-SMA (n = 18) and dACC (n = 14) are shown only for completeness despite the proportion of these cells not exceeding those expected by chance. All comparisons are based on two-sided permutation-based t-tests. Centre values denote mean ± s.e.m. *** p < 0.001, ** p < 0.01, * p < 0.05, ns = not significant.

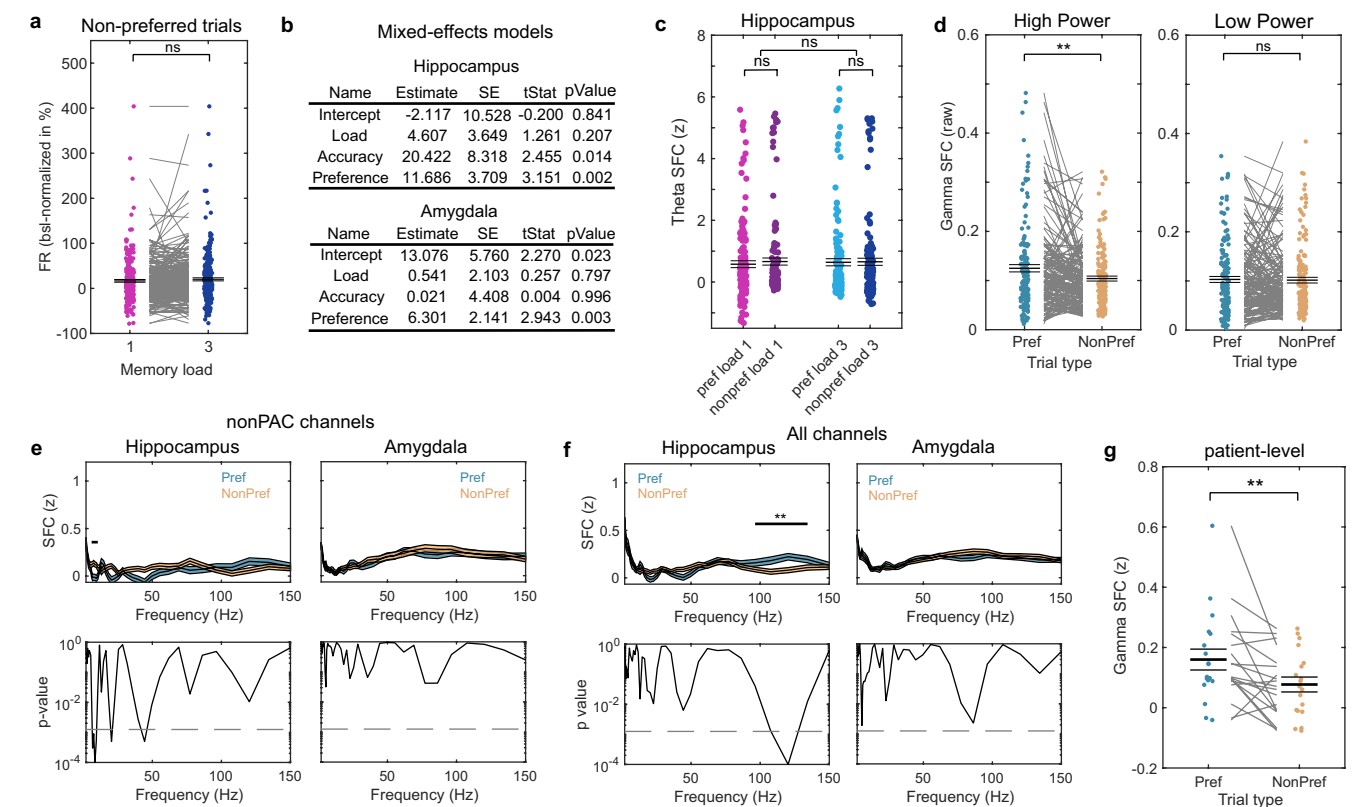

**Extended Data Fig. 5 | Additional analyses for category cells in the MTL.**
(**a**) Unlike for preferred trials, we did not observe a load effect for MTL category cells when non-preferred categories were maintained during the maintenance period (n = 270). (**b**) In addition to our statistics across single neurons, we performed nested random-intercept GLMs for patient-level statistics[84]. In the hippocampus, FR of category cells was significantly higher for preferred as compared to non-preferred as well as for correct vs. incorrect trials. There was no main effect of load as expected, which only emerged when we tested for load differences in preferred trials only (data not shown). In the amygdala, we observed a significant effect for preference, where FR was higher in preferred than non-preferred trials. Again, the load effect was only significant when tested in preferred trials only (data not shown). There was no effect for accuracy. (**c**) When averaging theta band (3–7 Hz) SFC values for hippocampal category neurons paired with significant PAC channels, we did not observe a significant main effect for *load* or *preference* nor a significant interaction (n = 151; permutation-based F-test). (**d**) We performed a median split of gamma amplitudes across trials and tested gamma SFC between category cells and significant PAC channels in the hippocampus separately for spikes that occurred during high and low gamma amplitudes (spike counts were adjusted across conditions). We observed a significant difference in gamma SFC between preferred and non-preferred trials only for spikes that occurred during high

(n = 151, p = 0.0015), not during low gamma amplitudes (n = 151, p = 0.84). (**e**) When paired with non-PAC channels, we did not observe differences in the gamma band between preferred and non-preferred trials for category cells in hippocampus or amygdala. In the hippocampus, we observed a significant difference in the alpha range (7–11 Hz) with SFC for non-preferred trials higher than for preferred trials, which we did not further consider in our analyses. (**f**) Comparing SFC for category neurons across all channels (not separated into PAC/Non-PAC channels) revealed significantly higher gamma-band SFC for preferred than non-preferred trials in the hippocampus (cluster-p = 0.007, two-sided cluster-based permutation *t*-test with Bonferroni-corrected alpha-level for two MTL areas), similar to what we observed for PAC channels only. There were no significant differences in the amygdala. (**g**) To test whether the gamma SFC effect for category cells in the hippocampus persisted at the patient level, we averaged gamma SFC across all category neuron to channel pairs within each patient and then compared the per-patient average between preferred and non-preferred trials. Patient-averaged gamma SFC was significantly higher for preferred trials, suggesting that the effect was not driven by a few channels or patients (n = 19, t(18) = 2.8512, p = 0.005). In (a,c,d,g), we performed two-sided permutation-based *t*-tests. Centre values demote mean ± s.e.m (coloured areas in e,f). *** p < 0.001; ** p < 0.01; * p < 0.05; ns = not significant.

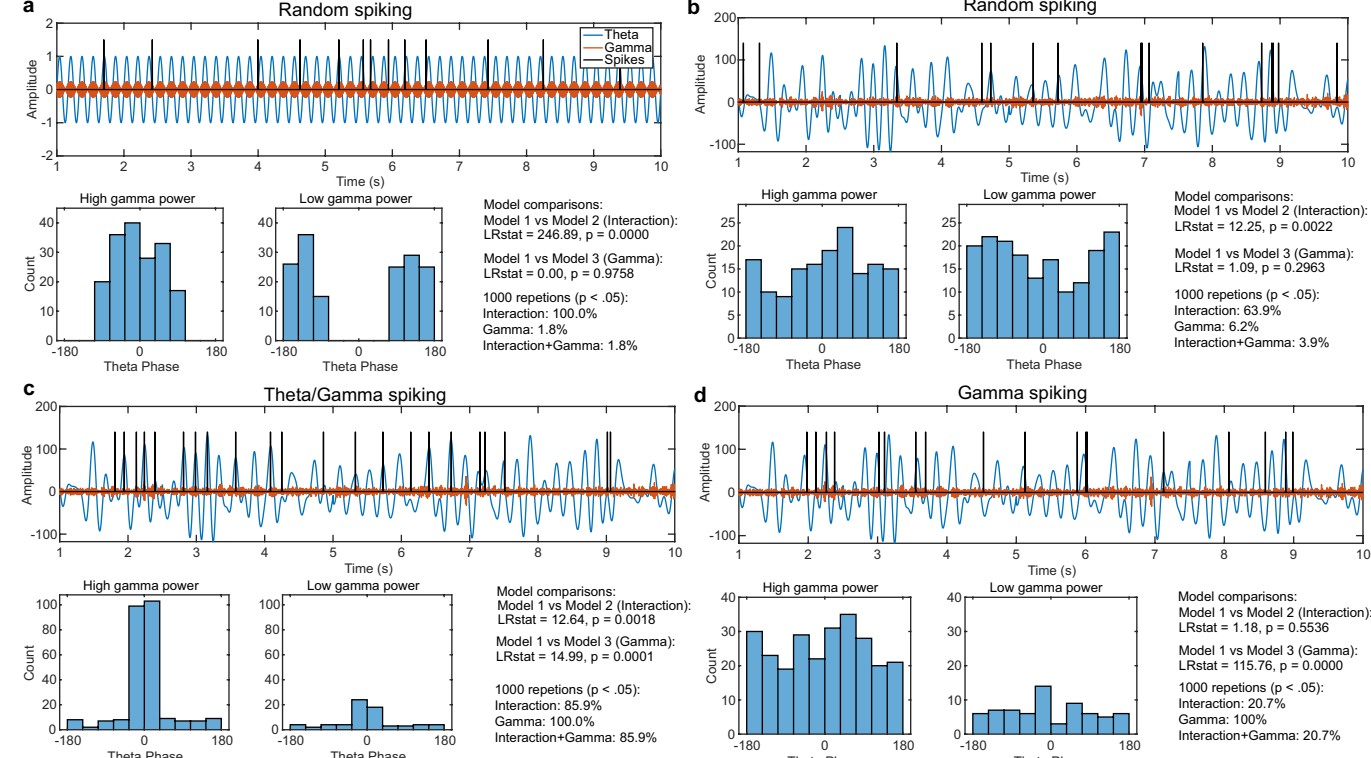

**Extended Data Fig. 6 | Simulations supporting the PAC neuron selection approach.** We argue that a neuron that fires randomly with respect to theta phase and gamma power could still be selected as a PAC neuron if only the GLM interaction term between theta phase and gamma amplitude is considered. In addition to selecting neurons whose FRs are better explained by a model including an interaction term as compared to a model with no interaction term, we therefore introduced a second criterion by comparing the full model against a model that lacks the gamma amplitude term. The simulations presented here are meant to visualize our reasoning. In (a), we simulate theta (6 Hz) and gamma (80 Hz) signals, where gamma amplitudes perfectly couple to theta phase. This highly artificial LFP signal only serves to simplify visualization. We also include illustrations in (b) to (d) using an originally recorded LFP channel from our dataset (filtered between 3–7 Hz and 70–140 Hz) that shows strong levels of PAC. This is to show that the same arguments also hold for real data. For the purpose of these illustrations, we used an LFP signal of roughly 160 s length and simulated 300 spike timestamps (black ticks), of which 9 s are plotted. (**a**) In this simulation, we modelled random spike timestamps with respect to theta phase and gamma amplitude (upper panel). According to our GLM selection approach, we grouped spikes in 10 theta phase bins and 2 gamma amplitude bins and determined spike counts in each bin (lower panels). As can be seen from the histograms in the lower panels, the theta phase distribution of spike counts differs between low and high gamma amplitudes, resulting in a highly significant interaction term between theta phase and gamma amplitude. The reason for this is that gamma amplitude itself is already perfectly coupled to theta phase. Separating spikes into low and high gamma will therefore also

result in different theta distributions among the two spike count groups. Thus, when testing a model that contains theta phase and gamma amplitudes as well as their interaction against a model without the interaction, spike counts will be highly significantly better explained by the full model, as was the case in this example (p < 0.001; see likelihood-ratio test results on the right). However, since the time stamps are random, we should not observe a difference in overall spike counts between low and high gamma amplitudes, which was also the case in this example (p = 0.98). Introducing such a gamma term comparison as a second selection criterion thus ensures that this simulated random neuron would not have been selected. In 1000 repetitions of this simulation, our approach would have selected only 1.8% of such randomly spiking neurons (see text on right side). (**b**) Similar to (a) but using a real LFP recording from our dataset that shows strong levels of PAC. 300 spike timestamps were again modelled randomly with respect to theta phase and gamma amplitude. Similar albeit weaker statistics were observed in these simulations. (**c**) Using the same LFP as in (b) but now simulating 300 spike timestamps that prefer high gamma amplitudes and a theta phase of 0 (i.e., PAC spiking plus 10% noise). Here, as desired, the full model explains spike counts significantly better than both the other models and this neuron would be selected as a PAC neuron. (**d**) In this example, we simulate a "gamma neuron", i.e., a neuron whose FR follows gamma amplitude, but not theta phase. In most cases (79.3% of 1000 repetitions), these gamma neurons were successfully rejected. Since we did not control for theta phase in these simulations using a strong LFP channel, however, around 20% of the simulations modelled PAC rather than pure gamma spiking.

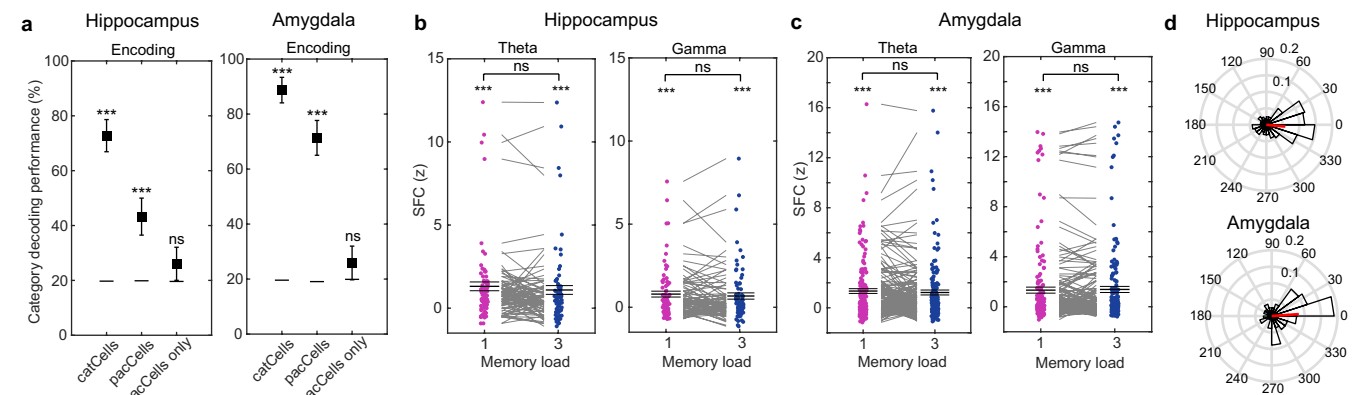

**Extended Data Fig. 7 | SFC and theta phase shift analysis for PAC neurons.**
(**a**) PAC neurons were not selective for category in hippocampus (left) and amygdala (right). Even during encoding, category could not be efficiently decoded from FR of "PAC only" neurons (all other p = 0.001, right-sided permutation test). Decoding performance is shown as mean ± s.d. across 1,000 decoding repetitions. Black horizontal lines indicate mean decoding of 1,000 randomly shuffled category labels (chance level). Decoding was performed for pseudo-populations of category or PAC neurons, respectively. (**b,c**) Theta and gamma SFC between PAC neurons and local LFP recordings did not differ as a function of load in (**b**) the hippocampus (theta: t(78) = −1.54, p = 0.13; gamma:

t(78) = −1.12, p = 0.27, n = 79), or (**c**) the amygdala (theta: t(162) = −0.71, p = 0.47; gamma: t(162) = 0.76, p = 0.45, n = 163). Theta and gamma SFC, however, were both significantly stronger than shuffled surrogates in both areas (all p = 0.0001). Each dot is a neuron-channel combination. In (a,b), we performed two-sided permutation-based *t*-tests and centre values denote mean ± s.e.m. (**d**) The preferred theta phase of PAC neurons did not differ significantly as a function of load in both areas of the MTL. Red bars show the mean difference in preferred theta phases between load 1 and 3 across all PAC neurons. *** p < 0.001; ns = not significant.

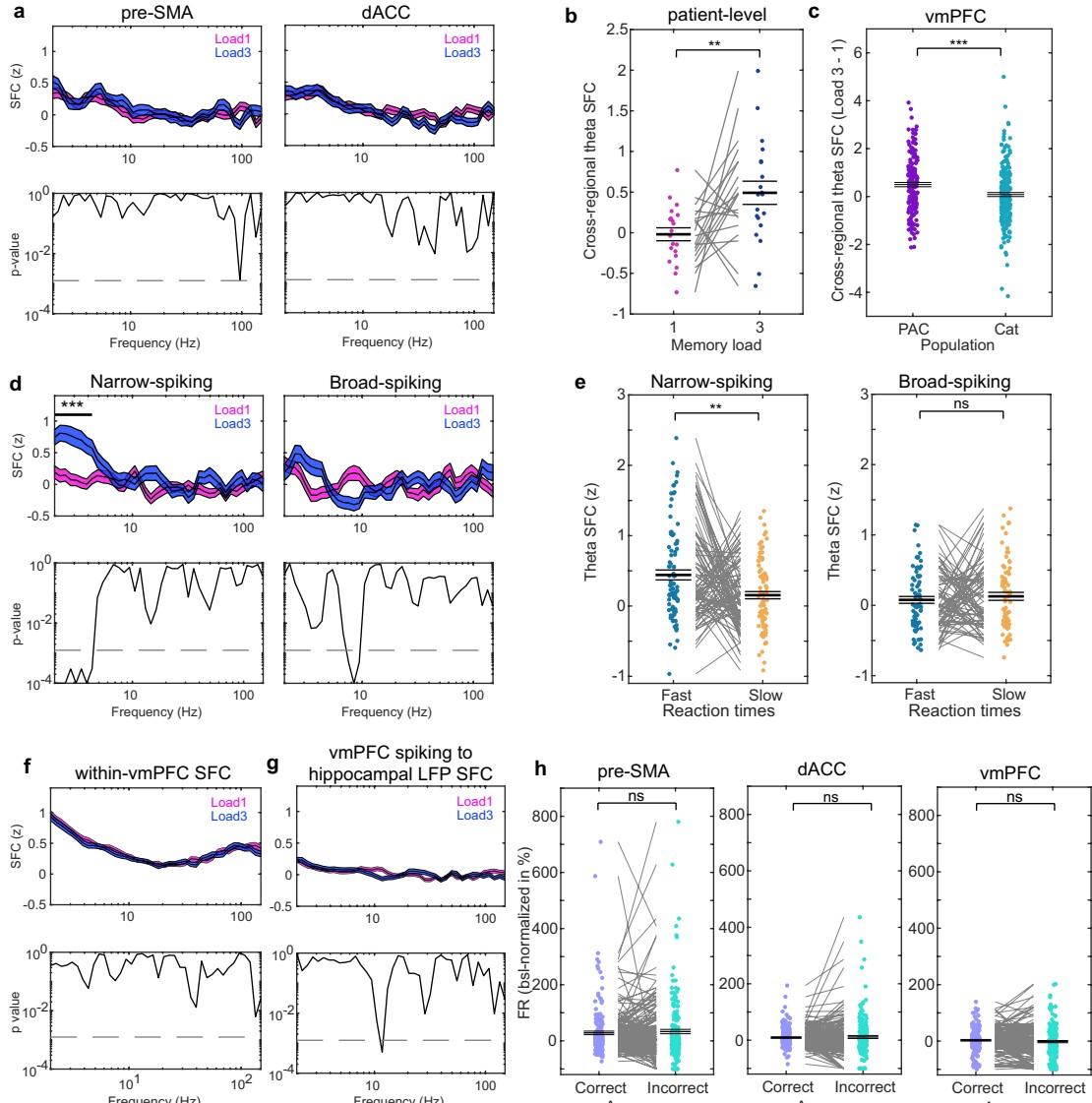

**Extended Data Fig. 8 | Cross-regional SFC for pre-SMA, dACC, and vmPFC, as well as for fast- and broad-spiking PAC neurons from the hippocampus.** Related to Fig. 5. (**a**) Cross-regional SFC between hippocampal PAC neurons and LFPs recorded in pre-SMA (left) or dACC (right) did not reveal any difference between the two WM load conditions in any of the frequencies. (**b**) To test whether the load modulation of theta-band cross-regional SFC between hippocampal PAC neurons and vmPFC LFPs persisted on the patient level, we averaged theta SFC across all PAC neuron to channel pairs within each patient and then compared the within-patient averages between the two load conditions. At the patient-level, theta cross-regional SFC was significantly higher for load 3 than load 1 trials, suggesting that the effect was not driven by a few channels or patients (n = 20; t(19) = −2.8297, p = 0.0071). (**c**) Comparison of cross-regional hippocampal-vmPFC SFC between PAC and category neurons revealed the load modulation of cross-regional theta SFC between hippocampal neurons and vmPFC LFPs was significantly stronger for PAC than for category neurons (PAC: n = 175, Cat: n = 215; t(376.07) = 3.3942, p = 0.0001; unpaired two-sided permutation *t*-test). Each dot is a neuron-channel combination. (**d**) Earlier work has suggested that cognitive control might especially be governed through long-range connections between frontal and sensory regions that target inhibitory interneurons to (dis-)inhibit local circuitries[41,53,85]. We thus asked if we observe a differential effect for the hippocampal PAC neuron connections after separating them into narrow- and broad-spiking

neurons based off their waveform shapes, which has been suggested to categorize neurons into inhibitory and excitatory neurons, respectively[86,87]. For analysis of connections involving narrow- and broad-spiking PAC neurons separately, we observed a significant difference in theta SFC between load 3 and load 1 only for the narrow-spiking PAC neurons (trough-to-peak time <0.5 ms; n = 91 connections; cluster-p = 0.0001, left). No effect was found for broad-spiking PAC neurons (n = 84) (**e**) Similarly, theta SFC for fast RT was significantly stronger than for slow RT only for narrow-spiking (t(90) = 3.02, p = 0.003, n = 91, left), not for broad-spiking PAC neuron connections between hippocampus and vmPFC (t(75) = −0.66, p = 0.52, n = 76, right; spikes were median split into fast and slow RT trials per load condition and then averaged across loads to avoid potential confounds). (**f,g**) We did not find significant differences between the load conditions for (**f**) within-region SFC or (**g**) cross-regional SFC to hippocampal LFPs across all neurons from the vmPFC. (**h**) We further tested whether there were any non-specific global state changes between correct and incorrect trials in any of the three frontal regions. FRs for all neurons recorded in the three frontal areas were not significantly different between correct and incorrect trials during the delay period (pre-SMA: n = 201; dACC: n = 180; vmPFC: n = 201). In (a,d,f,g) we performed two-sided cluster-based permutation *t*-tests, centre values denote mean, coloured areas s.e.m. In (b,c,e,h), we performed two-sided permutation *t*-tests and centre values denote mean ± s.e.m. *** p < 0.001; ** p < 0.01; ns = not significant.

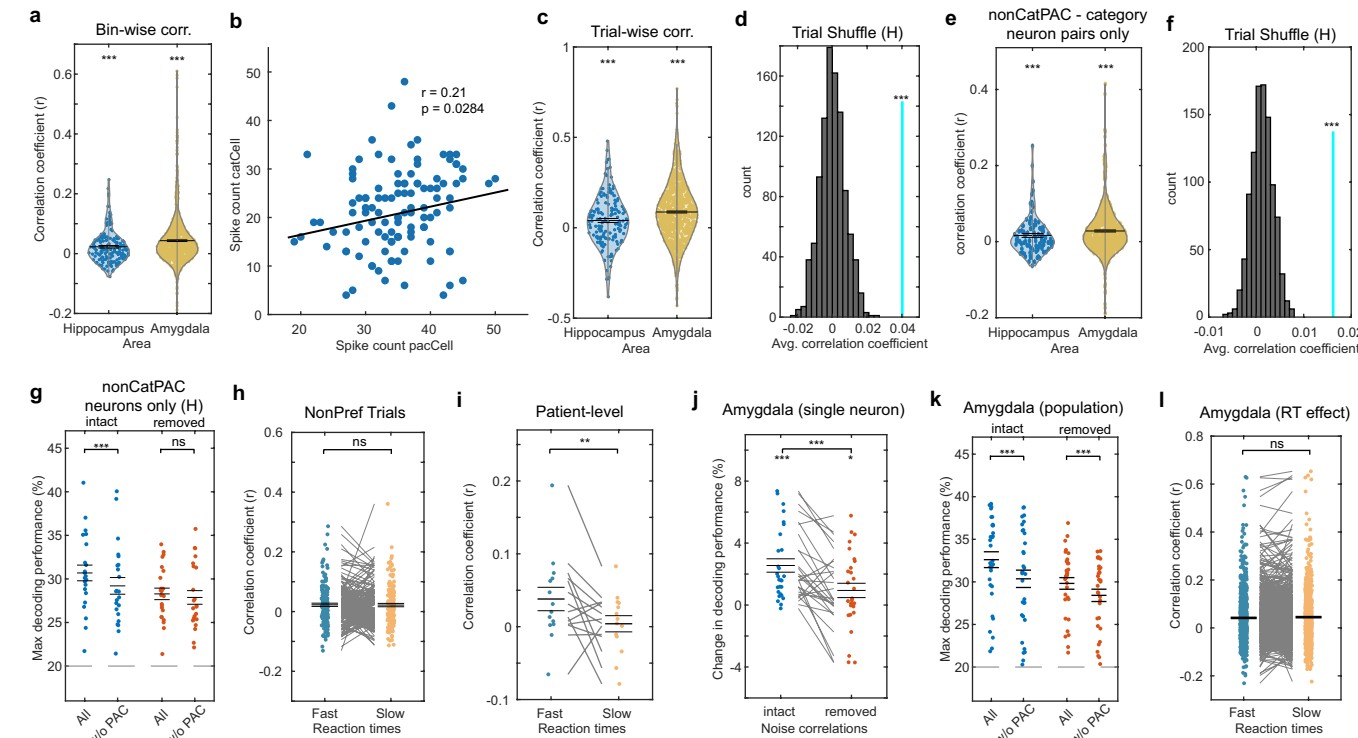

**Extended Data Fig. 9 | Further analysis of noise correlations among PAC and category neurons.** Related to Fig. 6. (**a**) Distribution of trial-averaged, bin-wise correlation coefficients for all possible pairs of category and PAC neurons in the hippocampus (n = 162) and amygdala (n = 892). In both regions, correlation coefficients were significantly higher than zero on average (both p = 0.0001). (**b**) Example showing trial-by-trial noise correlations for a pair of simultaneously recorded category and PAC neurons from the hippocampus. Each dot represents the spike count in a correct trial during the maintenance period for each neuron. For this example, the firing rate of the two neurons was positively correlated across trials. (**c**) Correlation coefficients for all possible pairs of category and PAC neurons in the hippocampus (n = 162) and amygdala (n = 892) for trial-by-trial noise correlations, computed within conditions and then averaged. In both regions, correlation coefficients covered a broad range of both positive and negative values and were significantly higher than zero on average (both p = 0.0001). (**d**) Repeat of the trial-wise correlation analysis for all possible PAC-category neuron pairs in the hippocampus. Shuffling trial labels within conditions for 1000 times resulted in far lower correlations between pairs of neurons than unshuffled trial labels (cyan line; mean of correlation coefficients across all pairs), showing that trial shuffling successfully removed noise correlations (p = 0.0001, right-sided permutation test). (**e**) Bin-wise correlations among pairs of category neurons and PAC neurons that were not also category neurons were significantly positive on average in hippocampus (n = 101, p = 0.0003) and amygdala (n = 555, p = 0.0001). (**f**) Correlations between pairs of category neuron and PAC neurons that were not also category neurons in the hippocampus (cyan line). Within-condition

trial shuffling (grey) significantly reduced noise correlations (p = 0.0001, right-sided permutation test). (**g**) Maximal decoding performance for intact and removed noise correlations before and after removing only PAC neurons that were not also category neurons from the ensembles ("nonCatPAC" neurons). Like for all PAC neurons, decoding performance was enhanced by nonCatPAC neurons only when noise correlations were intact (n = 23 sessions, p = 0.0005). (**h**) Bin-wise correlations (averaged across trials) among pairs of hippocampal PAC and category neurons did not differ between fast and slow RT trials for non-preferred trials (n = 162, p = 0.90). Each dot represents the correlations coefficient for a pair after averaging, computed per trial and then averaged across all considered trials. (**i**) Correlations for pairs of PAC and category neurons in preferred trials were averaged within each patient and then compared between fast and slow RTs across all patients (n = 16, p = 0.0027). Each dot is a patient. (**j**) In the amygdala, adding single PAC neurons (n = 28) to the decoding ensemble did not only enhance decodability when noise correlations were intact (p = 0.0001), but also when removed (p = 0.049; intact − removed: p = 0.0009). (**k**) Similarly, removing all PAC neurons from the ensembles in the amygdala – like removing randomly selected cells – led to a significant decrease in decoding for both, intact (p = 0.0001) and removed (p = 0.0001) noise correlations (n = 32 sessions). (**l**) Comparing correlations among PAC and category neurons between fast and slow RT trials in preferred trials did not reveal a significant difference in the amygdala (n = 884 pairs). In (a,c,e,g-l), we performed two-sided permutation-based *t*-tests and centre values denote mean ± s.e.m. *** p < 0.001; ** p < 0.01; * p < 0.05; ns = not significant.

# Reporting Summary

## Statistics

For all statistical analyses, confirm that the following items are present in the figure legend, table legend, main text, or Methods section.

| n/a | Confirmed | |
|---|---|---|
| ☐ | ☒ | The exact sample size (*n*) for each experimental group/condition, given as a discrete number and unit of measurement |
| ☐ | ☒ | A statement on whether measurements were taken from distinct samples or whether the same sample was measured repeatedly |
| ☐ | ☒ | The statistical test(s) used AND whether they are one- or two-sided *Only common tests should be described solely by name; describe more complex techniques in the Methods section.* |
| ☐ | ☒ | A description of all covariates tested |
| ☐ | ☒ | A description of any assumptions or corrections, such as tests of normality and adjustment for multiple comparisons |
| ☐ | ☒ | A full description of the statistical parameters including central tendency (e.g. means) or other basic estimates (e.g. regression coefficient) AND variation (e.g. standard deviation) or associated estimates of uncertainty (e.g. confidence intervals) |
| ☐ | ☒ | For null hypothesis testing, the test statistic (e.g. *F*, *t*, *r*) with confidence intervals, effect sizes, degrees of freedom and *P* value noted *Give P values as exact values whenever suitable.* |
| ☒ | ☐ | For Bayesian analysis, information on the choice of priors and Markov chain Monte Carlo settings |
| ☒ | ☐ | For hierarchical and complex designs, identification of the appropriate level for tests and full reporting of outcomes |
| ☐ | ☒ | Estimates of effect sizes (e.g. Cohen's *d*, Pearson's *r*), indicating how they were calculated |

*Our web collection on statistics for biologists contains articles on many of the points above.*

## Software and code

Policy information about availability of computer code

| | |
|---|---|
| Data collection | Neurophysiological data were collected using the ATLAS (Neuralynx Inc. @ CSMC and UHN) or Blackrock system (Blackrock Neurotech Inc. @ JHU) and hybrid Behnke-Fried depth electrodes (Ad-Tech Inc.). See detailed description in the Methods section. Data is available for download at https://dandiarchive.org/dandiset/000673 |
| Data analysis | Data analyses were performed using MATLAB 2019b, Fieldtrip 20200409, EEGLAB 2019.1, Python 3.10, and bycycle 1.1. Spike sorting was done using OSort 4.1 (https://rutishauserlab.org/osort). Anatomical data analysis was performed using FreeSurfer 6 (surfer.nmr.mgh.harvard.edu). Localization of electrodes was visualized using the CIT168 atlas (https://osf.io/r2hvk/) and the brainnetome atlas (https://atlas.brainnetome.org/). Bayes factors were calculated using the BayesFactor toolbox (https://zenodo.org/records/7006300). Specific functions used in this study are described in the Methods section. Code is available for download at https://zenodo.org/doi/10.5281/zenodo.10494533 |

For manuscripts utilizing custom algorithms or software that are central to the research but not yet described in published literature, software must be made available to editors and reviewers. We strongly encourage code deposition in a community repository (e.g. GitHub). See the Nature Portfolio guidelines for submitting code & software for further information.

## Data

Policy information about availability of data

All manuscripts must include a data availability statement. This statement should provide the following information, where applicable:
- Accession codes, unique identifiers, or web links for publicly available datasets
- A description of any restrictions on data availability
- For clinical datasets or third party data, please ensure that the statement adheres to our policy

> All data used in this study are publicly available in the DANDI Archive59 (https://dandiarchive.org/dandiset/000673/0.240118.2135). The published data set contains the timestamps and waveforms of the sorted neurons, LFPs, electrode coordinates, behavioral data, as well as the stimuli, triggers, experimental parameters, anonymized patient metadata of each session.

## Research involving human participants, their data, or biological material

Policy information about studies with human participants or human data. See also policy information about sex, gender (identity/presentation), and sexual orientation and race, ethnicity and racism.

| | |
|---|---|
| Reporting on sex and gender | The methods section contains the self-reported gender for all patients. |
| Reporting on race, ethnicity, or other socially relevant groupings | We do not use nor report socially relevant categorization variables in our study. |
| Population characteristics | We studied a group of 36 patients (21 females, 15 males) with an average age of 40.5 ± 13.8 years. All patients were diagnosed with pharmacologically-intractable epilepsy. Supplementary table S5 provides information about each patient. |
| Recruitment | Patients undergoing invasive electrophysiological recording for clinical purposes were recruited and consented to participate in this research study. Patients who were capable of and willing to participate in the task were recruited. All patients who spoke english and had a sufficient level of cognitive function were offered participation. Potential biases include subjects who do not speak english and low cognitive function. |
| Ethics oversight | The study was approved by the institutional review boards of Cedars-Sinai Medical Center, Toronto Western Hospital, and John's Hopkins School of Medicine. Patients provided informed consent. |

Note that full information on the approval of the study protocol must also be provided in the manuscript.

# Field-specific reporting

Please select the one below that is the best fit for your research. If you are not sure, read the appropriate sections before making your selection.

☒ Life sciences  ☐ Behavioural & social sciences  ☐ Ecological, evolutionary & environmental sciences

For a reference copy of the document with all sections, see nature.com/documents/nr-reporting-summary-flat.pdf

# Life sciences study design

All studies must disclose on these points even when the disclosure is negative.

| | |
|---|---|
| Sample size | Our analysis is based on 1452 neurons recorded from 36 patients. No statistical methods were used to pre-determine sample sizes. |
| Data exclusions | We excluded individual channels and trials that contained epileptic activity, electrical artifacts or movement-related electrical noise. The methods section contains detailed descriptions of the criteria used to exclude these data. These exclusion criteria were not pre-established but are commonly used. |
| Replication | The analyses were performed at the single neuron and channel level. The effects reported in the study were consistent and replicated across 36 subjects. |
| Randomization | Our design is a within-subject analysis: all the patients were in the same analysis set and had all types of trials. We performed permutation testing where appropriate to ensure statistical validity of our results. |
| Blinding | Patients were not aware of the goals of the study. There was no subjective measurement or decision that the investigator needed to make during the experiment. All the data are collected and analyzed off-line. Data collection and analysis were not performed blind to the conditions of the experiments as conditional information is required for further analyses. |

# Reporting for specific materials, systems and methods

We require information from authors about some types of materials, experimental systems and methods used in many studies. Here, indicate whether each material, system or method listed is relevant to your study. If you are not sure if a list item applies to your research, read the appropriate section before selecting a response.

## Materials & experimental systems

| n/a | Involved in the study |
|-----|----------------------|
| ☒ | ☐ Antibodies |
| ☒ | ☐ Eukaryotic cell lines |
| ☒ | ☐ Palaeontology and archaeology |
| ☒ | ☐ Animals and other organisms |
| ☒ | ☐ Clinical data |
| ☒ | ☐ Dual use research of concern |
| ☒ | ☐ Plants |

## Methods

| n/a | Involved in the study |
|-----|----------------------|
| ☒ | ☐ ChIP-seq |
| ☒ | ☐ Flow cytometry |
| ☒ | ☐ MRI-based neuroimaging |

