## [Peer Review File · Nature]

Manuscript Title: Control of working memory by phase amplitude coupling of human hippocampal neurons

Reviewer Comments & Author Rebuttals

Reviewer Reports on the Initial Version:

Referee expertise:

Referee #1: (neuroscience) intracranial recordings

Referee #2: (neuroscience) memory, electrophysiology

Referee #3: (neuroscience) hippocampal function

Referees' comments:

Referee #1 (Remarks to the Author):

Here, the authors provide an extraordinarily well-written and timely paper on the role of prefrontal-hippocampal coupling in working memory. The authors recorded single neural activities and local field potentials from multiple brain areas in humans and conducted a systematic investigation of these responses using a set of well-designed tasks and analyses. The authors provide evidence that working memory load relates to hippocampal theta-gamma PAC. They also describe the single neural activities underlying this PAC and demonstrate an independent relation between the activities of PAC entrained neurons and that of content/category selective neurons. Finally, they demonstrate that PAC neurons facilitate the decoding of the content held in working memory, together providing a comprehensive explanation of how prefrontal-hippocampal coupling affects working memory performance.

Overall, this paper significantly advances our understanding of working memory in humans. It also provides a strong complement to prior observations from animal models and builds on prior foundational work by the group (e.g., Science 2022). I believe that this work will be highly influential in the field.

Following are suggestions that could help further strengthen the paper.

1. While a major component of the author's findings relies on the differences in memory load, the effects did not appear to be particularly striking. For example, the authors attributed the low LFP PAC in load 3 to higher bits of information in load 3. In Figure 2e, though, the authors indicated that the gamma event count in load 3 was only a little bit higher than load 1 (~ 75 v.s. 72 counts). In addition, in Figure 2f, the duration between load 3 and 1 only differed by $\sim \pm 2$ ms, which is one order of magnitude lower than the duration of gamma band (80 Hz ~ 13 ms). I suspect other factors may contribute to the low LFP PAC in load 3: For instance, load 3 trials consist of 3 images with a duration of 2 seconds each, making them much longer trials than load 1. It would be helpful to address these more directly in the paper (unless I misunderstood the analysis). It may also be helpful to add a control experiment involving modified load 1 trials by duplicating the same image 3 times to more closely match to the duration of load 3 trials. These would allow testing whether high PAC in load 1 is related to the duration of trials. Further, in Figure 5b, the authors showed that PAC neurons in hippocampus synchronized with theta LFP in vmPFC are higher in load 3. If there is indeed more information carried in load 3, should we expect

to see higher PAC neurons in load 1 as well?

2. The authors suggested that the PAC neurons facilitate the WM content decoding by changing the geometry of the feature space but appeared to provide relatively little demonstration of this when describing the geometry (e.g., in Figure 6 and section starting in line 416). In particular, the decoding results in Figure 6d could alternatively be explained as PAC neurons encoding the residual of the decoding from category neurons. For example, suppose there are 5 observations with category [1,1,2,2,2], and a category neuron has firing rates [0,0,1,1,0]. Then, if another neuron has a firing rate of [0,0,0,0,1], this neuron will increase the decoding accuracy. If the authors suggest that the geometry changes towards better decodability, it would be helpful for the authors to provide further evidence/illustration of the specific manifold features that support it. Finally, it wasn't clear to me whether the relation between neural activity/PAC and behavioral performance was specific. For example, was there evidence of non-specific global changes in prefrontal activity on incorrect trials to suggest more generalized state changes?

3. It was a little difficult to follow how the PAC LFP/neurons precisely relate to the category neurons in hippocampus. I understand each result comes from different tests and emphasize different aspects. It's would be helpful though to provide some added discussion addressing the various relations between PAC and category neurons. Following are a few examples:

- a. Figure 3f: spikes of category neurons significantly phase locked to gamma LFP.
- b. Figure 6e: PAC and category single neuron correlation depends on reaction time and load.
- c. Line 280, 320: category neurons did not significantly overlap with PAC neurons.
- d. Line 440: Significant positive single-trial co-fluctuations of spike counts among pairs of category neurons and PAC neurons.

4. It would be important to confirm that the correlations between the neurons do not originate from common sensory inputs (e.g., Figure 6b; which is labelled as d in line 845). In particular, it would be helpful to test whether there were correlations between PAC/category neurons and neurons that were not responding to any of the features examined. On a related note, for Figure 6c which aimed to account for noise correlations, it was unclear why only 8 neurons are displayed. How does this compare across different sessions/participants.

5. The authors showed cross-regional SFC between PAC neurons and vmPFC LFPs (e.g., on the paragraph starting in line 386). It would be helpful to provide further information on how vmPFC neurons relate to vmPFC LFPs, and how the activity of vmPFC cells relate to hippocampus cell activities. Providing these comparisons could help further confirm the regional selectivity of the effects described above.

6. Finally, the paper (and Supplement) is very long which may make it a bit hard for more casual readers to go through. Shortening some of the methods, results and/or descriptions could help.

Referee #2 (Remarks to the Author):

The authors presents data collected from a large cohort of human iEEG patient performing a WM paradigm (Sternberg task) coupled with BF recordings from the prefrontal and hippocampal regions. The recordings focus on WM maintenance phases. They identify neurons whose FR is sensitive to phase and gamma power information (PAC neurons) and explore the properties of these neurons. This is an innovative idea, and they link such neurons to theories of WM activity. The authors use a unique and powerful dataset to address these questions. The size of the dataset means that they should present some subject—level analyses to complement their findings, ie using MEM or at minimum showing that subject—level effects persist for their key findings related to PAC neurons that drive their conclusions. The authors should be congratulated for developing

this dataset and considering a unique perspective from which to analyze the data (neurons sensitive to both phase and gamma power information).

Methodologically they filter for individual recording channels that exhibit a (relatively modest) significant PAC above chance. They focus the subsequent analysis on these channels. A significant percentage of the channels in the hippocampus (of the filtered electrodes) exhibit elevated PAC for the lower load condition, which to me is quite surprising. Was it true in the frontal cortex, perhaps for channels that show PAC even if the overall group of these electrodes do not. The authors analyze this finding by looking for differences in duration of gamma oscillations between load conditions. They test this in the PAC channels, but I don't know why this analysis should be restricted only to the significant PAC channels, as longer bouts of gamma should occur independent of the presence of PAC and if it occurs would actually reduce the likelihood of observing PAC in their filtering step/analysis as this was done across both load conditions. The authors invoke the models of Lisman here, but there is no analysis of phase organization of item related activity (as estimated here in gamma bouts, for example). They also do not explore further, perhaps more compelling alternatives related to differences in preferred frequency for PAC depending on load conditions (in terms of either theta or gamma), as suggested by previous work identifying different preferred frequencies for PAC across the cortex in mnemonic processing.

The control analysis for oscillatory effects impact on PAC is a necessary addition, but authors should consider more sophisticated methods such as that proposed by Vaz (NeuroImage). I would also be curious about the authors' opinion on the preferred frequency for coupling during WM in the cortex vs hippocampus, as other examinations of hipp—cortical PAC consider both lower and higher theta frequencies for coupling (Wang 2021 Hippocampus). Perhaps using more refined frequency ranges would improve PAC detection in the frontal cortex, which seems highly relevant for a WM analysis.

The authors then analyze category selective neurons. This was done only for the MTL? They report elevated firing for category selective neurons during WM maintenance. The plots in 3 b may be influenced by the outliers. Results hold when excluding these?

Please clarify in Results whether the high versus low load conditions included two of the same category for the elevated firing condition. This is evident in the figures but text is a little ambiguous. The load effect should be a direct statistical comparison between load 1 and load 3 and not the reporting of a significant effect for one and not the other, although given the results I think this is probably still significant.

An SFC analysis revealed that (during maintenance I think?) SFC was higher for preferred as compared to non preferred trials, although the firing rate differences during this period were quite modest. I don't understand why this was only done for PAC channels? It doesn't have anything to do with PAC per se. The authors tested whether SFC was correlated with PAC magnitude. This analysis was weakly significant, although if there is no theta SFC for the category specific neurons the implications of this finding are a little more ambiguous. Perhaps this just motivates the subsequent analysis of identification of PAC neurons.

The authors then seek to identify PAC neurons using a (circular/linear?) GLM that included phase terms and gamma power terms to predict FR. Some of these neurons overlapped with category selective neurons. Their activity does not distinguish load conditions, although they do predict successful trials, although there were very few incorrect trials as performance was near ceiling. The SFC analysis could be probably be moved to supplemental material.

The novel finding is that these PAC neurons but not the category neurons exhibit cross regional effects with PFC as a function of WM load. The authors should directly compare these two types of neurons rather than reporting significant effects for one and not the other. This occurred for mPFC but not other frontal regions. What was the correction for multiple comparisons across all the regions and bands?

The authors report firing rate correlations between PAC and category neurons. The authors excluded neurons that were better fit by category information or category models for this analysis? They should add a control looking at correlations with non PAC neurons, as described below. They do include a control with shuffled trial labels.

The authors then go on to propose that PAC neurons support memory information by improving decodability of category even though they are not sensitive to category information per se. Were the overlapping (category) neurons excluded from this analysis? They find that decoding ability is improved with PAC neurons. Do PAC neurons improve decoding ability better than other non PAC non category neurons? Perhaps this was included and I missed it, since they do this for the RT correlations.

The authors link their findings with proposals related to cognitive control and frontal control of WM—related activity in the MTL. This is really the core, novel result. In the context of this interpretation, did the authors test for interactions between category specific frontal lobe neurons and the MTL neurons, or PAC neurons in the MTL and other task sensitive neurons in the frontal cortex? The authors focus on the MTL after showing that PAC occurs at the LFP level in the MTL, but why were frontal neurons excluded from this subsequent analysis? Access to such populations is what differentiates this dataset from others and allows the authors to link with high impact work done in NHP that establishes key predictions for their analysis (see below). Related to this, when identifying PAC neurons, it seems that the theories the authors use to motivate this work would predict that phase information measured in the frontal cortex and not the MTL should be used in the models proposed for identifying PAC neurons. Wouldn't this be a more relevant analysis than local phase information in the MTL? Related to this, is focusing on theta appropriate? The key prediction from NHP WM data seems to be that item—relevant information being represented shifts from beta/alpha phase locking to gamma phase locking (see recent review from the Miller lab, summarizing several experiment). The authors do not analyze the data from this perspective.

The authors also ignore the specific phases of spiking for PAC neurons, which in turn seems surprising given invocation of the Lisman models. An obvious prediction that should be included is the preferred phase of spiking for different items in maintenance, which the authors are well positioned to analyze given the identification of category specific neurons.

Newer models of WM suggest that rather than persistent spiking activity present during maintenance, a composite attractor dynamic consisting of alternating beta/alpha vs gamma bursts occurring activation of ensembles linked with specific items. The authors should discuss their findings relative to these views and consider how phase organization of item—related activity in frequency bands outside of theta might affect their findings.

Overall, the authors show unique results with mostly appropriate methods. They have an opportunity to leverage this dataset to connect their findings with impactful models developed from NHP experiments to motivate publication in a journal of such stature.

Referee #3 (Remarks to the Author):

The paper presents a study focused on understanding the role of Phase-Amplitude Coupling (PAC) neurons in working memory (WM) maintenance. The researchers identified PAC neurons whose spiking activity followed the interactions between theta phase and gamma amplitude during the maintenance period of a Sternberg WM task. They found that unlike category neurons, which displayed memoranda-specific persistent activity, the activity of PAC neurons was not related to

WM content per se. Instead, the activity of PAC neurons in the hippocampus was related to the cognitive control processes that enable efficient and accurate maintenance of WM.

The researchers suggest that PAC neurons play a crucial role in cognitive control and shape WM fidelity through noise correlations with memoranda-selective persistently active neurons. This PAC-mediated interareal interaction might serve as a general mechanism for top-down control to influence bottom-up processes.

The paper aligns with previous studies showing that activity in the high gamma (70-140 Hz) frequency range reflects processing and WM maintenance of sensory information. However, it provides new insights into the role of PAC neurons in cognitive control and WM maintenance.

The paper presents compelling evidence for the role of PAC neurons in cognitive control and WM maintenance. It provides a new perspective on the neural mechanisms underlying WM maintenance and opens up new avenues for future research.

Minor point: The phrase "nuisance factors" suggests that the authors know these factors may affect the result, but they are things to be diminished or maneuvered around. This may not be the impression that the authors are trying to convey. Simply removing "nuisance" resolves this. Otherwise, it sounds like platitudes.

Does Figure 5 imply that long-range coupling in the human MTL is primarily conducted through low frequencies (10 Hz and lower)? The low values for frequencies above 10 Hz suggest that there is no coupling. This seems to contradict the author's assertion that high gamma routes information (lines 530-535). How do the authors reconcile the absence of SFC coupling in Figure 5 with the idea of routing? Or alternatively, do the authors think that their results better align with Mizuskei et al. (2009; PMID: 19874793), where regions of the MTL have significant independence? Please address.

Potential problems:

Among the first manuscripts that applied wavelet to neuroscience data, Tallon-Baudry et al. (1996) describe a trade-off between time and frequency in terms of resolution. For instance, the low-frequency wavelet would provide a precise realization of the frequency of the event with a reduced resolution of when the event occurred. However, as the wavelet narrows, it loses frequency resolution to obtain temporal precision. From the information provided, it is challenging to determine the parameters that went into the wavelet (often, descriptions provide the Gaussian width around a central frequency). However, the 5 Hz spacing between 70-140 Hz or the 40 log spaced wavelets between 2-150 Hz, raise the specter that there may be a great deal of redundant capture across wavelets (e.g., multiple wavelets may overlap with 100 Hz, resulting in an overly convolved representation). Given the importance of the paper, the authors should provide benchmark tests of their decomposition in the supplement:

- To assess the quality of the wavelet decomposition, the authors should compare A) the original signal with B) the signal obtained after applying the CWT, and then the iCWT. The difference between the original signal and the reconstructed signal (i.e., the residuals) can provide valuable insights into the performance of the wavelet transform. If the residuals are small, this suggests that the CWT and iCWT are accurately capturing and reconstructing the key features of the signal. On the other hand, large residuals might indicate that important information is being lost or distorted. It's worth noting that the residuals might not be uniformly distributed across the signal. For example, the CWT might accurately capture the signal's behavior at certain times or frequencies but not others. Therefore, it could be useful to examine the residuals as a function of time or frequency to see if there are any patterns. However, keep in mind that even if the residuals are small, this doesn't necessarily mean that the CWT and iCWT are perfect. It's possible that the wavelet transform could introduce artifacts or distortions that aren't apparent when looking at the residuals alone.

- Concatenate all the scaled and translated versions of the mother wavelet into a single time series and then apply the wavelet transform to that time series. This will create a sort of "wavelet spectrogram" that would allow you to visualize how the frequency content of the wavelet changes with scale (which is related to frequency) and position (which is related to time). This could provide valuable insights into the temporal and spectral characteristics of the wavelet. For example, you could see how the wavelet's frequency content changes with scale, or how well the wavelet localizes different frequencies in time. This could help you understand why the wavelet transform gives the results it does when applied to your data.

- Please provide an example of a raw Local Field Potential (LFP) trace of 1-2 seconds from a subject, perhaps in Figure 2 or as a supplemental figure, along with a Log-Log power spectral density using a fast-Fourier transform with a temporal window of support of 1 second or more. This would provide insights into the general shape of power spectra. Also, does the $1/f$ slope of this power spectra change as a function of memory performance/load?

I am uncomfortable with using channels or cells as samples as these describe aspects of the data that cannot be considered independent. How the author conducted their statistics makes it vulnerable to false positives. Using a sample size of subjects would be appropriate (Aarts et al., 2014). Moreover, in some conditions, the distributions do not appear to be parametric (e.g., Figure 2b), suggesting that a t-test may be inappropriate. Creating shuffle distributions seems artificially stacked in the author's favor to find significance. I am not certain why this practice is done, but it assumes that the null distribution describes the situation "what if the brain is completely random without any correlation"? Creating a shuffled distribution allows the authors to find statistical significance in Figure 6, when on a biological level, a correlation of 0.02 offers that knowing one variable has little predictive value on the other. Therefore, touting the positive correlation (lines 440-445) seems like a gross misrepresentation of what is actually occurring. This analysis, again, was conducted on cell pairs that seem overpowered, and the distributions may not be normal. Statistical issues seem to persist throughout the manuscript (e.g., Figure 3).

The use of a z-score surrounding the Modulation Index is also puzzling. Usually, the raw values of the modulation indices are depicted (perhaps the authors may wish to include this?). Providing the z-score of the values relative to a surrogate distribution may artificially inflate the statistical power, resulting in significance when the values may be small, leaning towards biologically insignificant.

Author Rebuttals to Initial Comments:

Reply to reviewers for Daume et al. submission 2023-05-07411A

Color code: Original (black), **Our reply (blue)**, **Edits in revised manuscript (bold)**

We thank the reviewers for thoroughly assessing our manuscript and providing critical and constructive feedback for various parts of the manuscript. We are delighted to see that all three reviewers recognized the importance, novelty, and rigor of our findings. We greatly appreciate the many suggestions offered by reviewers, which we took to heart and implemented. The resulting changes have substantially improved the manuscript. Based on the reviewers' suggestions, we have made many changes to the manuscript, revised our figures, and added new critical analyses.

Detailed replies are provided below in our step-by-step reply. In summary, major changes include:

- Results for frontal regions are now more prominently presented throughout the manuscript.
- We revised the introduction and discussion to incorporate a clearer account of interactions between PAC and category neurons as well as relations of our work to findings in NHP.
- New Extended Data Fig. 3 provides all suggested new control analyses for theta-gamma PAC in the hippocampus.
- For all main results we now also provide session-level statistics using mixed models.
- Fig. 6 now shows comparisons for noise correlations with non-PAC cells as well as new analysis on the manifold feature that describes the geometrical changes that result from PAC-cell related noise correlations.
- New Extended Data Fig. 1 now includes benchmark tests on our wavelets, and we revised our description of those in the methods section.

In addition to what was requested, we would also like to highlight that we have finished curating the dataset and made it available for reviewers to inspect in the same format that will be released publicly upon publication of the paper. We converted all the data needed to reproduce the results of this paper into a single Neural Data Without Borders (NWB) file for each experimental session. This file contains the sorted neurons (spike times), the raw local field potentials, the stimuli shown to subjects, and the behavior in the standardized NWB format. We are also making available example code that shows how to utilize the files. The NWB files are available at the reviewer-only link <https://figshare.com/s/cc0f8bd03ff45fe70200> (these files will be uploaded to the public DANDI archive upon acceptance), and the code is available at <https://github.com/rutishauserlab/SBCAT-release-NWB>.

Point-by-Point reply

Referee #1

Here, the authors provide an extraordinarily well-written and timely paper on the role of prefrontal-hippocampal coupling in working memory. The authors recorded single neural activities and local field potentials from multiple brain areas in humans and conducted a systematic investigation of these responses using a set of well-designed tasks and analyses. The authors provide evidence that working memory load relates to hippocampal theta-gamma PAC. They also describe the single neural activities underlying this PAC and demonstrate an independent relation between the activities of PAC entrained neurons and that of content/category selective neurons. Finally, they demonstrate that PAC neurons facilitate the decoding of the content held in working memory, together providing a comprehensive explanation of how prefrontal-hippocampal coupling affects working memory performance.

Overall, this paper significantly advances our understanding of working memory in humans. It also provides a strong complement to prior observations from animal models and builds on prior foundational work by the group (e.g., Science 2022). I believe that this work will be highly influential in the field.

We thank reviewer 1 for the kind words. We agree, we believe that our work provides new important insights into how brain areas and cell populations interact during WM maintenance. With the new geometry analysis added that the reviewer suggested, we feel that the overall paper got even stronger and now provides even deeper insights into how the interactions between PAC and category cells support WM maintenance. Below, we respond to each separate point raised by the reviewer.

Following are suggestions that could help further strengthen the paper.

1. While a major component of the author's findings relies on the differences in memory load, the effects did not appear to be particularly striking. For example, the authors attributed the low LFP PAC in load 3 to higher bits of information in load 3. In Figure 2e, though, the authors indicated that the gamma event count in load 3 was only a little bit higher than load 1 (~ 75 v.s. 72 counts). In addition, in Figure 2f, the duration between load 3 and 1 only differed by ~ +/- 2 ms, which is one order of magnitude lower than the duration of gamma band (80 Hz ~ 13 ms). I suspect other factors may contribute to the low LFP PAC in load 3: For instance, load 3 trials consist of 3 images with a duration of 2 seconds each, making them much longer trials than load 1. It would be helpful to address these more directly in the paper (unless I misunderstood the analysis). It may also be helpful to add a control experiment involving modified load 1 trials by duplicating the same image 3 times to more closely match to the duration of load 3 trials. These would allow testing whether high PAC in load 1 is related to the duration of trials. Further, in Figure 5b, the authors showed that PAC neurons in hippocampus synchronized with theta LFP in vmPFC are higher in load 3. If there is indeed more information carried in load 3, should we expect to see higher PAC neurons in load 1 as well?

The reviewer correctly points out that differences due to memory load in behavior, PAC, firing rates, local SFC, and inter-areal SFC are key features of our results. The reviewer, however, points out that parts of our effects - in particular, the gamma event count as well as the duration - might be explained by other factors of the task such as the trial length. In the below, we outline our thoughts on how we resolved this question.

First, we are afraid that the trial length question is likely a misunderstanding. The reviewer asks whether the observed differences between load 1 and 3 might be explained by differences in the length of the analysis window. However, there is no such difference: all analysis of load effects (including the PAC analysis) is based on neural data acquired during the delay period of the task (0 – 2.5 s after delay onset), which was the same for both loads. Therefore, in both load conditions the analysis window had the same length. The reviewer is correct that load 3 trials are longer than load 1 trials, but this length difference is in the encoding part of the trial which is not being analyzed for PAC. We note that the fact that the encoding period is longer and contains more items is what makes load 3 trials require more cognitive control, i.e., the phenomenon we are investigating. For these reasons, we are not concerned that our PAC results are confounded by the length of the analysis window (which is the same). We revised our methods and results sections (lines 126-129, 1079-1083) to clarify this point and now mention explicitly that the increased cognitive control demands of load 3 trials are by design (lines 1079-1083). For background (for the reviewer), we note that when originally designing our experiment we had carefully considered the variant of the load 1 trials that the reviewer suggests. However, we concluded that showing the load 1 item three times would decrease cognitive control demands too much and cause other unwanted effects such as repetition suppression and enhanced memory strength due to repeated presentation of the same stimulus in load 1 trials. Differences in firing rates between the load conditions for persistently active neurons could then be attributed to these other factors rather than cognitive load, which we seek to study here. We therefore decided to conduct the experiment in the way it is reported in this manuscript.

Second, the reviewer questions the premise of the analysis we offered for explaining why PAC is lower in load 3 compared to load 1 trials and criticizes that the observed differences in gamma duration are a magnitude smaller than one cycle length of an 80 Hz oscillations. We performed additional simulations that show that changes in gamma event duration of the kind we found are sufficient to produce the changes in PAC effect size that we have observed (please see reviewer figure 1.1 below). However, we have decided to remove this analysis entirely from the paper because it is not core to our principal result. While it is a very interesting question of what field potential changes give rise to PAC changes, our paper is focused on examining a different question: the single-neuron correlate of PAC. We therefore felt that this aspect of our paper was distracting from our main result. We therefore removed figure panels 1e and 1f of the original manuscript and are only showing the simulation mentioned as reviewer figure 1.1 below.

To better illustrate why PAC is lower in load 3 than load 1 trials, we added a new plot that shows the distribution of gamma power as a function of theta phase for an example channel together with the resulting values of PAC (Fig. 2d; quantified across all channels in Extended Data Fig. 3a). These plots support the idea that gamma is more uniformly distributed across theta when more items are stored in memory (see Heusser et al. 2016). To further quantify this observation across the population more directly, we computed kappa (the variance of a circular distribution) – which measures the concentration of gamma amplitude over theta phase – and observe that kappa is smaller in load 3 than load 1, which speaks to the idea of a wider distribution with higher memory load. We added this new analysis to Extended Data Fig. 3a.

Third, regarding the stronger SFC effect in load 3 for PAC neurons, our interpretation of this result is that PAC neurons are under tighter frontal cognitive control when needs for cognitive control are high. Precise spike timing leads to more efficient communication between brain areas (see Fries 2005, 2015), which in turn might lead to enhanced local coordination between PAC and content-tuned neurons to stabilize the population code during WM maintenance. This interpretation is supported by the fact that inter-areal theta SFC was stronger in fast than in slow RT trials (see Fig. 5f), suggesting a behavioral benefit for stronger inter-areal connectivity during WM maintenance. Thus, we hypothesize that not the number of neurons engaged or their firing rate but the extent of their spike timing coordination within and across areas is what allows better cognitive control. We adjusted our discussion to make our view clearer (lines 703-728).

Reviewer Figure 1.1: (a) We simulated a 4 Hz theta rhythm and added a (non-modulated) 80 Hz gamma rhythm to it (plus normally distributed noise not shown here for simplicity). The gamma rhythm was then multiplied with a gaussian taper centered at each theta peak such that gamma amplitude was 1 at the peak

and around 0 at the trough of theta, i.e., producing theta-gamma PAC. We used two tapers with different temporal widths – a “narrow” and “wide” taper. When multiplying gamma using the wide taper, gamma amplitude had a “wider” distribution across the theta cycle than when using the narrow taper, causing weaker PAC for the wider taper as compared to the narrow taper. (b) The stronger the difference in temporal width between the two tapers (measured as the temporal standard deviation of each taper), the smaller the proportion between the two PAC estimates obtained from using the narrow and the wide taper. We located the difference in temporal width that leads to a reduction of around 15% in PAC (proportion of 0.85), which is what we observed on average between load 1 and load 3 in our data. (c,d) Comodulogram and gamma distribution across all theta bins for (c) the narrow taper, and (d) the wide taper, using a temporal width difference of 12 ms as determined in (b). (e,f) In our simulation, we could not z-transform the data to determine “peak gamma events” and their length at a z-score of 2 in order to replicate the analysis we performed in our original manuscript. This is because in the simulation each gamma event had roughly the same peak amplitude and did not strongly vary in peak amplitude like in real data. Any variation in peak power in the simulation was due to added noise, which was equally added to both signals to avoid biasing the results. Thus, in order to be able to compare the simulation with the real data, we determined the “mean duration” (i.e., length) of all “gamma events” at different amplitudes of gamma. In our recorded data, we measured the gamma duration at a z-scored amplitude of 2, resulting in a mean gamma duration of ~31 ms across both loads. The difference in gamma duration between load 1 and 3 at this point was ~2 ms (see Fig. 1f in the original manuscript). In the simulation, we measured the mean duration of gamma at different amplitudes for (e) the narrow-tapered gamma, and (f) the wide-tapered gamma, and then measured the difference in gamma duration observable at each of those “mean durations”. This enabled us to find the expected differences in gamma duration for the point where the mean gamma duration equaled ~ 30 ms (like in our recorded data; see horizontal lines in (e,f)). At a mean duration of 30 ms, we observed a difference in gamma duration between the wide and the narrow taper of 4 -5 ms (measured separately for both, the wide and the narrow taper). This confirms that a difference in gamma duration of <5 ms causes a reduction in CFC magnitude to the extent that we found in our data. The reason for why this small amount of width change changes CFC to this large extent is that we measured the duration of gamma at a z-score of 2 and thus “at the tip of the iceberg”, where the duration of gamma (as well as the difference) is much shorter than for the full event. This explains why the difference in duration is shorter than expected from the length of a full cycle at 80 Hz. We note that we cannot measure gamma events with lower z-scores because in this situation “real gamma events” cannot be differentiated from noise. The measured duration would thus be grossly inflated, requiring that it be measured at “the tip of the iceberg” as we do here. While this simulation shows that our original conclusion was valid, we nevertheless decided to remove this analysis entirely from the manuscript and provide this simulation as a reviewer figure only. This is because measuring the duration of gamma events was not central to our study and can remain for a separate study with this central aim.

2. The authors suggested that the PAC neurons facilitate the WM content decoding by changing the geometry of the feature space but appeared to provide relatively little demonstration of this when describing the geometry (e.g., in Figure 6 and section starting in line 416). In particular, the decoding results in Figure 6d could alternatively be explained as PAC neurons encoding the residual of the decoding from category neurons. For example, suppose there are 5 observations with category [1,1,2,2,2], and a category neuron has firing rates [0,0,1,1,0]. Then, if another neuron has a firing rate of [0,0,0,0,1], this neuron will increase the decoding accuracy. If the authors suggest that the geometry changes towards better decodability, it would be helpful for the authors to provide further evidence/illustration of the specific manifold features that support it. Finally, it wasn't clear to me whether the relation between neural activity/PAC and behavioral performance was specific. For example, was there evidence of non-specific global changes in prefrontal activity on incorrect trials to suggest more generalized state changes?

We added additional analyses along the lines suggested by the reviewer to provide deeper insight into the specific manifold features that are influenced by noise correlations. These new analyses address all issues raised above as following:

First, we would like to point out that the alternative coding scheme proposed by the reviewer, i.e., that PAC neurons encode the residuals of the decoding of category neurons, cannot explain why PAC neurons enhanced the decoding performance when being added to the decoding ensemble. This is because PAC neurons only enhanced the decoding performance when the data used from each neuron was recorded simultaneously (i.e., with intact noise correlations). If PAC neurons would encode the residuals of category information in the way suggested by the reviewer, they should be enhancing decoding performance even after the trial order is scrambled within each category. This is because within-category scrambling, which is done when removing noise correlations, does not change the mean response differences between categories and would thus also not remove the type of 'residual coding' suggested. This is, however, not what we observed in our data because when we scramble trials within-category, the beneficial effect of adding PAC neurons to the population goes away (see Fig. 6c for "intact" vs "removed"). This therefore excludes the coding scheme suggested by the reviewer as a possibility. We modified the results (lines 499-504) and the methods (lines 1361-1364) to point this out.

Second, we added new analyses to quantify and illustrate how noise correlations due to PAC neurons change the population geometry (Fig. 6f-h). To quantify the geometry, we assessed the angle between the signal axis and the noise axis in the neural state space. We did this in each session separately and then compared the angles between when noise correlations were intact vs. removed. This revealed that the signal-noise angle became significantly larger when noise correlations were present. To motivate this analysis, consider that noise correlations can be information limiting or enhancing depending on the angle between the signal axis and the noise axis of the population response (see Fig. 1g in Panzeri et al. 2022). If the angle between the signal and the noise axis is relatively large to begin with, and if this angle is further

made larger by noise correlations, noise correlations would enhance the information encoded. On the other hand, decreasing the signal-noise angle would limit information in this scenario. We therefore hypothesized that if noise correlations in our recordings are information enhancing, the angle between the signal and the noise axes should (a) be relatively large, and (b) should increase when noise correlations are present compared to when they are absent. Thus, for all sessions that contained at least two simultaneously recorded neurons in the hippocampus, we computed the angles between the signal and the noise axis in each session and compared conditions where noise correlations were intact vs. removed (see Fig. 6g). When noise correlations were intact, we observed an angle of on average ~69 degrees (which is large given that 90 deg is the maximum and 0 deg the minimum). When removing noise correlations from the neuronal population, the angle became significantly smaller. This analysis therefore shows that the manifold feature that is modified by noise correlation is the angle between the signal and noise axis. We added these new results to Fig. 6g. We also added an illustration of the manifold feature noise-signal axis angle in Fig. 6f (based on a simulation).

Third, to further investigate the role that PAC neurons play in shaping the geometry of the population response, we projected the population response onto the signal axis and determined the standard deviation of the data separately for each cluster (Averbeck and Lee 2006). We did so both with intact and removed noise correlations, as well as with and without PAC neurons present in the population. If the angle between the noise and the signal axis becomes smaller, the variance of the projected data along the signal axis gets larger. We found that removing PAC neurons from the ensembles significantly increased the standard deviation of the projected data (significant main effect of *ensemble*). Moreover, the standard deviation of the projected values was larger when noise correlations were removed, but only when PAC neurons were part of the ensemble. We did not observe a significant difference in the variance of the projected data between intact and removed noise correlations when PAC neurons were removed. These findings suggest that specifically those noise correlations that were introduced by PAC neurons alter the geometry of the population code in a way that can enhance the decodability of the memory content by increasing the angle between the noise and the signal axis in the population response. We added this new result to Fig. 6h.

Finally, the reviewer asked whether there were any non-specific global changes in firing rates of prefrontal neurons on incorrect trials. We tested firing rates between correct and incorrect trials across all recorded neurons in each of the three frontal regions. However, we do not observe such global state changes in any of the areas. We added these results to Extended Data Fig. 8h.

3. It was a little difficult to follow how the PAC LFP/neurons precisely relate to the category neurons in hippocampus. I understand each result comes from different tests and emphasize different aspects. It's would be helpful though to provide some added discussion addressing the various relations between PAC and category neurons. Following are a few examples:

- a. Figure 3f: spikes of category neurons significantly phase locked to gamma LFP.
- b. Figure 6e: PAC and category single neuron correlation depends on reaction time and load.
- c. Line 280, 320: category neurons did not significantly overlap with PAC neurons.
- d. Line 440: Significant positive single-trial co-fluctuations of spike counts among pairs of category neurons and PAC neurons.

In the revised version of the manuscript, we have updated the introduction as well as the discussion at several instances and hope this makes it clearer how we think the PAC neurons relate to category neurons in the hippocampus. In summary, our results suggest that interactions between the hippocampus and frontal regions were mediated by neurons within the hippocampus whose activity was a function of the interaction of theta phase and gamma power. These are what we call “PAC neurons”. Surprisingly, we find that PAC neurons are a separate population from persistently active category neurons, thereby not supporting the hypothesis that persistently active neurons directly coordinate their spike time with the frontal lobe as a way of control. Rather, we find that it is the spike timing of PAC cells that is coordinated with frontal regions. This way of engaging cognitive control nevertheless enhanced the fidelity of the representation of WM content because PAC neurons had positive noise correlations with persistently active category cells that were structured such that the population-level representation improved when PAC cells were present. Lastly, these correlations were observable in two ways: at the population level as co-fluctuating firing rates of pairs of neurons (noise correlations), and as relationships between firing rates and local gamma (through local SFC and power correlations).

4. It would be important to confirm that the correlations between the neurons do not originate from common sensory inputs (e.g., Figure 6b; which is labelled as d in line 845). In particular, it would be helpful to test whether there were correlations between PAC/category neurons and neurons that were not responding to any of the features examined. On a related note, for Figure 6c which aimed to account for noise correlations, it was unclear why only 8 neurons are displayed. How does this compare across different sessions/participants.

We performed additional analysis to examine that the correlations we examined do not originate from common sensory input.

First, we note that the null distribution shown in the revised Fig. 6a shows the observed correlations relative to shuffling trials within categories (preferred/non-preferred) and load groups (referred to as “conditions” this was not the case in the original submission). Computed this way, if they are present, signal correlations remain in the null distribution with only noise correlations removed (Cohen and Kohn 2011).

We note that the effect remains essentially identical to Fig. 6b in the original submission, with the null distribution (which is the signal correlations alone) not significantly different from 0, indicating there are little signal correlations. If the pair-wise correlations between PAC neurons and category neurons were driven by common sensory input, the correlations should be similarly strong even after shuffling trials within condition. The shuffling procedure, however, almost completely removed correlations among pairs of neurons, suggesting that the pair-wise correlations were mainly driven by noise correlations. We would further like to point out that we computed correlations among pairs of neurons during the *delay period* where no stimulus was presented on the screen. The activity of neurons in this period could not be explained by strong bottom-up, stimulus-related inputs.

Moreover, we emphasize that noise correlations are a population-level phenomenon, meaning that we would not expect some pairs of neurons to have correlations and others not. We would, however, expect that noise correlations between some pairs are of more functional relevance than others (Kohn et al. 2015). We find both of these results. First, noise correlations strength of pairs of category neurons and non-PAC non-category neurons were not significantly different from pairs of PAC and category neurons (see Reviewer Fig. 1.4 below; see also reviewer 2 point 8). This further indicates that the correlations between PAC and category neurons were not specifically driven by common sensory signals because they are also present in non-PAC non-category neurons that have no tuning to sensory signals. Second, the functional relevance of noise correlations can vary among groups of neurons, depending on their relation between signal and noise correlations (as pointed out in our response to point 2). We find that noise correlations for pairs of category and PAC neurons had specific functional importance for information coding and behavior because their strength varied between slow and fast trials, which was not the case for pairs of category and randomly chosen non-PAC neurons (see Fig 6i,h, and our response to reviewer 2 on point 8 and 9). We have added this reasoning to our discussion (lines 781-785).

Second, to clarify Fig. 6c in the original manuscript, this figure is showing the decoding performance in an example session for all hippocampal neurons until the maximal decoding performance was reached (also see Leavitt et al., 2017). The remaining neurons did not further contribute to category decoding and were therefore not further considered. We realized that this procedure might have biased the analysis to only a subset of neurons that contributed to category neurons and have therefore changed this in the revised manuscript. We are now, in addition, comparing the maximal difference in decoding levels between intact and removed noise correlations among all neurons in an ensemble, and after removing all PAC neurons from the analysis (see Fig. 6d,e and Extended Data Fig. 9k). We believe that this is a more unbiased way to assess the influence of PAC neurons and their noise correlations on decoding levels. Related to the question of how this result compares across sessions/subjects, note that each dot in one of the swarm plots shown in Fig. 6d,e is an individual session, thereby showing the result across all sessions.

Reviewer Figure 1.4: Mean correlations for CAT-PAC pairs (162 pairs; cyan line) compared to 10,000 iterations of selecting the same number of random pairs for non-PAC non-category to category neuron pairs.

5. The authors showed cross-regional SFC between PAC neurons and vmPFC LFPs (e.g., on the paragraph starting in line 386). It would be helpful to provide further information on how vmPFC neurons relate to vmPFC LFPs, and how the activity of vmPFC cells relate to hippocampus cell activities. Providing these comparisons could help further confirm the regional selectivity of the effects described above.

We added new analysis of within-region SFC across all frequencies for all vmPFC neurons as a function of memory load (in Extended Data Fig. 8f). This analysis shows no differences in local SFC within the vmPFC as a function of load, confirming the specificity of the load effect in cross-regional SFC.

We also examined the relation of the activity of vmPFC cells to hippocampal LFP. Similarly to local SFC within the vmPFC, we did not observe significant differences between the load conditions in any of the frequencies (see Extended Data Fig. 8g). Together, this new analysis of the vmPFC data supports the regional as well as neuronal specificity of the observed effects between hippocampal PAC neurons and LFPs recorded in vmPFC.

6. Finally, the paper (and Supplement) is very long which may make it a bit hard for more casual readers to go through. Shortening some of the methods, results and/or descriptions could help.

In an effort to shorten the results, as outlined in our response to point 1, we removed our analysis on gamma duration and PAC load differences from the manuscript. We also moved parts of figures to the supplements to make space for new analysis related to points raised by the reviewers. For now, we have otherwise chosen to keep the manuscript text assembled as whole. Upon acceptance, we will move select

parts of the text to the supplement to accommodate space constraints. However, we felt that it will be easier to review the paper if the main text remains as it is now.

Referee #2

1. The authors presents data collected from a large cohort of human iEEG patient performing a WM paradigm (Sternberg task) coupled with BF recordings from the prefrontal and hippocampal regions. The recordings focus on WM maintenance phases. They identify neurons whose FR is sensitive to phase and gamma power information (PAC neurons) and explore the properties of these neurons. This is an innovative idea, and they link such neurons to theories of WM activity. The authors use a unique and powerful dataset to address these questions. The size of the dataset means that they should present some subject—level analyses to complement their findings, ie using MEM or at minimum showing that subject—level effects persist for their key findings related to PAC neurons that drive their conclusions. The authors should be congratulated for developing this dataset and considering a unique perspective from which to analyze the data (neurons sensitive to both phase and gamma power information).

We thank reviewer 2 for their thorough consideration of our work. We are delighted to see that the reviewer recognizes the importance and novelty of our identification of PAC neurons and their role WM maintenance. The reviewer raised important points for which we are grateful and which we fully addressed as outlined below. In the revised manuscript, we now present results for activity recorded in frontal cortex more prominently. For easier referability, we numbered each point raised by reviewer 2. Please find our responses below.

As requested, in the revised version of the manuscript, we now present session-level effects either using multilevel, random-effects models as well as repeating key analysis after averaging within each session (Aarts et al. 2014) for the main effects reported in this study. This new analysis (see Extended Fig. 5b,g, Extended Fig. 8b, Extended Fig. 9i) shows that all effects persist on the session level (we note that results for theta-gamma PAC were already provided at the per-session level in the original submission, now Extended Fig. 2b). We are therefore confident that our results were not driven by only a few sessions or by elevated false positive rates due to large sample sizes.

2. Methodologically they filter for individual recording channels that exhibit a (relatively modest) significant PAC above chance. They focus the subsequent analysis on these channels. A significant percentage of the channels in the hippocampus (of the filtered electrodes) exhibit elevated PAC for the lower load condition, which to me is quite surprising. Was it true in the frontal cortex, perhaps for channels that show PAC even if the overall group of these electrodes do not. The authors analyze this finding by looking for differences in duration of gamma oscillations between load conditions. They test this in the PAC channels, but I don't know why this analysis should be restricted only to the significant PAC channels, as longer bouts of gamma should occur independent of the presence of PAC and if it occurs would actually reduce the

likelihood of observing PAC in their filtering step/analysis as this was done across both load conditions. The authors invoke the models of Lisman here, but there is no analysis of phase organization of item related activity (as estimated here in gamma bouts, for example). They also do not explore further, perhaps more compelling alternatives related to differences in preferred frequency for PAC depending on load conditions (in terms of either theta or gamma), as suggested by previous work identifying different preferred frequencies for PAC across the cortex in mnemonic processing.

As the reviewer recognizes, a key observation that motivates our paper is that theta-gamma PAC in the hippocampus is stronger for low (load1) compared to high (load 3) memory loads. The reviewer asks whether PAC is similarly modulated in frontal cortex. First, before answering this important question, we would like to clarify what might otherwise be a misunderstanding: we do not select for channels at which PAC was significantly different between the loads. Rather, we only select all channels for which theta-gamma PAC is significantly larger than expected by chance across both load conditions. For this subset of channels, we then examine whether they as a group differ in theta-gamma PAC strength between the two memory loads. This revealed differences between loads only in hippocampus, but not the other areas examined (amygdala and the three frontal areas dACC, pre-SMA, and vmPFC). We revised Fig. 2 by removing the percentages below the x-axis in panel c and added panel 2b to clarify that the percentage of significant PAC channels is not related to the memory load differences.

Second, we understand the reviewer's concern that selecting for significant channels might have obscured load-dependent effects that might be present in frontal areas when testing among the entire group of recording channels, not only significant ones. As requested, we therefore tested PAC between the load conditions across all channels available in each area to determine if load-dependent effects consisted for the overall group of channels within the theta (3-7 Hz) and high-gamma frequency band (70-140 Hz). However, even when using all channels, we only observe significant differences between the load conditions in the hippocampus. No other area showed significant differences in those frequency ranges (see Reviewer Fig. 2.2 below; see also Reviewer Figure 3.5 where we compare PAC between load conditions in each area using raw modulation indices). We also conducted more area- and frequency-specific analyses to determine whether we observe load-dependent effects in other areas or frequencies (see response to point 3 below).

Third, the reviewer asked what the reason is for reduced theta-gamma PAC in load 3 compared to load 1. We had offered analysis on the length of gamma bouts. We further substantiated this analysis with a simulation but have now removed this analysis from the paper because it is not core to our principal result and we were asked to shorten the paper. Please see reply 1 to reviewer 1 for further details.

Fourth, we removed the invocation of the Lisman model. We do not set out to test the core prediction of the Lisman model in our paper (as the reviewer points out), and we should therefore not invoke it when describing our analysis. Indeed, we do not provide an analysis of item-specific phase organization. This

critical prediction of the Lisman model therefore remains to be examined, and we do not do so here. We modified our results and discussion parts accordingly.

Lastly, the reviewer is asking whether the preferred frequencies for theta and gamma might change as a function of memory load. As the reviewer notes, earlier research shows that different gamma bands are coupled to different phases of the underlying theta rhythm (Colgin et al. 2009; Schomburg et al. 2014; Colgin 2016; Fernandez-Ruiz et al. 2017) with different implications on mnemonic processing (Lega et al. 2016). We observed prominent PAC in both high-and lower gamma range (70-140 and 30-55 Hz) and were thus eager to test whether the peak in theta-low gamma PAC was also related to our manipulation of memory load. However, testing between the load conditions did not reveal a significant difference in this frequency combination in any of the studied areas (see Extended Data Fig. 2c). Motivated by the reviewer's question, we performed additional analysis to examine (i) whether preferred frequencies systematically differed as a function of load (Extended Data Fig. 3h) and (ii) whether the preferred theta phase in theta-high gamma PAC differed between the two load conditions in hippocampal PAC channels (see Extended Data Fig. 3g). We revealed no significant differences in preferred frequencies for theta or gamma, nor for the preferred phase of theta as a function of load. This confirms our conclusion that neither the preferred frequencies nor the preferred theta phase differed as a function of memory load, leaving the effect of memory load to be specific to only one specific combination: that of theta-high gamma.

Reviewer Fig. 2.2: Comparison of theta-high gamma PAC between load conditions across all available channels per area. The only significant difference was observed in the hippocampus (p values of permutation tests shown in the figure below each panel).

3. The control analysis for oscillatory effects impact on PAC is a necessary addition, but authors should consider more sophisticated methods such as that proposed by Vaz (NeuroImage). I would also be curious about the authors' opinion on the preferred frequency for coupling during WM in the cortex vs hippocampus,

as other examinations of hipp—cortical PAC consider both lower and higher theta frequencies for coupling (Wang 2021 Hippocampus). Perhaps using more refined frequency ranges would improve PAC detection in the frontal cortex, which seems highly relevant for a WM analysis.

First, we implemented the suggested control analysis following Vaz et al. (Vaz et al. 2017). Using this method, we analyzed the hippocampal PAC channels to assess how many channels show nesting, i.e., gamma oscillations nested into theta cycles, based on the methodology published by Vaz et al. 110 out of 137 PAC channels (80.29%) showed nesting as determined by the Vaz et al method, therefore providing independent confirmation for our selection method. Two exemplar channels are plotted in Extended Data Fig. 3e. We further tested whether the main effect between the two load conditions persists after excluding the remaining ~20% of the channels that do not show nesting (i.e., which were selected as exhibiting PAC by our method but not by that of Vaz et al.) and found that the results are qualitatively similar, i.e., PAC was stronger in load 1 as compared to load 3 (see Extended Data Fig. 3e).

Second, we examined more refined frequency ranges as suggested by the reviewer. To do so, we re-computed PAC in all channels of our dataset using a finer low-frequency resolution, i.e., using steps of 0.5 Hz instead of 2 Hz. The grand average plot using this higher frequency resolution across all recording channels and both conditions is plotted below (Reviewer Fig. 2.3a). Qualitatively, this plot is very similar to our initial analysis plotted in Fig. 2a. In Extended Data Fig. 2a (for a step size of 2 Hz) and below (for 0.5 Hz), we have also plotted comodulograms for each area separately to observe whether interesting patterns of PAC emerge in any of the areas that might have been missed in the grand average plot across all areas. These plots show that PAC is very prominent in MTL structures and much weaker in prefrontal cortex and are qualitatively very similar to our original analysis (we therefore decided to keep our original analysis using a step size of 2 Hz, except for the peak frequency analysis for point 2 as shown in Extended Data Fig. 3h). Pre-SMA as well as vmPFC have somewhat stronger PAC in lower frequencies for phase, but the peak is lower than our chosen cutoff-frequency of 2 Hz. Note that the cutoff of 2 Hz was not arbitrarily chosen but rather was informed by the duration of the available data, which is 2.5 s long maintenance period of the task. This cutoff frequency ensures that a minimum of ~3 cycles were included in each frequency from each single trial. Please note that 2 Hz here refers to the center frequency, meaning that the bandpass filter applied to the signal in fact included frequencies between 1 and 3 Hz for the lowest frequency bin. Choosing a frequency cutoff lower than that could thus include frequency ranges that might consist of only a single cycle during the analysis window. PAC in these very low frequency ranges is vulnerable to non-stationarities (such as phase resets) in the LFP signal and prone to misinterpretations (see Aru et al. 2015; Vaz et al. 2017). To determine whether we observe significant differences between the load conditions in more refined, possibly lower frequency ranges, we analyzed PAC using all combinations of slow (2-5 Hz) or fast theta (5-9 Hz) and low (30-55 Hz) or high gamma (70-140 Hz; similar to (Lega et al. 2016; Wang et al. 2021). This revealed that PAC differences as a function of load were only

seen in the high gamma band in the hippocampus (see Table in Reviewer Fig. 2.3b below as well as supplementary Table S1). Furthermore, in the hippocampus, PAC was significantly different for both fast and slow theta ranges, but stronger for fast. None of the prefrontal areas showed an effect between the two load conditions in any of the frequency combinations (all $p > 0.05$; see Reviewer Fig. 2.3b, supplementary Table S1).

Fig. 2b shows that among the three prefrontal areas, the vmPFC was the only area with somewhat elevated levels with respect to the number of significant PAC channels for theta-high gamma PAC. To have a more detailed look into the effects that drive those PAC channels, below (see Reviewer Fig. 2.3c) we plotted their PAC comodulogram across all frequency combinations for each load condition (like what we plotted for the hippocampus in Fig. 2e). Unlike in the hippocampus, these channels did not show strong PAC in the chosen frequency band (theta-high gamma) but seem to be driven by lower frequencies between 2-5 Hz. However, as mentioned above, even when combining slow theta frequency ranges (2-5 Hz) with any of the two gamma frequency ranges, we did not observe significant differences between the two load conditions (see Reviewer Fig. 2.3b).

Based on these analyses, we posit that task/WM-relevant theta-gamma PAC differences were only present in the hippocampus where theta-gamma PAC was strong and differed as a function of WM load. However, note that we are not saying that theta-gamma PAC in general is not present in frontal areas. Rather, what our results show is that only in the hippocampus does theta-gamma PAC vary as a function of WM load and relates to behavior (see Fig. 2f, which now also shows results for the vmPFC). Indeed, while on a relatively small percentage of channels, we did observe PAC between slower theta and gamma oscillations in prefrontal areas, particularly in vmPFC channels (see Fig. 2b and supplementary Table S1). However, PAC on these channels was not a function of memory load. To make this even clearer to the reader we replaced the panel in Fig. 2 that showed PAC for all frontal regions combined (labelled 'MFC') with showing vmPFC only since this was the only frontal region with a proportion of PAC larger than expected by chance. While not shown in the figure, load comparisons for PAC channels in pre-SMA and dACC are reported in the main text.

The functional relevance of this non-memory load modulated PAC in frontal cortex for WM remains to be addressed in future work and we do not make claims about it. We also note that we investigated only areas within *medial* frontal cortex, and not lateral frontal areas such as DLPFC. It therefore remains an open question whether our findings extend to DLPFC or not. We added these caveats to the discussion (lines 661-670). Nevertheless, we point out that our findings are compatible with those of Johnson et al. (Johnson et al. 2018) who observed theta-gamma PAC in OFC (labeled as vmPFC in our study) as well as lateral PFC in a WM task. Both regions, however, did not show task-related modulations of within-region PAC. Within-region PAC changes related to WM manipulations were only found within the MTL.

Reviewer Figure 2.3. (a) Grand-average and region-specific comodulograms for finer resolution PAC (0.5 Hz steps for the frequency for phase). (b) PAC comparisons between load conditions for all combinations of slow/fast theta and low/high gamma in each area. We added these results to the supplements as Table S1. (c) PAC comodulogram for each load condition for the 40 vmPFC channels that show significant levels of theta-high gamma PAC.

4. The authors then analyze category selective neurons. This was done only for the MTL? They report elevated firing for category selective neurons during WM maintenance. The plots in 3 b may be influenced by the outliers. Results hold when excluding these?

We focused the analysis of category selectivity on the MTL because only neurons in this part of the brain show category selectivity during the maintenance period. As requested, we now added this analysis for all brain areas so that readers can appreciate this remarkable specificity. Extended Data Fig. 4a now includes statistical tests on the number of category neurons in each area. Consistent with previous work (Kamiński et al. 2017, 2020), we find significant numbers of category neurons in hippocampus and amygdala, but not in dACC and pre-SMA. A novel result not previously shown is that we find significant numbers of category neurons also in vmPFC. However, category neurons in vmPFC (selected during encoding) did not remain selective during the delay period of the task. That is, category neurons in the vmPFC were selective for picture category only during the stimulus presentation window but not during the maintenance period. Their activity therefore was not related to WM maintenance. For this reason, we focused on the MTL for our analysis, as this was the only area of the brain that exhibited delay period activity that was informative about working memory content.

With respect to the concern that outliers may drive our results in Fig. 3b, as a control we excluded neurons with baseline normalized firing rates higher than 1.5 in both areas of the MTL. Reviewer Fig. 2.4 below shows that the results persisted even after taking these neurons out.

Reviewer Figure 2.4: Delay period activity of category neurons from MTL compared between preferred vs non-preferred categories after taking neurons that have a normalized FR higher than 1.5 out. Compare to Extended Data Fig. 4b.

5. Please clarify in Results whether the high versus low load conditions included two of the same category for the elevated firing condition. This is evident in the figures but text is a little ambiguous. The load effect should be a direct statistical comparison between load 1 and load 3 and not the reporting of a significant effect for one and not the other, although given the results I think this is probably still significant.

We clarified this point in the manuscript (results, lines 239-242). During encoding, each load 3 trial always contained pictures from three different categories. A given trial thus never consisted of two pictures from the same category to be maintained in WM. That is, when comparing trials between load 1 and 3 for preferred trials, each load condition always contained exactly one item from the preferred category. Also, we confirm that we directly contrast load 1 vs load 3 as the reviewer recommended, rather than compare each load condition separately vs baseline. In our plots, we indicate that we compare the two loads by connecting the data from the two loads with lines in all plots (see, for example, Fig. 3c).

6. An SFC analysis revealed that (during maintenance I think?) SFC was higher for preferred as compared to non preferred trials, although the firing rate differences during this period were quite modest. I don't understand why this was only done for PAC channels? It doesn't have anything to do with PAC per se. The authors tested whether SFC was correlated with PAC magnitude. This analysis was weakly significant,

although if there is no theta SFC for the category specific neurons the implications of this finding are a little more ambiguous. Perhaps this just motivates the subsequent analysis of identification of PAC neurons.

First, we confirm that the reviewer's interpretation is correct that the SFC was computed during the maintenance period. Second, we note that we performed SFC analysis for both PAC and non-PAC channels. This comparison revealed a significant difference in gamma but not theta-band SFC only in PAC but not non-PAC channels (see Extended Data Fig. 5e). Nevertheless, we now added SFC analysis including all channels (see Extended Data Fig 5f). Comparing SFC between preferred and non-preferred trials using all channels revealed qualitatively similar effects as compared to testing PAC channels alone. As the reviewer indicates, it is surprising that there was no content-related difference in theta-band SFC for these cells, indicating that the relationship between persistent activity and PAC cannot be understood by examining local SFC within the hippocampus. Indeed, this finding (as the reviewer suspects) serves to motivate the procedure we developed on identifying PAC cells that Fig. 4 shows. We revised the results (lines 299-309) and introduction (lines 75-82) to motivate our reasoning better.

7. The authors then seek to identify PAC neurons using a (circular/linear?) GLM that included phase terms and gamma power terms to predict FR. Some of these neurons overlapped with category selective neurons. Their activity does not distinguish load conditions, although they do predict successful trials, although there were very few incorrect trials as performance was near ceiling. The SFC analysis could be probably be moved to supplemental material. The novel finding is that these PAC neurons but not the category neurons exhibit cross regional effects with PFC as a function of WM load. The authors should directly compare these two types of neurons rather than reporting significant effects for one and not the other. This occurred for mPFC but not other frontal regions. What was the correction for multiple comparisons across all the regions and bands?

First, we applied sine as well as cosine functions to the theta phases and included both terms in the model. This ensured that we could treat theta phase as a linear instead of a circular variable (Al-Daffaie and Khan 2017). We now make this clearer in our methods section (lines 1279-1280).

Second, we thank the reviewer for suggesting comparing long-range SFC directly between category and PAC neurons. We agree that this is an important control analysis which we now provide. Testing the difference between load 3 and load 1 trials in the theta band directly between the two groups of neuron-to-channel connections revealed that PAC neurons showed a significantly stronger difference between the load conditions than the category neurons (Extended Data Fig. 8c). We further moved the within-region SFC analysis for PAC neurons to the supplementary information, as suggested by the reviewer, rather than showing it in the main manuscript (see Extended Data Fig. 7a,b).

Third, when comparing SFC across regions, multiple comparisons were corrected for using cluster-based permutation tests with a Bonferroni-corrected alpha level for 3 frontal regions, 2 cell groups, and 2 MTL seed areas (12 tests). We added this information to the results section to make it more accessible (lines 413-414; see also the “Statistics” section in the Methods).

8. The authors report firing rate correlations between PAC and category neurons. The authors excluded neurons that were better fit by category information or category models for this analysis? They should add a control looking at correlations with non PAC neurons, as described below. They do include a control with shuffled trial labels.

PAC neurons that were not also category neurons were not excluded from our original analysis (Fig. 6a in the original manuscript). We repeated this analysis after excluding category-selective neurons as requested. This revealed that noise correlations between category neurons and the subset of PAC neurons that were not category selective were significantly positive in both areas as in our original analysis (see Extended Data Fig. 9e). The mean correlation coefficient was qualitatively comparable to the analysis using all PAC neurons. Below, we further provide a figure that shows correlations among pairs of category neurons and non-PAC non-category neurons (Reviewer Fig. 2.8; same as Reviewer Fig. 1.4). Similar to being paired with PAC neurons, these pairs of neurons also showed significant positive correlations as an average across all pairs that were not significantly different from correlations among pairs of category and PAC neurons. We note that this finding is expected: noise correlations are a population-level phenomenon. We would therefore expect that most cell pairs recorded in the same area in close proximity would show similar sign and strength of noise correlations on average [as we find here] (Cohen and Kohn 2011) (see also our response to reviewer 1 point 4). However, on a trial-by-trial basis, we hypothesized that noise correlations between some pairs of neurons are more important for WM than between other pairs. This is indeed what we find: specifically the noise correlations between pairs of category and PAC neurons had a significant effect on decoding performance as well as on behavior (Fig. 6). This therefore shows that indeed the noise correlations of PAC neurons with other neurons were of functional relevance for WM processing. We clarified this reasoning in the discussion (lines 781-785). Please see our response to point 9 below for the requested additional control analysis.

Reviewer Figure 2.8: Mean correlations for PAC to category neuron pairs (162 pairs; cyan line) compared to 10,000 iterations of selecting the same number of random pairs for non-PAC non-category to category neuron pairs. Same as Reviewer Fig. 1.4.

9. The authors then go on to propose that PAC neurons support memory information by improving decodability of category even though they are not sensitive to category information per se. Were the overlapping (category) neurons excluded from this analysis? They find that decoding ability is improved with PAC neurons. Do PAC neurons improve decoding ability better than other non PAC non category neurons? Perhaps this was included and I missed it, since they do this for the RT correlations.

We thank the reviewer for suggesting these two new analyses, which we have added: Fig. 6d,e now show that specifically the noise correlations of PAC neurons improve decoding performance. To determine the specificity of noise correlations between PAC and category neurons on the decoding performance, we now compare maximal decoding performance as well as the maximal difference between intact and removed noise correlations before and after PAC neurons were removed from the ensemble of neurons (see Fig. 6d), and then compare this effect to taking out the same number of randomly chosen non-PAC cells from the population (averaged across 500 iterations.). This allowed us to observe the functional specificity of noise correlations introduced by PAC neurons on the encoded information content in the population. In the hippocampus, we found that when PAC neurons were removed from population, the maximal decoding performance in the intact noise correlations condition significantly decreased as compared to when PAC neurons were still part of the ensembles (tested across all sessions that had at least 2 neurons left after removing all PAC neurons). In the condition where noise correlations were removed, however, maximal decoding performance did not significantly differ. This replicates our previous finding that PAC neurons enhanced the decodability of category during the WM delay period, but only when noise correlations were intact. This was further corroborated by the fact that the maximal differences between intact and removed noise correlations were larger when PAC neurons were part of the ensembles

as compared to being removed (see Fig. 6e). These results also persisted when we only removed PAC neurons that were not also category neurons (see Extended Data Fig. 9g), which shows that the drop in decoding performance was not caused by removing category neurons from the ensembles, but, indeed, PAC neurons. Performing the same analysis after removing randomly chosen non-PAC neurons further confirmed the specific role of PAC cells in adding information-enhancing noise correlations: removing non-PAC neurons also caused a decrease in decoding performance when noise correlations were removed, indicating that the information they added was not due to the correlations. This shows that it was not specifically the noise correlations that influenced the decoding performance but (residual) category-specific activity that decreased information content independent of their noise correlations. Accordingly, we did not find a significant difference when testing the maximal difference between intact and removed noise correlations for ensembles for which we removed randomly selected neurons, which further supports this interpretation (Fig. 6e). In the Amygdala, removing PAC neurons from the ensemble also decreased the decoding performance. However, this effect was not specific to intact noise correlations (as we have seen already in our original analysis) (see Extended Data Fig. 9k). We added these new results to the revised manuscript.

10. The authors link their findings with proposals related to cognitive control and frontal control of WM—related activity in the MTL. This is really the core, novel result. In the context of this interpretation, did the authors test for interactions between category specific frontal lobe neurons and the MTL neurons, or PAC neurons in the MTL and other task sensitive neurons in the frontal cortex? The authors focus on the MTL after showing that PAC occurs at the LFP level in the MTL, but why were frontal neurons excluded from this subsequent analysis? Access to such populations is what differentiates this dataset from others and allows the authors to link with high impact work done in NHP that establishes key predictions for their analysis (see below). Related to this, when identifying PAC neurons, it seems that the theories the authors use to motivate this work would predict that phase information measured in the frontal cortex and not the MTL should be used in the models proposed for identifying PAC neurons. Wouldn't this be a more relevant analysis than local phase information in the MTL? Related to this, is focusing on theta appropriate? The key prediction from NHP WM data seems to be that item—relevant information being represented shifts from beta/alpha phase locking to gamma phase locking (see recent review from the Miller lab, summarizing several experiment). The authors do not analyze the data from this perspective.

We are pleased that the reviewer agrees that a core novel result is that we show how frontal control acts on WM-related activity in the MTL. We fully agree that this is a core result. We answer the questions asked as part of this issue in sequence below.

First, we added additional analysis regarding category specific neurons in the frontal lobe. For background, a main motivation for the present experiment was that in earlier work we found that neurons whose activity were indicative of working memory content during working memory maintenance were present in the MTL (hippocampus and amygdala) but – importantly – not in frontal cortex (dACC and pre-SMA) (Kamiński et al. 2017). Correspondingly, in that work, we showed that working memory content was decodable from the MTL but not the dACC or pre-SMA. In the present results, we confirm these findings: persistent activity during the delay period that is indicative of working memory content was only present in the MTL. However, we did not include these results in the initial version of the manuscript, which we have now done (see Extended Data Fig. 4). This new analysis shows that category neurons did not exist more than expected by chance in dACC and pre-SMA but do exist in vmPFC. While category neurons were present in vmPFC, they did not exhibit selectivity during the maintenance period (a property that extends to neurons selected in dACC and preSMA, which did not exist more than expected by chance but shown for completeness). Thus, we have not excluded frontal neurons at all - we considered them, but they did not exhibit the property of content selectivity during maintenance. Based on our earlier work (also see below) and our new findings shown here, we therefore hypothesized that working memory content is maintained in the MTL whereas medial frontal areas are involved in the control of maintenance processes. We realize we did not properly express this motivation and have adjusted our introduction to do so (lines 75-82).

Second, we found that theta-gamma PAC is related to WM processing only in the MTL and not the medial frontal areas we examined. This is compatible with other work: In another line of research, Daume et al. (Daume et al. 2017a, b), leveraging a whole-brain MEG design, have shown that during the WM delay period, theta-beta/gamma PAC can be observed in areas of the temporal, but not frontal lobe. These temporal areas, exerting elevated levels of *local* PAC, were further interacting with frontal areas via theta long-range phase synchronization when levels of cognitive control were high, suggesting that low-frequency theta oscillations control stimulus-related activities observable in higher-frequency. This MEG work further motivates our work, especially the selection of PAC neurons based on local PAC, and we are stressing this now in the introduction.

Third, the reviewer asks whether our focus on theta is appropriate. We note that the focus on theta in our work is data driven rather than a-priori. We found that PAC is prominent only in the theta-gamma range [we examined 2-14Hz for the modulating frequency band]. Also, all our SFC analyses include all frequencies between 2 and 150 Hz, with the theta-specificity a result rather than by design. For none of the SFC analyses have we observed significant effects within the beta range. Given this set of findings, focusing the main result of the paper on theta seems appropriate as it was, besides gamma, the only frequency band with significant effects throughout our analysis. We explicitly mention this now in the revised discussion (lines 655-660).

Lastly, the reviewer asks regarding the relationship of our findings to findings in NHP that show a shift between beta/alpha to gamma activity during working memory maintenance. Recent findings from the Miller lab (Lundqvist et al. 2016, 2018; Bastos et al. 2018; Miller et al. 2018; Buschman and Miller 2022) suggest that information in working memory is carried by brief bursts of gamma oscillations that correlate with enhanced spiking of information-carrying neurons in frontal cortex. Bursts of gamma power showed an anti-correlated relationship with bursts of alpha/beta power. These alpha/beta bursts did not coincide with information-carrying spiking and due to their anti-correlated relation were suggested to exert inhibitory control over gamma and neural spiking. Our finding that content-tuned category neurons relate to gamma oscillations when their preferred stimulus was maintained in memory strongly supports these reports. In line with their findings, interactions between spiking and gamma oscillations were especially strong when gamma power was high (see Extended Data Fig. 5d) – one of their key observations we can here confirm in the human hippocampus! Moreover, in our study, gamma power was modulated by an underlying theta rhythm and therefore also came in bursts rather than sustained increases during the delay period – which confirms another of their key observations. We note that the work by the Miller group, however, did not investigate phase relationships between spiking and oscillations. Their analyses focus on power relations between gamma, beta, and spiking only – not on phase-relationships (SFC or PAC) that are modulated by WM. Moreover, the reported power-power relationships did not persist on the single-trial level but were only observed as an average across trials (Lundqvist et al. 2016) – an important difference to the temporal precision of the spike-spike, spike-LFP, and LFP-LFP relationships revealed in our study. In our project, we thus provide novel insights into how content-carrying neurons relate to the phase of power-modulated gamma oscillations during the delay period in the hippocampus. Moreover, we also provide insights into how such information coding might be controlled by PAC-modulated neurons. Importantly, we emphasize that Miler and colleagues relate their observations at different stages of the task (i.e., encoding, maintenance, retrieval) to signaling “cognitive control” of working memory without directly manipulating levels of cognitive control at any point in their task. A core novel insight in our study is how cognitive control that is needed to support multi-item WM is implemented at the single-neuron level. Thus, while their NHP findings are very interesting and replicating their results in humans remains to be done, doing so is not the goal of our study. Rather, we study a different question: we focus on phase relationships and how these relate to different levels of cognitive control during the delay period. This reveals a new PAC-mediate mechanism for WM maintenance in humans that the NHP work has not touched upon at all (it remains an open question whether the hippocampus plays a similar role in NHPs). As asked in point 12 as well, we added this discussion to our revised manuscript (lines 643-660).

11. The authors also ignore the specific phases of spiking for PAC neurons, which in turn seems surprising given invocation of the Lisman models. An obvious prediction that should be included is the preferred phase

of spiking for different items in maintenance, which the authors are well positioned to analyze given the identification of category specific neurons.

As mentioned earlier, we removed the invocation of the Lisman model in our revised manuscript as testing its core predictions was not the goal of our study. Nevertheless, in our original analysis we provide indirect evidence that PAC neurons did not shift their spikes depending on where a given category was presented in the sequence. The Lisman model predicts that neurons that are tuned to a given stimulus shift their spiking with respect to an underlying theta rhythm depending on where in the sequence this stimulus is presented. Across all load 3 trials, where a given category can take any of the three possible positions, the range of corresponding theta phases should thus be more variable than compared to load 1 trials, where the position of a picture does not change. If PAC neurons shifted their preferred spiking phase depending on where in the sequence a given category was presented, theta SFC for PAC neurons should therefore be significantly weaker in load 3 than in load 1 since a more variable phase should lead to weaker SFC. Our results, however, show that this was not the case (now in Extended Data Fig. 7). Even when only testing PAC neurons that were also category neurons, theta SFC was not significantly weaker in load 3 than load 1 ($t(27) = -0.044$, $p = 0.964$). In addition to this, in the revised manuscript we now provide a more direct comparison of preferred theta phases between load 1 and load 3 trials. On average, there was no significant shift in the mean theta angle across all PAC neurons between load 1 and 3 (Extended Data Fig. 7c). We would like to note that in our study the order of the items was not task relevant. We suspect that whether the stimulus sequence is task-relevant or not could influence on whether spike shifting can be observed or not, and this question is therefore out of scope of the current project. Other studies that directly ask participants to keep in memory the order of stimuli are more suited to answer this question (see, e.g., Liebe et al. 2022).

12. Newer models of WM suggest that rather than persistent spiking activity present during maintenance, a composite attractor dynamic consisting of alternating beta/alpha vs gamma bursts occurring activation of ensembles linked with specific items. The authors should discuss their findings relative to these views and consider how phase organization of item—related activity in frequency bands outside of theta might affect their findings.

We have added the following paragraph to our discussion. Please also see our response to point 10 above.

Lines 643-660:

In non-human primates (Lundqvist et al. 2016, 2018, 2023; Bastos et al. 2018; Miller et al. 2018), spiking of frontal cortex neurons is most informative about WM content during brief bursts of gamma oscillations, which occur when beta activity is low. Our finding that the activity of content-tuned category neurons is related to gamma power and phase when their preferred stimulus was maintained in memory shows that a similar relationship is also present in the human hippocampus. In line with the NHP findings, interactions between WM content-related spiking and gamma rhythms in our study were especially strong when gamma power was high (Extended Data Fig. 5d). Moreover, gamma in our study was modulated by an underlying theta rhythm, showing that gamma activity was not monotonically sustained throughout the delay period. In contrast to the NHP findings, however, we did not observe information-carrying neurons that remained active during the maintenance period in frontal cortex. Indeed, no such neurons have been shown so far in human frontal cortex. A second notable difference is that in the hippocampus, low frequency modulations were related to the theta rather than the beta band as reported in NHP frontal cortex. It remains an open question of whether this difference in findings is due to a species difference, extent of training that the NHPs receive (Miller et al. 2022), or exact location of recordings within the frontal cortex.

Referee #3

The paper presents a study focused on understanding the role of Phase-Amplitude Coupling (PAC) neurons in working memory (WM) maintenance. The researchers identified PAC neurons whose spiking activity followed the interactions between theta phase and gamma amplitude during the maintenance period of a Sternberg WM task. They found that unlike category neurons, which displayed memoranda-specific persistent activity, the activity of PAC neurons was not related to WM content per se. Instead, the activity of PAC neurons in the hippocampus was related to the cognitive control processes that enable efficient and accurate maintenance of WM.

The researchers suggest that PAC neurons play a crucial role in cognitive control and shape WM fidelity through noise correlations with memoranda-selective persistently active neurons. This PAC-mediated interareal interaction might serve as a general mechanism for top-down control to influence bottom-up processes.

The paper aligns with previous studies showing that activity in the high gamma (70-140 Hz) frequency range reflects processing and WM maintenance of sensory information. However, it provides new insights into the role of PAC neurons in cognitive control and WM maintenance.

The paper presents compelling evidence for the role of PAC neurons in cognitive control and WM maintenance. It provides a new perspective on the neural mechanisms underlying WM maintenance and opens up new avenues for future research.

We thank the reviewer for their positive and constructive feedback on our work. The reviewer raised important points for which we are grateful and which we fully addressed as outlined below. In particular, we revised our methods and added the requested benchmark tests on the wavelets we used. For easier referability, we have numbered each point raised by reviewer 3. Please find our responses below.

1. Minor point: The phrase "nuisance factors" suggests that the authors know these factors may affect the result, but they are things to be diminished or maneuvered around. This may not be the impression that the authors are trying to convey. Simply removing "nuisance" resolves this. Otherwise, it sounds like platitudes.

We thank the reviewer for pointing this out to us. We removed this phrase from the manuscript.

2. Does Figure 5 imply that long-range coupling in the human MTL is primarily conducted through low frequencies (10 Hz and lower)? The low values for frequencies above 10 Hz suggest that there is no coupling. This seems to contradict the author's assertion that high gamma routes information (lines 530-

535). How do the authors reconcile the absence of SFC coupling in Figure 5 with the idea of routing? Or alternatively, do the authors think that their results better align with Mizuskei et al. (2009; PMID: 19874793), where regions of the MTL have significant independence? Please address.

The reviewer is correct, our results indeed suggest that long-range coupling between the hippocampus and vmPFC is primarily mediated by low frequencies, which in our task was stronger when needs of cognitive control were high. We note that this is in line with prior non-invasive work that also suggests that low frequencies mediate cognitive control (Miller and Cohen 2001; Liebe et al. 2012; Daume et al. 2017a), but so far this has not been shown relative to the hippocampus. We also note that we previously found that it was low frequencies that coordinated activity between a different part of the medial frontal cortex (dACC and preSMA) with the hippocampus during memory-based decision making (Minxha et al. 2020).

Regarding the statement that ‘high gamma routes information’, this statement was referring to routing of information from sensory cortex to the hippocampus during the presence of stimuli on the screen. In contrast, in our work, we are concerned with the maintenance of already encoded information after external sensory information has been routed through the visual system. It has been suggested that control mechanisms act upon the maintenance of encoded information through long-range low-frequency phase synchronization when requirements of control are high. This is what Fig. 5 shows: long-range SFC interactions between hippocampal PAC cells and the vmPFC during high memory load (“load 3”) are enhanced relative to low memory load in low frequencies. This effect was specific to PAC cells, whose spike timing was coordinated with the frontal cortex in low frequencies. We adjusted parts of the discussion to focus the discussion more on processing of encoded information and routing of cognitive control than of externally available sensory information (lines 627-628 and 716-719).

We thank the reviewer for pointing us to the work from Mizuseki et al. (Mizuseki et al. 2009). Our results could indeed also be viewed within the model framework proposed in that paper, in which theta oscillations (in our case from vmPFC instead of EC) act upon a subset of hippocampal cells (i.e., PAC neurons) to enable the interaction of local, within-hippocampal circuits (i.e., among category neurons). This could depict a mechanism that maintains WM content for the duration of the delay period through self-sustained activity (i.e., persistent activity). We have added this possibility to our paper (Discussion, lines 768-774).

Potential problems:

3. Among the first manuscripts that applied wavelet to neuroscience data, Tallon-Baudry et al. (1996) describe a trade-off between time and frequency in terms of resolution. For instance, the low-frequency wavelet would provide a precise realization of the frequency of the event with a reduced resolution of when the event occurred. However, as the wavelet narrows, it loses frequency resolution to obtain temporal

precision. From the information provided, it is challenging to determine the parameters that went into the wavelet (often, descriptions provide the Gaussian width around a central frequency). However, the 5 Hz spacing between 70-140 Hz or the 40 log spaced wavelets between 2-150 Hz, raise the specter that there may be a great deal of redundant capture across wavelets (e.g., multiple wavelets may overlap with 100 Hz, resulting in an overly convolved representation). Given the importance of the paper, the authors should provide benchmark tests of their decomposition in the supplement:

- a) To assess the quality of the wavelet decomposition, the authors should compare A) the original signal with B) the signal obtained after applying the CWT, and then the iCWT. The difference between the original signal and the reconstructed signal (i.e., the residuals) can provide valuable insights into the performance of the wavelet transform. If the residuals are small, this suggests that the CWT and iCWT are accurately capturing and reconstructing the key features of the signal. On the other hand, large residuals might indicate that important information is being lost or distorted. It's worth noting that the residuals might not be uniformly distributed across the signal. For example, the CWT might accurately capture the signal's behavior at certain times or frequencies but not others. Therefore, it could be useful to examine the residuals as a function of time or frequency to see if there are any patterns. However, keep in mind that even if the residuals are small, this doesn't necessarily mean that the CWT and iCWT are perfect. It's possible that the wavelet transform could introduce artifacts or distortions that aren't apparent when looking at the residuals alone.

We thank the reviewer for this suggestion, which we have addressed in two ways.

First, we revised the methods (lines 1210-1236) to specify the wavelets and their parameters used thoroughly (we apologize for this oversight in our original submission). The parameters we chose for our study follow those of earlier publications (Cohen and Donner 2013; Cohen 2014). In particular, addressing the issues raised by the reviewer, the number of cycles used for each wavelet increased with increasing frequency between 3 (at 2 Hz) and 10 cycles (at 150 Hz; 40 log-spaced steps). This ensures a better temporal precision for lower frequency and a better frequency precision for higher frequency wavelets as compared to an equal number of cycles across all wavelets (Cohen 2014). Moreover, we used log-spaced wavelet frequencies. This approach is commonly used in the field given the log-spaced bandwidth of neuronal oscillations that is pervasive throughout most studies of local field potentials (Buzsaki 2006).

Second, as requested we added new analysis to the supplement to validate the quality of the wavelet composition. In Extended Data Fig. 1h-k we now provide plots that show how well the signal can be reconstructed from our wavelet transform as a function of time and frequency. Using linear models to test how well the reconstructed signal can predict the original signal, we computed R-squared values across different time windows and frequency bands (see Methods). The results show that we can reconstruct the original signal after applying our continuous wavelet transform in all frequency ranges and time points with

high accuracy (Extended Data Fig. 1k). The slight decrease in R-squared in the lowest and highest frequency ranges, i.e., around 2 and 150 Hz, are expected given that frequencies lower than 2 Hz and higher than 150 Hz were not represented by our wavelets.

- b) Concatenate all the scaled and translated versions of the mother wavelet into a single time series and then apply the wavelet transform to that time series. This will create a sort of “wavelet spectrogram” that would allow you to visualize how the frequency content of the wavelet changes with scale (which is related to frequency) and position (which is related to time). This could provide valuable insights into the temporal and spectral characteristics of the wavelet. For example, you could see how the wavelet’s frequency content changes with scale, or how well the wavelet localizes different frequencies in time. This could help you understand why the wavelet transform gives the results it does when applied to your data.

As requested, we added information on the temporal and spectral characteristics of the wavelets we used. New figure panels Extended Data Fig. 1h,i show the frequency bandwidth as well as the temporal smoothing for each wavelet. With regard to the reviewer’s concern that there might be a strong overlap at around 100 Hz, leading to an overly convolved representation of the signal, Extended Data Fig. 1h (right) shows that the overlap of wavelets at around 100 Hz was minimal. This analysis shows that the characteristics of the wavelets we used are appropriate to accurately represent the spectral content of our data in the frequency ranges of interest.

- c) Please provide an example of a raw Local Field Potential (LFP) trace of 1-2 seconds from a subject, perhaps in Figure 2 or as a supplemental figure, along with a Log-Log power spectral density using a fast-Fourier transform with a temporal window of support of 1 second or more. This would provide insights into the general shape of power spectra. Also, does the $1/f$ slope of this power spectra change as a function of memory performance/load?

We added an example of a raw LFP in Extended Data Fig. 1f and Fig. Extended Data Fig. 1g provides the power spectrum of that signal to show the overall shape of the power distribution. The slope of log-log power spectra did not differ between load1 and load3 trials in hippocampal channels ($n = 586$ channels; mean slope -1.7526 ± 0.3902 vs -1.7517 ± 0.3928 , $t(585) = -0.86$, $p = 0.39$, paired permutation t-test).

4. I am uncomfortable with using channels or cells as samples as these describe aspects of the data that cannot be considered independent. How the author conducted their statistics makes it vulnerable to false positives. Using a sample size of subjects would be appropriate (Aarts et al., 2014). Moreover, in some conditions, the distributions do not appear to be parametric (e.g., Figure 2b), suggesting that a t-test may

be inappropriate. Creating shuffle distributions seems artificially stacked in the author's favor to find significance. I am not certain why this practice is done, but it assumes that the null distribution describes the situation "what if the brain is completely random without any correlation"? Creating a shuffled distribution allows the authors to find statistical significance in Figure 6, when on a biological level, correlation of 0.02 offers that knowing one variable has little predictive value on the other. Therefore, touting the positive correlation (lines 440-445) seems like a gross misrepresentation of what is actually occurring. This analysis, again, was conducted on cell pairs that seem overpowered, and the distributions may not be normal. Statistical issues seem to persist throughout the manuscript (e.g., Figure 3).

We appreciate the careful consideration of potential statistical issues in our manuscript, which we are happy to address. Below, we address each issue raised. Jointly, we are confident that we addressed all statistical concerns conclusively.

First, the reviewer is certainly correct that a cell-by-cell analysis makes assumptions of independence that might not be warranted. We address this concern by adding new analysis based on multi-level random effects models and conventional statistics across sessions after averaging channels or neurons within each session (Aarts et al. 2014). All the main effects reported in our manuscript show significant effects when tested on a session-based level (see also response to reviewer 2 point 1). We added these new results to the manuscript in Extended Fig. 5b,g, Extended Fig. 8b, and Extended Fig. 9i.

Second, the reviewer is correct that the distributions for some of the tested variables were non-parametric, and that the application of standard t-tests and ANOVAs would not be appropriate in those cases. However, we would like to emphasize that none of our reported results are based on parametric tests (this was already the case in the original submission). Rather, all our statistics are computed using non-parametric (cluster-based) permutations tests (using the EEGLab function `statcond.m` or FieldTrip's `ft_freqstatistics.m`) that do not assume a specific distribution (Maris and Oostenveld 2007). We revised our methods section to make this clearer to the reader (lines 1425-1428). For clarity, we now always refer to "permutation-based ANOVA" and "permutation-based t-test" to make clear that we are not using the parametric versions of these tests.

Third, we note that we constructed all our shuffle distributions carefully so that only the effect of relevance would be expected to be destroyed, leaving all others in place. For example, for assessing the significance of PAC differences between loads, we use surrogates where phase and amplitude information is taken from random subsets of trials of the same load. This way of creating the null distribution preserves everything about the signal except for PAC differences due to load. In particular, it preserves any other differences that may arise due to load such as systematic power or phase differences or correlations within the data across time. Similarly, for the construction of the null distribution to assess the significance of noise correlations (Fig. 6a), we only shuffle the order of the trials within a given condition. This approach leaves all other properties of the signal intact and ensures that only the parameter of interest is destroyed (which

here is the correlation when recorded simultaneously in the same trial). All other parameters including the interspike-interval distributions, firing rate distributions, temporal relations to task events, etc., remain intact and are therefore not random. Therefore, the null distribution is not assuming 'the brain is random' as signal correlations are preserved and is appropriately powered because the level of analysis is that of simultaneously recorded pairs of cells. In our view, our way of constructing the null distributions is one of the most conservative and the most appropriate to test the relevance of a given parameter (Pipa and Grün 2003). We adjusted our results (lines 465-470) and methods (lines 1146-1152; 1333-1339) to more clearly point out how we constructed our null distributions.

Fourth, the reviewer asked about the biological significance of the strength of noise correlations between pairs of cells that we found (which was approximately $r=0.02$). We note that there is a large literature that shows both experimentally and theoretically that noise correlations of this magnitude are both common and highly biologically significant. For example, a review of a large number of studies shows the most common strength of such correlations to be in the same range as we report (Cohen and Kohn 2011), and theoretical work (Zohary et al. 1994) shows that correlations of this strength have substantial (usually information limiting) effects at the population level. We added this reasoning to the discussion to make this point clearer (lines 807-813).

Lastly, regarding the cell pairs, we would like to point out that we further compare our observed effect against 10,000 iterations of randomly selecting the same number of nonPAC-CAT cell pairs as being available for PAC-CAT cell pairs. Even if the effect was overpowered by the number of cell pairs (which we don't believe as the effect persists also on the session level, see Extended Data Fig. 9i), the effect should be the same for the high number of randomly selected cell pairs. Since the observed effect is stronger for PAC-CAT pairs than randomly selected nonPAC-CAT pairs, though, we are confident that our results are not just inflated by the number of cell pairs tested.

5. The use of a z-score surrounding the Modulation Index is also puzzling. Usually, the raw values of the modulation indices are depicted (perhaps the authors may wish to include this?). Providing the z-score of the values relative to a surrogate distribution may artificially inflate the statistical power, resulting in significance when the values may be small, leaning towards biologically insignificant.

As the reviewer notes, we stated the modulation indices in our paper in units of z-scores. This approach allowed us to express the strength of the modulation indices relative to that seen in a null distribution that preserves all other aspects of the signal except the parameter under investigation (here, load; see our response to point 4 above for how the surrogate distribution was computed). In our opinion, it is crucial to show the MI after z-scoring using within-condition surrogates due to the intrinsic bias of the modulation index towards lower frequencies (Aru et al. 2015; Jones 2016; Vaz et al. 2017). Slow frequencies are more

vulnerable to non-specific correlations to high-frequency power due to non-stationarities in the LFP signal, such as for example due to phase resets at stimulus onset. Comparing raw modulation indices to trial-shuffled surrogates will remove any PAC that is caused by such non-specific interactions (as they should be present in every trial; for a more detailed discussion on this, see Aru et al., 2015). We added this rationale to the methods section (lines 1146-1152). In addition, raw MI values are not normalized with respect to effects that are still present in the surrogates, which include effects due to power changes as a function of time. For this reason, it is advisable (and indeed common practice in the field) to normalize the modulation index using surrogate data to compare PAC across conditions, frequencies, and channels (Vaz et al. 2017), and many studies we are aware of show MI scores in units of z-scores, including the seminal work by Canolty et al. (Canolty et al. 2006) that we follow here.

Nevertheless, as requested by the reviewer, we now also include the raw values of an exemplar channel in Fig. 2d. In addition, in Extended Data Fig. 3f we now show the results of testing PAC differences between the load conditions in the hippocampus using the raw MI. Reviewer figure 3.5 below includes tests on PAC differences between the load conditions using the raw MI in each area. Overall, the results were comparable to our original analysis, including the key result that PAC differs significantly between loads only in the hippocampus, with load 3 exhibiting lower MIs compared to load 1. The raw values of the MI are known to be small (see Tort et al. 2010), but that does not mean that they are biologically insignificant. Rather, the numerical value of the raw MI is small because of how the modulation index is defined and the choice of parameters made in estimating MI (such as the bin size). Given this, small values of the raw MI are expected rather than surprising. The MI is scaled between 0 and 1. A uniform distribution of power values across all phase bins leads to a value of 0. The value 1 can only be reached when the power distribution across theta phase follows a Dirac delta function, where one phase bin carries all power and all other phase bins are 0. If bin sizes are small (as in our analysis, where bin size=20 deg), this is exceedingly unlikely. This definition also shows that the raw MI value depends on the bin size chosen, making the raw MI value uninterpretable. z-scoring the MI values relative to surrogates removes this caveat and makes the numerical values interpretable. In biological data, even strong PAC visible in the raw data results in MI values within the range that we observe also in humans (see Fig. 2d for an example, for which theta-gamma PAC is easily visible in the raw data and which had a raw MI of 0.5×10^{-3} and a z-score of 12.2 as an average across both conditions). Indeed, the values that we observed in our study are very similar to other reports of theta-gamma PAC in physiological signals (e.g., Tort et al. 2009; Vaz et al. 2017).

Reviewer Figure 3.5. Control analysis, showing the comparison of theta-gamma PAC between the load conditions in each area using the raw modulation index. Significant differences between the conditions were only found in the Hippocampus.

References

- Aarts E, Verhage M, Veenliet JV, et al (2014) A solution to dependency: using multilevel analysis to accommodate nested data. *Nat Neurosci* 17:491–496. <https://doi.org/10.1038/nn.3648>
- Al-Daffaie K, Khan S (2017) Logistic regression for circular data. *AIP Conference Proceedings* 1842:030022. <https://doi.org/10.1063/1.4982860>
- Aru J, Aru J, Priesemann V, et al (2015) Untangling cross-frequency coupling in neuroscience. *Curr Opin Neurobiol* 31:51–61. <https://doi.org/10.1016/j.conb.2014.08.002>
- Bastos AM, Loonis R, Kornblith S, et al (2018) Laminar recordings in frontal cortex suggest distinct layers for maintenance and control of working memory. *Proc National Acad Sci* 115:1117–1122. <https://doi.org/10.1073/pnas.1710323115>
- Buschman TJ, Miller EK (2022) Working Memory Is Complex and Dynamic, Like Your Thoughts. *J Cogn Neurosci* 35:17–23. https://doi.org/10.1162/jocn_a_01940
- Buzsaki G (2006) *Rhythms of the brain*. Oxford University Press., New York
- Canolty RT, Edwards E, Dalal SS, et al (2006) High Gamma Power Is Phase-Locked to Theta Oscillations in Human Neocortex. *Science* 313:1626–1628. <https://doi.org/10.1126/science.1128115>
- Cohen M (2014) *Analyzing Neural Time Series Data: Theory and Practice*. MIT Press, Cambridge
- Cohen MR, Kohn A (2011) Measuring and interpreting neuronal correlations. *Nat Neurosci* 14:811–819. <https://doi.org/10.1038/nn.2842>

- Cohen MX, Donner TH (2013) Midfrontal conflict-related theta-band power reflects neural oscillations that predict behavior. *J Neurophysiol* 110:2752–2763. <https://doi.org/10.1152/jn.00479.2013>
- Colgin LL (2016) Rhythms of the hippocampal network. *Nat Rev Neurosci* 17:239–249. <https://doi.org/10.1038/nrn.2016.21>
- Colgin LL, Denninger T, Fyhn M, et al (2009) Frequency of gamma oscillations routes flow of information in the hippocampus. *Nature* 462:353–357. <https://doi.org/10.1038/nature08573>
- Daume J, Graetz S, Gruber T, et al (2017a) Cognitive control during audiovisual working memory engages frontotemporal theta-band interactions. *Sci Rep-uk* 7:12585. <https://doi.org/10.1038/s41598-017-12511-3>
- Daume J, Gruber T, Engel AK, Fries U (2017b) Phase-Amplitude Coupling and Long-Range Phase Synchronization Reveal Frontotemporal Interactions during Visual Working Memory. *J Neurosci* 37:313–322. <https://doi.org/10.1523/jneurosci.2130-16.2017>
- Fernandez-Ruiz A, Oliva A, Nagy GA, et al (2017) Entorhinal-CA3 Dual-Input Control of Spike Timing in the Hippocampus by Theta-Gamma Coupling. *Neuron* 93:1213–1226.e5. <https://doi.org/10.1016/j.neuron.2017.02.017>
- Fries P (2005) A mechanism for cognitive dynamics: neuronal communication through neuronal coherence. *Trends Cogn Sci* 9:474–480. <https://doi.org/10.1016/j.tics.2005.08.011>
- Fries P (2015) Rhythms for Cognition: Communication through Coherence. *Neuron* 88:220–235. <https://doi.org/10.1016/j.neuron.2015.09.034>
- Heusser AC, Poeppel D, Ezzyat Y, Davachi L (2016) Episodic sequence memory is supported by a theta–gamma phase code. *Nat Neurosci* 19:1374–1380. <https://doi.org/10.1038/nn.4374>
- Johnson EL, Adams JN, Solbakk A-K, et al (2018) Dynamic frontotemporal systems process space and time in working memory. *PLoS Biol* 16:e2004274. <https://doi.org/10.1371/journal.pbio.2004274>
- Jones SR (2016) When brain rhythms aren't 'rhythmic': implication for their mechanisms and meaning. *Curr Opin Neurobiol* 40:72–80. <https://doi.org/10.1016/j.conb.2016.06.010>
- Kamiński J, Brzezicka A, Mamelak AN, Rutishauser U (2020) Combined Phase-Rate Coding by Persistently Active Neurons as a Mechanism for Maintaining Multiple Items in Working Memory in Humans. *Neuron* 106:256–264.e3. <https://doi.org/10.1016/j.neuron.2020.01.032>
- Kamiński J, Sullivan S, Chung JM, et al (2017) Persistently active neurons in human medial frontal and medial temporal lobe support working memory. *Nat Neurosci* 20:590–601. <https://doi.org/10.1038/nn.4509>
- Kohn A, Coen-Cagli R, Kanitscheider I, Pouget A (2015) Correlations and Neuronal Population Information. *Annu Rev Neurosci* 39:1–20. <https://doi.org/10.1146/annurev-neuro-070815-013851>
- Lega B, Burke J, Jacobs J, Kahana MJ (2016) Slow-Theta-to-Gamma Phase–Amplitude Coupling in Human Hippocampus Supports the Formation of New Episodic Memories. *Cereb Cortex* 26:268–278. <https://doi.org/10.1093/cercor/bhu232>
- Liebe S, Hoerzer GM, Logothetis NK, Rainer G (2012) Theta coupling between V4 and prefrontal cortex predicts visual short-term memory performance. *Nat Neurosci* 15:456–462. <https://doi.org/10.1038/nn.3038>
- Liebe S, Niediek J, Pals M, et al (2022) Phase of firing does not reflect temporal order in sequence memory of humans and recurrent neural networks. *Biorxiv* 2022.09.25.509370. <https://doi.org/10.1101/2022.09.25.509370>

- Lundqvist M, Herman P, Warden MR, et al (2018) Gamma and beta bursts during working memory readout suggest roles in its volitional control. *Nat Commun* 9:394. <https://doi.org/10.1038/s41467-017-02791-8>
- Lundqvist M, Rose J, Herman P, et al (2016) Gamma and Beta Bursts Underlie Working Memory. *Neuron* 90:152–164. <https://doi.org/10.1016/j.neuron.2016.02.028>
- Maris E, Oostenveld R (2007) Nonparametric statistical testing of EEG- and MEG-data. *J Neurosci Meth* 164:177–190. <https://doi.org/10.1016/j.jneumeth.2007.03.024>
- Miller EK, Cohen JD (2001) AN INTEGRATIVE THEORY OF PREFRONTAL CORTEX FUNCTION. *Annu Rev Neurosci* 24:167–202. <https://doi.org/10.1146/annurev.neuro.24.1.167>
- Miller EK, Lundqvist M, Bastos AM (2018) Working Memory 2.0. *Neuron* 100:463–475. <https://doi.org/10.1016/j.neuron.2018.09.023>
- Minxha J, Adolphs R, Fusi S, et al (2020) Flexible recruitment of memory-based choice representations by the human medial frontal cortex. *Science* 368:eaba3313. <https://doi.org/10.1126/science.aba3313>
- Mizuseki K, Sirota A, Pastalkova E, Buzsáki G (2009) Theta Oscillations Provide Temporal Windows for Local Circuit Computation in the Entorhinal-Hippocampal Loop. *Neuron* 64:267–280. <https://doi.org/10.1016/j.neuron.2009.08.037>
- Panzeri S, Moroni M, Safaai H, Harvey CD (2022) The structures and functions of correlations in neural population codes. *Nat Rev Neurosci* 1–17. <https://doi.org/10.1038/s41583-022-00606-4>
- Pipa G, Grün S (2003) Non-parametric significance estimation of joint-spike events by shuffling and resampling. *Neurocomputing* 52:31–37. [https://doi.org/10.1016/s0925-2312\(02\)00823-8](https://doi.org/10.1016/s0925-2312(02)00823-8)
- Schomburg EW, Fernandez-Ruiz A, Mizuseki K, et al (2014) Theta Phase Segregation of Input-Specific Gamma Patterns in Entorhinal-Hippocampal Networks. *Neuron* 84:470–485. <https://doi.org/10.1016/j.neuron.2014.08.051>
- Tort ABL, Komorowski R, Eichenbaum H, Kopell N (2010) Measuring Phase-Amplitude Coupling Between Neuronal Oscillations of Different Frequencies. *J Neurophysiol* 104:1195–1210. <https://doi.org/10.1152/jn.00106.2010>
- Tort ABL, Komorowski RW, Manns JR, et al (2009) Theta-gamma coupling increases during the learning of item-context associations. *Proc National Acad Sci* 106:20942–7. <https://doi.org/10.1073/pnas.0911331106>
- Vaz AP, Yaffe RB, Wittig JH, et al (2017) Dual origins of measured phase-amplitude coupling reveal distinct neural mechanisms underlying episodic memory in the human cortex. *Neuroimage* 148:148–159. <https://doi.org/10.1016/j.neuroimage.2017.01.001>
- Wang DX, Schmitt K, Seger S, et al (2021) Cross-regional phase amplitude coupling supports the encoding of episodic memories. *Hippocampus* 31:481–492. <https://doi.org/10.1002/hipo.23309>
- Zohary E, Shadlen MN, Newsome WT (1994) Correlated neuronal discharge rate and its implications for psychophysical performance. *Nature* 370:140–143. <https://doi.org/10.1038/370140a0>

Reviewer Reports on the First Revision:

Referees' comments:

Referee #1 (Remarks to the Author):

The authors have done a fantastic job at addressing my initial concerns, performing extensive additional analyses and controls, improving readability of the manuscript and refining interpretation of their findings. Overall, the paper is markedly improved and, I believe, is poised to make a fundamental and important impact to the field. In particular, the authors have added new panels and analyses that better illustrate why PAC differs across working memory loads and further illustrate the distribution of gamma power as a function of theta phase. They also added new analyses that further quantify how noise correlations due to PAC neurons change the population's geometry. As part of their revision, the authors confirmed that differences in geometry on incorrect trials are not explained by global changes in population activity further strengthening their findings. Beyond these measures, they have also updated the introduction and discussion as well as updated many of their figures which are now clearer and provide a more detailed explanation of how the activity of PAC neurons relate to that of category neurons in the hippocampus. Finally, I commend the authors for making their data available and for providing well-annotated and accessible codes which will enhance the impact of the paper and its use. This paper represents a true tour de force and an important addition to our understanding of working memory in humans. I fully support publication of the paper and have no further comments.

Referee #2 (Remarks to the Author):

The authors have done a comprehensive and impressive job responding to criticisms with additional control analyses and creative presentation of the data.

I have only a couple of minor points. The axis labels for phase versus amplitude seem to be flipped for reviewer figure 1

Please include the distribution of the magnitude of the noise correlations in Figure 6 so readers can see how this differ from published examples in other populations.

The authors should be congratulated on a unique and comprehensive paper.

EDITOR'S NOTE: As noted, we could not obtain a re-report from R3 so asked R1, who has overlapping statistical expertise, to comment specifically on R3's statistical comments. I'm including R1's responses to your responses to R3 below, and ask that you add at least one of the analyses recommended.

Some of the comments I've summarized, and some I've included verbatim.

R1's comments on your response to R3's statistical concerns:

R1 felt that you used appropriate statistics and included a number of well-designed and valid controls to demonstrate your effects. However, R1 also did not feel that R3's original concern about sample size and power were fully addressed. R3 says:

"For example, performing a back-of-the-envelope calculation and from what I can tell from the numbers provided (e.g., 137/586 significant channels in Fig 2b), the authors would need a small

sample of 18 channels to obtain an 80% power with an alpha of 0.05. In other words, while the results are significant and well-controlled (e.g., comparing PAC-CAT pairs to randomly selected nonPAC-CAT pairs), the authors would have a good likelihood of identifying at least some modulation by chance (even for lower alphas). This point is also relevant given the relatively small difference in modulation across memory loads, although this is not necessarily unexpected for this type of cognitive process and similar degrees of modulation have been observed previously.

There are a number of ways to potentially address these points which could help further satisfy concerns raised by the reviewer. First, it would be helpful to perform a bootstrap analysis whereby the electrodes are randomly subsampled (e.g., from the 586) to provide an estimated rate of false positive errors. This should be replicated across areas, frequency bands and conditions. They can also consider a Neyman-Pearson approach to further confirm that their test is optimally powered. Second, it could be helpful for the authors to estimate the sample size of subjects needed to support their statistics (as recommended by reviewer #3 in comment 4). Although the authors estimated this across sessions, it would be more convincing for them to perform their analyses across subjects ($n = 36$). Third, while the authors performed a permutation-based t-test and ANOVA, it would also be useful to perform a rank-sum or Kruskal-Wallis test on their raw data (e.g., Reviewer Figure 3.5). To further validate their findings, they can also perform a decoding approach in which they use raw PAC values to determine the memory load from data not used for model training. Demonstrating significant decoding would provide strong evidence that the effects are in fact biologically plausible. Finally, while the authors use a modulation index (MI) for their analyses and cite two articles by Tort et al from 2009 and 2010, it is not a widely used metric and it's uncommon to z-score such indices. It's also not fully clear how their permutation test was performed. Here, the authors should perform a paired subtraction across conditions and compare these differences to zero when calculating their t-statistics."

Author Rebuttals to First Revision:

Reply to reviewers and editor for Daume et al. submission 2023-05-07411B

Color code: Original (black), Our reply (blue)

We very much thank the reviewers for their positive feedback and their full support of our manuscript. We are very grateful for all their thoughtful ideas and suggestions, and we truly think that their contributions substantially improved our manuscript. We feel very honored to see that they agree with us that our paper will have an important impact on the field. Below, we outline our point-by-point response to the reviewers.

Referee #1

The authors have done a fantastic job at addressing my initial concerns, performing extensive additional analyses and controls, improving readability of the manuscript and refining interpretation of their findings. Overall, the paper is markedly improved and, I believe, is poised to make a fundamental and important impact to the field. In particular, the authors have added new panels and analyses that better illustrate why PAC differs across working memory loads and further illustrate the distribution of gamma power as a function of theta phase. They also added new analyses that further quantify how noise correlations due to PAC neurons change the population's geometry. As part of their revision, the authors confirmed that differences in geometry on incorrect trials are not explained by global changes in population activity further strengthening their findings. Beyond these measures, they have also updated the introduction and discussion as well as updated many of their figures which are now clearer and provide a more detailed explanation of how the activity of PAC neurons relate to that of category neurons in the hippocampus. Finally, I commend the authors for making their data available and for providing well-annotated and accessible codes which will enhance the impact of the paper and its use. This paper represents a true tour de force and an important addition to our understanding of working memory in humans. I fully support publication of the paper and have no further comments.

We thank Reviewer 1 for their kind words and their full support of our work. We are especially thankful for the suggestion to analyze the geometrical changes caused by the noise correlations, which substantially enhanced the impact of the results. This strongly improves our understanding of the underlying structure of the neural response and the role they play in working memory.

Referee #2

The authors have done a comprehensive and impressive job responding to criticisms with additional control analyses and creative presentation of the data.

We thank Reviewer 2 for their very thorough assessment of our results and the many thoughtful and interesting suggestions. We think that especially adding the PAC control analyses and a better description of frontal activity improved the manuscript. We are pleased to see that Reviewer 2 now fully supports our manuscript.

I have only a couple of minor points. The axis labels for phase versus amplitude seem to be flipped for reviewer figure 1.

We apologize for this oversight. The reviewer is correct that the labels are flipped. The x-axis should say *frequency-for-phase* while the y-axis should denote *frequency-for-amplitude*. Since the figure is meant to be a reviewer figure only and will not appear in the paper, however, we did not further address this point.

Please include the distribution of the magnitude of the noise correlations in Figure 6 so readers can see how this differ from published examples in other populations.

As requested by the reviewer, we added the distribution of the magnitude of the noise correlations between pairs of PAC and category neurons to Fig. 6a. We had previously moved this panel to Extended Data Figure 9a but agree that showing this distribution in the main figure helps to compare our results to other publications and have thus moved this panel back to the main figure.

The authors should be congratulated on a unique and comprehensive paper.

Editor

As noted, we could not obtain a re-report from R3 so asked R1, who has overlapping statistical expertise, to comment specifically on R3's statistical comments. I'm including R1's responses to your responses to R3 below, and ask that you add at least one of the analyses recommended.

Some of the comments I've summarized, and some I've included verbatim.

R1's comments on your response to R3's statistical concerns:

R1 felt that you used appropriate statistics and included a number of well-designed and valid controls to demonstrate your effects. However, R1 also did not feel that R3's original concern about sample size and power were fully addressed. R3 says:

"For example, performing a back-of-the-envelope calculation and from what I can tell from the numbers provided (e.g., 137/586 significant channels in Fig 2b), the authors would need a small sample of 18 channels to obtain an 80% power with an alpha of 0.05. In other words, while the results are significant and well-controlled (e.g., comparing PAC-CAT pairs to randomly selected nonPAC-CAT pairs), the authors would have a good likelihood of identifying at least some modulation by chance (even for lower alphas). This point is also relevant given the relatively small difference in modulation across memory loads, although this is not necessarily unexpected for this type of cognitive process and similar degrees of modulation have been observed previously.

There are a number of ways to potentially address these points which could help further satisfy concerns raised by the reviewer. First, it would be helpful to perform a bootstrap analysis whereby the electrodes are randomly subsampled (e.g., from the 586) to provide an estimated rate of false positive errors. This should be replicated across areas, frequency bands and conditions. They can also consider a Neyman-Pearson approach to further confirm that their test is optimally powered. Second, it could be helpful for the authors to estimate the sample size of subjects needed to support their statistics (as recommended by reviewer #3 in comment 4). Although the authors estimated this across sessions, it would be more convincing for them to perform their analyses across subjects ($n = 36$). Third, while the authors performed a permutation-based t-test and ANOVA, it would also be useful to perform a rank-sum or Kruskal-Wallis test on their raw data (e.g., Reviewer Figure 3.5). To further validate their findings, they can also perform a decoding approach in which they use raw PAC values to determine the memory load from data not used for model training. Demonstrating significant decoding would provide strong evidence that the effects are in fact biologically plausible. Finally, while the authors use a modulation index (MI) for their analyses and cite two articles by Tort et al from 2009 and 2010, it is not a widely used metric and it's uncommon to z-score such indices. It's also not fully clear how their permutation test was performed. Here, the authors should perform a paired subtraction across conditions and compare these differences to zero when calculating their t-statistics."

We thank Reviewer 1 for assessing our response to Reviewer 3. We are delighted to see that Reviewer 1 agrees that our statistics are appropriate and well controlled. The reviewer suggested additional analyses. In the revised version of the manuscript, we have now included several of the suggested additional controls.

First, as the reviewer suggested, we performed a bootstrap analysis to estimate the rate of false positive errors in selecting channels with PAC. We plot the result by indicating the 99th percentile of the

null distribution for the proportion of significant PAC channels selected (see Fig. 2b, horizontal lines). The upper bound of the null distribution is at ~7%, showing that the false positive rate is well controlled. We assessed the underlying null distribution per area and frequency bin as suggested and then averaged the chance level across the same phase-frequency bin as used for the analysis shown (theta to high-gamma frequency). We note that the selection of PAC channels was done across both loads to avoid biasing our subsequent load condition comparison (Fig. 2c). We therefore also computed the chance levels on averages across both loads. In summary, this new control analysis further underscores, as stated in our original analysis, that the selection of PAC channels is well above chance in the hippocampus, amygdala, and the vmPFC, but not in the dACC and pre-SMA.

Second, we changed all group statistics from a session- to a patient-level, as the reviewer suggested in point 2 (modified figures are Extended Data Figures 2b, 5b,g, 8b, and 9i which all present patient-level statistics now). We agree that this now better aligns with the original suggestion by Reviewer 3, and we apologize for this slight oversight. Each of the new results at the patient-level is qualitatively comparable to our initial group statistics at the session-level. The interpretation of our results therefore did not change.

Third, we note that our permutation statistics are already performed as the reviewer suggests we do under the point “**It’s also not fully ...**”. We revised our methods section (lines 1531-1540) to clarify how we performed the permutation statistics. We confirm that for all our permutation-based t-statistics (using the EEGLab permutation-statistics function `statcond.m` and option ‘perm’) are equivalent to paired subtractions across conditions and comparisons of these differences to zero, as suggested by the reviewer. Regarding the concern that z-scoring MI indices is not commonly done: we note that in the original revision, we added plots that in addition also show the raw MI values (see Extended Data Fig. 3f), thereby now showing both z-scored and raw value versions of the MI.

Reviewer Reports on the Second Revision:

Referees' comments:

Referee #1 (Remarks to the Author):

I greatly appreciate all the work and effort that the authors have put into these added analyses. They have gone beyond what is needed to demonstrate the consistency and robustness of their results, and I believe that they have fully addressed R3's remaining concerns. Overall, the paper is much improved and provides important new insights into working memory in humans. I have no further suggestions and strongly support its publication.